# NEURAL LOGISTIC BANDITS

## ABSTRACT

We study the problem of *neural logistic bandits*, where the main task is to learn an unknown reward function within a logistic link function using a neural network. Existing approaches either exhibit unfavorable dependencies on $\kappa$, where $1/\kappa$ represents the minimum variance of reward distributions, or suffer from direct dependence on the feature dimension $d$, which can be huge in neural network–based settings. In this work, we introduce a novel Bernstein-type inequality for self-normalized vector-valued martingales that is designed to bypass a direct dependence on the ambient dimension. This lets us deduce a regret upper bound that grows with the *effective dimension* $\widetilde{d}$, not the feature dimension, while keeping a minimal dependence on $\kappa$. Based on the concentration inequality, we propose two algorithms, NeuralLog-UCB-1 and NeuralLog-UCB-2, that guarantee regret upper bounds of order $\widetilde{O}(\widetilde{d}\sqrt{\kappa T})$ and $\widetilde{O}(\widetilde{d}\sqrt{T/\kappa})$, respectively, improving on the existing results. Lastly, we report numerical results on both synthetic and real datasets to validate our theoretical findings.

## 1 INTRODUCTION

Contextual bandits form the foundation of modern sequential decision-making problems, driving applications such as recommendation systems, advertising, and interactive information retrieval Li et al. (2010). Although upper confidence bound (UCB)–based linear contextual bandit algorithms achieve near-optimal guarantees when rewards are linear in the feature vector Abbasi-Yadkori et al. (2011), many real-world scenarios exhibit nonlinear reward structures that demand more expressive models. Motivated by this, several approaches have been developed to capture complex reward functions that go beyond the linear case, such as those based on generalized linear models Filippi et al. (2010); Li et al. (2017), reproducing kernel Hilbert space Srinivas et al. (2010); Valko et al. (2013), and deep neural networks Riquelme et al. (2018); Zhou et al. (2020).

Among these settings, *logistic bandits* are particularly relevant when the reward is binary (e.g., click vs. no-click); the random reward in each round follows a Bernoulli distribution, whose parameter is determined by the chosen action. Extending logistic bandits via a neural network-based approximation framework, we consider *neural logistic bandits* and address two significant challenges: (i) handling the nonlinearity of the reward function, characterized by the worst-case variance of a reward distribution $1/\kappa$ where $\kappa$ scales exponentially with the size of the decision set, and (ii) controlling the dependence on the feature dimension $d$, which can be extremely large due to the substantial number of parameters in deep neural networks.

For logistic bandits, Faury et al. (2020) introduced a variance-adaptive analysis by incorporating the true reward variance of each action into the design matrix. This avoids using a uniform worst-case variance bound of $1/\kappa$ for all actions, thus reducing the dependency of the final regret on $\kappa$. Building on this, Abeille et al. (2021) achieved the best-known $\kappa$ dependence. However, both algorithms explicitly rely on the ambient feature dimension $d$, so their direct extensions to the neural bandit setting induce poor regret performance. On the other hand, Verma et al. (2025) derived a regret upper bound for the neural logistic bandit that scales with a data-adaptive effective dimension $\widetilde{d}$ rather than the full ambient dimension $d$. This approach offers an improved performance measure as $d$ increases with the number of parameters in the neural network, often deliberately overparameterized to avoid strong assumptions about the reward function. However, their method still relies on a pessimistic variance estimate, and integrating the variance-aware analysis of Faury et al. (2020) into a data-adaptive regret framework remains challenging, resulting in a suboptimal dependence on $\kappa$.

Table 1: Comparison of algorithms for (neural) logistic bandits. $d$ denotes the dimension of the feature vector, and $T$ represents the total number of rounds. $p$ denotes the total number of parameters of the underlying neural network, and $\widetilde{d}$ denotes the effective dimension.

| Algorithm | Regret $\widetilde{\mathcal{O}}(\cdot)$ | |
| --- | --- | --- |
| | Logistic Bandits | Neural Logistic Bandits |
| NCBF-UCB Verma et al. (2025) | $\kappa d\sqrt{T}$ | $\kappa\widetilde{d}\sqrt{T}$ |
| Logistic-UCB-1 Faury et al. (2020) | $d\sqrt{\kappa T}$ | $p\sqrt{\kappa T}$ |
| **NeuralLog-UCB-1 (Algorithm 1)** | $d\sqrt{\kappa T}$ | $\widetilde{d}\sqrt{\kappa T}$ |
| ada-OFU-ECOLog Faury et al. (2022) | $d\sqrt{T/\kappa^*}$ | $p\sqrt{T/\kappa^*}$ |
| **NeuralLog-UCB-2 (Algorithm 2)** | $d\sqrt{T/\kappa^*}$ | $\widetilde{d}\sqrt{T/\kappa^*}$ |

Motivated by these limitations, we propose algorithms that do not require worst-case estimates in both the variance of the reward distribution and the feature dimension, thus achieving the most favorable regret bound for neural logistic bandits. Central to this approach is our new Bernstein-type self-normalized inequality for vector-valued martingales, which allows us to derive a regret upper bound that scales with the effective dimension $\widetilde{d}$, and at the same time, matches the best-known dependency on $\kappa$. Our main contributions are summarized below:

- We tackle the two main challenges in *neural logistic bandits*: (i) a practical regret upper bound should avoid a direct dependence on $d$, the ambient dimension of the feature vector, and (ii) it needs to minimize the factor of $\kappa$, a problem-dependent constant that increases exponentially with the size of the decision set. To address these challenges, we propose a new Bernstein-type tail inequality for self-normalized vector-valued martingales that yields a bound of order $\widetilde{\mathcal{O}}(\sqrt{\widetilde{d}})$, where $\widetilde{d}$ is a data-adaptive effective dimension. This is the first tail inequality that achieves favorable results in both respects, while the previous bound of Faury et al. (2020) is $\widetilde{\mathcal{O}}(\sqrt{d})$ which directly depends on $d$, and that of Verma et al. (2025) is $\widetilde{\mathcal{O}}(\sqrt{\kappa\widetilde{d}})$ that includes an additional factor of $\sqrt{\kappa}$.

- Based on our tail inequality, we develop our first algorithm, NeuralLog-UCB-1 which guarantees a regret upper bound of order $\widetilde{O}(\widetilde{d}\sqrt{\kappa T})$. This improves upon the regret upper bound of order $\widetilde{\mathcal{O}}(\kappa\widetilde{d}\sqrt{T})$ due to Verma et al. (2025). Furthermore, we provide a fully data-adaptive UCB on $\widetilde{d}$ by adaptively controlling the regularization term of our loss function according to previous observations. Our choice of UCB also avoids the projection step required in the previous approach of Faury et al. (2020), which was to constrain the parameters to a certain set during training.

- We propose our second algorithm, NeuralLog-UCB-2, as a refined variant of NeuralLog-UCB-1. We show that NeuralLog-UCB-2 achieves a regret upper bound of $\widetilde{O}(\widetilde{d}\sqrt{T/\kappa^*})$. This result matches the best-known dependency on $\kappa$ while avoiding the direct dependence on $d$ seen in $\widetilde{O}(d\sqrt{T/\kappa^*})$ given by Abeille et al. (2021). The improvement comes from the fact that NeuralLog-UCB-2 replaces the true reward variance within the design matrix with a neural network estimated variance, thereby maintaining sufficient statistics for our variance-adaptive UCB in each round and completely removing the worst-case estimate of variance $\kappa$. Our numerical results show that NeuralLog-UCB-2 outperforms all baselines, thus validating our theoretical framework.

## 2 Preliminaries

**Logistic bandits.** We consider the contextual logistic bandit problem. Let $T$ be the total number of rounds. In each round $t \in [T]$, the agent observes an action set $\mathcal{X}_t$, consisting of $K$ contexts drawn from a feasible set $\mathcal{X} \subset \mathbb{R}^d$. The agent then selects an action $x_t \in \mathcal{X}_t$ and observes a binary (random) reward $r_t \in \{0, 1\}$. This reward is generated by the logistic model governed by the unknown latent reward function $h : \mathbb{R}^d \to \mathbb{R}$. Specifically, we define a sigmoid function $\mu(x) = (1 + \exp(-x))^{-1}$ and denote its first and second derivatives as $\dot{\mu}$ and $\ddot{\mu}$. Then, the probability distribution of the

reward $r_t$ under action $x$ is given by $r_t \sim \text{Bern}(\mu(h(x_t)))$. Let $x_t^*$ be an optimal action in round $t$, i.e., $x_t^* = \arg\max_{x \in \mathcal{X}_t} \mu(h(x))$. Then the agent's goal is to minimize the cumulative regret, defined as $\text{Regret}(T) = \sum_{t=1}^T \mu(h(x_t^*)) - \sum_{t=1}^T \mu(h(x_t))$. Finally, we introduce the standard assumption on the problem-dependent parameters $\kappa$ and $R$ Faury et al. (2020); Verma et al. (2025):

**Assumption 2.1** (Informal). *There exist constants $\kappa, R > 0$, such that $1/\kappa \leq \dot\mu(\cdot) \leq R$.*

The formal definition of $\kappa$ and $R$ for the arm set $\mathcal{X}$ and the parameter set $\Theta$ is deferred to Assumption 6.3. Notice that for the sigmoid link function, we have $\mu(\cdot), R \leq 1/4$.

**Neural bandits.** Neural contextual bandit methods address the limitation of traditional (generalized) linear reward models Filippi et al. (2010); Faury et al. (2020) by approximating $h(\cdot)$ with a fully connected deep neural network $f(x; \theta)$, which allows them to capture complex, possibly nonlinear, reward structures. In this work, we consider a neural network given by $f(x; \theta) = \sqrt{m} W_L \text{ReLU}(W_{L-1} \text{ReLU}(\cdots \text{ReLU}(W_1 x)))$, where $L \geq 2$ is the depth of the neural network, $\text{ReLU}(x) = \max\{x, 0\}$, $W_1 \in \mathbb{R}^{m \times d}$, $W_i \in \mathbb{R}^{m \times m}$ for $2 \leq i \leq L-1$, and $W_L \in \mathbb{R}^{1 \times m}$. The flattened parameter vector is given by $\theta = [\text{vec}(W_1)^\top, \dots, \text{vec}(W_L)^\top]^\top \in \mathbb{R}^p$, where $p$ is the total number of parameters, i.e., $p = m + md + m^2(L-1)$. We denote the gradient of the neural network by $g(x; \theta) = \nabla_\theta f(x; \theta) \in \mathbb{R}^p$.

**Notation.** For a positive integer $n$, let $[n] = \{1, \dots, n\}$. For any $x \in \mathbb{R}^d$, $\|x\|_2$ denotes the $\ell_2$ norm, and $[x]_i$ denotes its $i$-th coordinate. Given $x \in \mathbb{R}^d$ and a positive-definite matrix $A \in \mathbb{R}^{d \times d}$, we define $\|x\|_A = \sqrt{x^\top A x}$. We use $\widetilde{O}(\cdot)$ to hide the logarithmic factors.

# 3 VARIANCE- AND DATA-ADAPTIVE SELF-NORMALIZED MARTINGALE TAIL INEQUALITY

In this section, we first introduce our new Bernstein-type tail inequality for self-normalized martingales, which leads to a regret analysis that is variance- and data-adaptive. Then we compare it with some existing tail inequalities from prior works.

**Theorem 3.1.** *Let $\{\mathcal{G}_t\}_{t=1}^\infty$ be a filtration, and $\{x_t, \eta_t\}_{t \geq 1}$ be a stochastic process where $x_t \in \mathbb{R}^d$ is $\mathcal{G}_t$-measurable and $\eta_t \in \mathbb{R}$ is $\mathcal{G}_{t+1}$-measurable. Suppose there exist constants $M, R, N, \lambda > 0$ and the parameter $\theta^* \in \mathbb{R}^d$ such that for all $t \geq 1$, $|\eta_t| \leq M$, $\mathbb{E}[\eta_t | \mathcal{G}_t] = 0$, $\mathbb{E}[\eta_t^2 | \mathcal{G}_t] \leq \dot\mu(x_t^\top \theta^*)$, and $\|x_t\|_2 \leq N$. Define $H_t$ and $s_t$ as follows:*

$$H_t = \sum_{i=1}^t \dot\mu(x_i^\top \theta^*) x_i x_i^\top + \lambda \mathbf{I}, \qquad s_t = \sum_{i=1}^t x_i \eta_i.$$

*Then, for any $0 < \delta < 1$ and any $t > 0$, with probability at least $1 - \delta$:*

$$\|s_t\|_{H_t^{-1}} \leq 8 \sqrt{\log \frac{\det H_t}{\det \lambda \mathbf{I}} \log(4t^2/\delta)} + \frac{4MN}{\sqrt{\lambda}} \log(4t^2/\delta)$$

$$\leq 8 \sqrt{\log \det \left( \sum_{i=1}^t \frac{R}{\lambda} x_i x_i^\top + \mathbf{I} \right) \log(4t^2/\delta)} + \frac{4MN}{\sqrt{\lambda}} \log(4t^2/\delta).$$

Our proof of Theorem 3.1 is given in Section D. The second inequality in Theorem 3.1 follows from Assumption 2.1 which states that $\dot\mu(\cdot) \leq R \leq 1/4$. Notice that the tail inequality is data-adaptive, as it does not explicitly depend on $d$. Moreover, the term $\log \frac{\det H_t}{\det \lambda \mathbf{I}}$ can decrease depending on the observed feature vectors (e.g., it becomes 0 if $\{x_i\}_{i=1}^t$ are all $\mathbf{0}$). By incorporating non-uniform variances when defining $H_t$, our design matrix enables a variance-adaptive analysis and eliminates the worst-case variance dependency $\kappa$.

The seminal work of Abbasi-Yadkori et al. (2011) provided a variant of the Azuma-Hoeffding tail inequality for vector-valued martingales, under the assumption that the martingale difference $\eta_t$ is $M$-sub-Gaussian. Their tail bound shows that $\|s_t\|_{\widetilde{V}_t^{-1}} = \widetilde{\mathcal{O}}(M\sqrt{d})$, where $\widetilde{V}_t = \sum_{i=1}^t x_i x_i^\top + \lambda \mathbf{I}$. Extending this result to (neural) logistic bandits, Verma et al. (2025) incorporated the worst-case

variance $\kappa$ into the design matrix $V_t = \sum_{i=1}^{t} x_i x_i^\top + \kappa \lambda I$ to deduce

$$\|s_t\|_{H_t^{-1}} \leq \sqrt{\kappa} \|s_t\|_{V_t^{-1}} \leq M \sqrt{\kappa \log \frac{\det V_t}{\det \kappa \lambda \mathbf{I}} + 2\kappa \log(1/\delta)}.$$

Here, the first inequality is a consequence of $H_t \succeq (1/\kappa) V_t$, and this step incurs the factor $\sqrt{\kappa}$. Note that this bound is also data-adaptive, yielding an overall order of $\widetilde{\mathcal{O}}(\sqrt{\kappa \tilde{d}})$.

Another line of work by Faury et al. (2020) provided a Bernstein-type tail inequality for the same setting considered in Theorem 3.1, using $|\eta_t| \leq M(=1)$, $\mathbb{E}[\eta_t^2 | \mathcal{G}_t] = \sigma_t^2$, and $\|x_t\|_2 \leq N(=1)$ for all $t \geq 1$. Their analysis directly takes the design matrix $H_t$, and they deduce the following inequality avoiding the $\sqrt{\kappa}$ factor:

$$\|s_t\|_{H_t^{-1}} \leq \frac{2MN}{\sqrt{\lambda}} \Big( \log \frac{\det H_t}{\det \lambda \mathbf{I}} + \log(1/\delta) + d \log(2) \Big) + \frac{\sqrt{\lambda}}{2MN}. \tag{1}$$

The inequality requires a specific $\lambda$ value for the regularization term, given by $\lambda = \widetilde{\mathcal{O}}(dM^2N^2)$, to achieve the final order of $\widetilde{\mathcal{O}}(\sqrt{d})$. Although $\log \frac{\det H_t}{\det \lambda \mathbf{I}}$ is data-adaptive, the term $d \log(2)$ introduces an explicit dependence on $d$ that cannot be removed (even with a different choice of $\lambda$). The tail bound has been used in subsequent works Abeille et al. (2021); Faury et al. (2022), making a dependence on $d$ inherent. Hence, we need a new variance-adaptive analysis for neural logistic bandits.

Compared with Faury et al. (2020), our tail inequality in Theorem 3.1 is derived from a different technique based on Freedman's inequality (Freedman (1975), Lemma H.1), which is the key factor behind our improvement. Unlike Faury et al. (2020), which works with a $d$-dimensional martingale and thereby incurs an explicit dependence on $d$, we instead use a one-dimensional martingale to track the growth of the self-normalized error $\|s_t\|_{H_t^{-1}}$, bypassing this vector-level issue. As a result, we obtain a data- and variance-adaptive inequality whose leading term depends on the effective dimension $\tilde{d}$, together with an improved dependence on $\kappa$, thanks to the variance sensitivity of Freedman's inequality.

## 4 NEURAL LOGISTIC BANDITS WITH IMPROVED UCB

This section introduces our first algorithm, NeuralLog-UCB-1, described in Algorithm 1. In the initialization step, we set the initial parameter $\theta_0$ of the neural network according to the standard initialization process described in Zhou et al. (2020). For $1 \leq l \leq L-1$, $W_l$ is set as $\begin{bmatrix} W & 0 \\ 0 & W \end{bmatrix}$, where each entry of $W$ is independently sampled from $N(0, 4/m)$ while $W_L$ is set to $[w, -w]$, where each entry of $w$ is independently sampled from $N(0, 2/m)$. Next, we set the initial regularization parameter as $\lambda_0 = 8\sqrt{2} C_1 L^{1/2} S^{-1} \log(4/\delta)$ for some absolute constant $C_1 > 0$. The value of $\lambda_0$ is chosen so that $\lambda_0$ is less than the minimum value among $\lambda_1, \ldots, \lambda_T$, where $\lambda_t$ is updated as in Equation (4). We can verify this by showing that $\lambda_0 \leq \min \lambda_1$ and that $\{\lambda_t\}_{t \geq 1}$ is monotonically non-decreasing in $t$, which implies $\lambda_0 \leq \min\{\lambda_t\}_{t \geq 1}$.

After the initialization step, in each round $t$, the agent receives the context set $\mathcal{X}_t \subset X$ and calculates $\text{UCB}_t(x) = \mu(f(x; \theta_{t-1})) + R\sqrt{\kappa} \nu_{t-1}^{(1)} \|g(x; \theta_0)/\sqrt{m}\|_{V_{t-1}^{-1}}$ for every action $x \in \mathcal{X}_t$, where

$$\nu_t^{(1)} = C_6 \big( 1 + \sqrt{L}S + LS^2 \big) \iota_t + 1, \tag{2}$$

$$\iota_t = 16 \sqrt{\log \det \Big( \sum_{i=1}^{t} \frac{1}{4m\lambda_0} g(x_i; \theta_0) g(x_i; \theta_0)^\top + \mathbf{I} \Big) \log \frac{4t^2}{\delta}} + 8C_1 \sqrt{\frac{L}{\lambda_0}} \log \frac{4t^2}{\delta}, \tag{3}$$

where $\delta \in (0, 1)$ is a confidence parameter, and $S$ is a norm parameter of the parameter set defined in Definition 6.2 for some absolute constants $C_1, C_6 > 0$. The first term, $\mu(f(x; \theta_{t-1}))$, estimates the expected value of the reward, and the second term can be viewed as the exploration bonus. Then, we choose our action $x_t$ optimistically by maximizing the UCB value, i.e., $x_t = \arg\max_{x \in \mathcal{X}_t} \text{UCB}_t(x)$, and receive a reward $r_t$. At the end of each round, we update the parameters based on the observations $\{x_i, r_i\}_{i=1}^{t}$ collected so far. We set the regularization parameter

---

**Algorithm 1** NeuralLog-UCB-1

---

**Input:** Neural network $f(x; \theta)$ with width $m$ and depth $L$, initialized with parameter $\theta_0$, step size $\eta$, number of gradient descent steps $L$, norm parameter $S$, confidence parameter $\delta$

**Initialize:** $\lambda_0 = 8\sqrt{2}C_1 L^{1/2} S^{-1} \log(4/\delta)$, $V_0 = \kappa\lambda_0 \mathbf{I}$

1: **for** $t = 1, \ldots, T$ **do**
2:      $x_t \leftarrow \arg\max_{x \in \mathcal{X}_t} \mu(f(x; \theta_{t-1})) + R\sqrt{\kappa}\nu_{t-1}^{(1)} \|g(x; \theta_0)/\sqrt{m}\|_{V_{t-1}^{-1}}$
3:      Select $x_t$ and receive $r_t$
4:      Update $\lambda_t$ as in Equation (4), $\iota_t$ as in Equation (3), $\nu_t^{(1)}$ as in Equation (2)
5:      $\theta_t \leftarrow \mathtt{TrainNN}(\lambda_t, \eta, J, m, \{x_i, r_i\}_{i=1}^t, \theta_0)$
6:      $V_t \leftarrow \sum_{i=1}^t \frac{1}{m} g(x_i; \theta_0) g(x_i; \theta_0)^\top + \kappa\lambda_t \mathbf{I}$
7: **end for**

---

**Subroutine** `TrainNN`

---

**Input:** Regularization parameter $\lambda_t$, step size $\eta$, number of gradient descent steps $J$, network width $m$, observations $\{x_s, r_s\}_{s=1}^t$, initial parameter $\theta_0$

1: Define $\mathcal{L}_t(\theta) = -\sum_{i=1}^t [r_i \log \mu(f(x_i; \theta)) + (1 - r_i) \log(1 - \mu(f(x_i; \theta)))] + \frac{1}{2}m\lambda_t \|\theta - \theta_0\|_2^2$
2: **for** $j = 1, \ldots, J - 1$ **do**
3:      $\theta^{(j+1)} = \theta^{(j)} - \eta\nabla\mathcal{L}_t(\theta^{(j)})$
4: **end for**
5: **return** $\theta^{(J)}$

---

$\lambda_t$, with an absolute constant $C_1 > 0$, as follows:

$$\lambda_t \leftarrow \frac{64}{S^2} \log \det \left( \sum_{i=1}^t \frac{1}{4m\lambda_0} g(x_i; \theta_0) g(x_i; \theta_0)^\top + \mathbf{I} \right) \log \frac{4t^2}{\delta} + \frac{16C_1^2 L}{S^2 \lambda_0} \log^2 \frac{4t^2}{\delta}. \quad (4)$$

Then we update $\iota_t$, $\nu_t^{(1)}$, and $V_t$. Lastly, as described in Subroutine `TrainNN`, we update the parameters of the neural network through gradient descent to minimize the regularized negative log-likelihood loss function $\mathcal{L}_t(\theta)$ and obtain $\theta_t$. In contrast to Verma et al. (2025), which used a constant regularization parameter $\lambda$, we adaptively control the regularization parameter $\lambda_t$ and employ it in both our design matrix $V_t$ and our loss function $\mathcal{L}_t(\theta)$. This yields a fully data-adaptive concentration inequality between $\theta_t$ and the desired parameter, as will be shown in Lemma 4.5.

Now, we present our theoretical results for Algorithm 1. Let $\mathbf{H}$ denote the neural tangent kernel (NTK) matrix computed on all $TK$ context–arm feature vectors over $T$ rounds. Its formal definition is deferred to Definition C.1. Define $\mathbf{h} = [h(x)]_{x \in \mathcal{X}_t, t \in [T]} \in \mathbb{R}^{TK}$. We begin with the following assumptions.

**Assumption 4.1.** *There exists $\lambda_{\mathbf{H}} > 0$ such that $\mathbf{H} \succeq \lambda_{\mathbf{H}} \mathbf{I}$.*

**Assumption 4.2.** *For every $x \in \mathcal{X}_t$ and $t \in [T]$, we have $\|x\|_2 = 1$ and $[x]_j = [x]_{j+d/2}$.*

Both assumptions are mild and standard in the neural bandit literature Zhou et al. (2020); Zhang et al. (2021). Assumption 4.1 states that the NTK matrix is nonsingular, which holds if no two context vectors are parallel. Assumption 4.2 ensures that $f(x^i; \theta_0) = 0$ for all $i \in [TK]$ at initialization. This assumption is made for analytical convenience and can be ensured by building a new context $x' = [x^\top, x^\top]^\top / \sqrt{2}$.

Next, define $\widetilde{\mathbf{H}} = \sum_{t=1}^T \sum_{x \in \mathcal{X}_t} \frac{1}{m} g(x_i; \theta_0) g(x_i; \theta_0)^\top$, which is the design matrix containing all possible context-arm feature vectors over the $T$ rounds. Then, we can use the following definition:

**Definition 4.3.** *Let $\widetilde{d} := \log \det(\frac{R}{\lambda_0}\widetilde{\mathbf{H}} + \mathbf{I})$ denote the effective dimension.*

We mention that previous works Zhou et al. (2020); Verma et al. (2025) for neural contextual bandits have defined the effective dimension $\widetilde{d}$ in slightly different ways. Zhou et al. (2020) set $\widetilde{d} = \log \det(\frac{1}{\lambda}\mathbf{H} + \mathbf{I}) / \log(1 + TK/\lambda)$ for neural contextual linear bandits, while Verma et al. (2025) defined $\widetilde{d} = \log \det(\frac{1}{\kappa\lambda}\widetilde{\mathbf{H}} + \mathbf{I})$. However, these definitions have the same asymptotic order as ours in Definition 4.3 up to logarithmic factors.

---

**Algorithm 2** NeuralLog-UCB-2

---

**Input:** Neural network $f(x; \theta)$ with width $m$ and depth $L$, initialized with parameter $\theta_0$, step size $\eta$, number of gradient descent steps $L$, norm parameter $S$, confidence parameter $\delta$

**Initialize:** $\lambda_0 = 8\sqrt{2}C_1 L^{1/2} S^{-1} \log(4/\delta)$, $W_0 = \lambda_0 \mathbf{I}$

1: **for** $t = 1, \dots, T$ **do**
2: $\quad x_t \leftarrow \arg\max_{x \in \mathcal{X}_t} g(x; \theta_0)^\top (\theta_{t-1} - \theta_0) + \nu_{t-1}^{(2)} \|g(x; \theta_0)/\sqrt{m}\|_{W_{t-1}^{-1}}$
3: $\quad$ Select $x_t$ and receive $r_t$
4: $\quad$ Update $\lambda_t$ as in Equation (4), $\iota_t$ as in Equation (3), $\nu_t^{(2)}$ as in Equation (5)
5: $\quad \theta_t \leftarrow \texttt{TrainNN}(\lambda_t, \eta, J, m, \{x_i, r_i\}_{i=1}^t, \theta_0)$
6: $\quad W_t \leftarrow \sum_{i=1}^t \frac{\dot{\mu}(f(x_i; \theta_i))}{m} g(x_i; \theta_0) g(x_i; \theta_0)^\top + \lambda_t \mathbf{I}$
7: **end for**

---

Next, to improve readability, we summarize the conditions for the upcoming theorems and lemmas:

**Condition 4.4.** *Suppose Assumptions 2.1, 4.1 and 4.2 hold (a formal definition of Assumption 2.1 is deferred to Assumption 6.3). The width $m$ is large enough to control the estimation error of the NN (details are deferred to Condition C.2). Set $S$ as a norm parameter satisfying $S \geq \sqrt{2\mathbf{h}^\top \mathbf{H}^{-1} \mathbf{h}}$. The regularization parameter $\lambda_t$ follows the update rule in Equation (4). For training the NN, set the number of gradient descent iterations as $J = 2\log(\lambda_t S/(\sqrt{T}\lambda_t + C_4 T^{3/2} L))TL/\lambda_t$, and the step size as $\eta = C_5(mTL + m\lambda_t)^{-1}$ for some absolute constants $C_4, C_5 > 0$.*

In particular, when $m$ is sufficiently large, we observe that the true reward function $h(x)$ behaves like a linear function (see Lemma 6.1). Then, using the tail inequality given in Theorem 3.1 and the update rule for $\lambda_t$, we obtain the following data- and variance-adaptive concentration inequality between $\theta_t$ and $\theta^*$:

**Lemma 4.5.** *Define $H_t := \sum_{i=1}^t \frac{\dot{\mu}(g(x_i; \theta_0)^\top (\theta - \theta_0))}{m} g(x_i; \theta_0) g(x_i; \theta_0)^\top + \lambda_t \mathbf{I}$. Under Condition 4.4, there exists an absolute constant $C_1, C_6 > 0$, such that for all $t > 0$ with probability at least $1 - \delta$, $\sqrt{m}\|\theta^* - \theta_t\|_{H_t(\theta^*)} \leq \nu_t^{(1)}$, where $\nu_t^{(1)}$ is defined in Equation (2).*

The proof is deferred to Section E.1. We now present Theorem 4.6, which gives the desired regret upper bound of Algorithm 1.

**Theorem 4.6.** *Under Condition 4.4, with probability at least $1 - \delta$, the regret of Algorithm 1 satisfies*

$$Regret(T) = \widetilde{\mathcal{O}}\left(S^2 \widetilde{d}\sqrt{\kappa T} + S^{2.5}\sqrt{\kappa \widetilde{d}T}\right).$$

**Remark 1.** *Our results, especially Theorem 3.1, extend naturally to the (neural) dueling bandit setting. In this variant, the learner selects a pair of context–arms $\{x_{t,1}, x_{t,2}\}$ in each round $t$ and observes a binary outcome $r_t \in \{0, 1\}$ indicating whether $x_{t,1}$ is preferred over $x_{t,2}$. The preference probability is modeled as $\mathbb{P}(x_{t,1} \succ x_{t,2}) = \mathbb{P}(r_t = 1 | x_{t,1}, x_{t,2}) = \mu(h(x_{t,1}) - h(x_{t,2}))$. The prior work of (Verma et al., 2025, Theorem 3) establishes a regret upper bound of $\widetilde{\mathcal{O}}(\kappa \widetilde{d}\sqrt{T})$, whereas our analysis can improve this to $\widetilde{\mathcal{O}}(\widetilde{d}\sqrt{\kappa T})$.*

## 5 REFINED ALGORITHM WITH NEURAL NETWORK-ESTIMATED VARIANCE

In this section, we explain NeuralLog-UCB-2, which guarantees the tightest regret upper bounds. Although Lemma 4.5 establishes a variance-adaptive concentration inequality with $H_t(\theta^*)$, the agent lacks full knowledge of $\theta^*$ and must therefore use the crude bound $H_t(\theta^*) \preceq \kappa^{-1} V_t$, which incurs an extra factor of $\sqrt{\kappa}$. In this section, we introduce NeuralLog-UCB-2, which replaces $H_t(\theta^*)$ with a neural network–estimated variance-adaptive design matrix $W_t$. We begin by stating a concentration result for $\theta^*$ around $\theta_t$ using the new design matrix $W_t$.

**Lemma 5.1.** *Define $W_t = \sum_{i=1}^t \frac{\dot{\mu}(f(x_i; \theta_i))}{m} g(x_i; \theta_0) g(x_i; \theta_0)^\top + \lambda_t \mathbf{I}$ and a confidence set $\mathcal{W}_t$ as*

$$\mathcal{W}_t = \left\{\theta : \sqrt{m}\|\theta - \theta_t\|_{W_t} \leq C_7(1 + \sqrt{L}S + LS^2)\iota_t + 1 =: \nu_t^{(2)}\right\}, \tag{5}$$

*with an absolute constant $C_7 > 0$, where $\iota_t$ is defined at Equation (3). Then under Condition 4.4, for all $t > 0$, $\theta^* \in \mathcal{W}_t$ with probability at least $1 - \delta$.*

We give the proof of the lemma in Section F.1. The matrix $W_t$ maintains sufficient statistics via the neural network-estimated variance, and the ellipsoidal confidence set $\mathcal{W}_t$ changes the original problem into a closed-form optimistic formulation. Specifically, after the same initialization step as for Algorithm 1, the agent selects action $x_t$ in each round $t$ according to the following rule:

$$x_t \leftarrow \underset{x \in \mathcal{X}_t, \theta \in \mathcal{W}_{t-1}}{\arg\max} \ \langle g(x;\theta_0), (\theta - \theta_0) \rangle = \underset{x \in \mathcal{X}_t}{\arg\max} \ g(x;\theta_0)^\top (\theta_{t-1} - \theta_0) + \nu_{t-1}^{(2)} \|x\|_{W_{t-1}^{-1}}. \quad (6)$$

For the regret upper bound of Algorithm 2, we define another problem-dependent quantity $\kappa^*$ as $1/\kappa^* = \frac{1}{T}\sum_{t=1}^T \dot{\mu}(h(x_t^*))$, consistent with the definition in Abeille et al. (2021). Both $\kappa^*$ and $\kappa$ scale exponentially with $S$. We now state our regret upper bound for NeuralLog-UCB-2 and provide a proof outline.

**Theorem 5.2.** *Under Condition 4.4, with probability at least $1-\delta$, the regret of Algorithm 2 satisfies:*

$$Regret(T) = \widetilde{\mathcal{O}}\Big( S^2 \widetilde{d}\sqrt{T/\kappa^*} + S^{2.5}\widetilde{d}^{0.5}\sqrt{T/\kappa^*} + S^4 \kappa \widetilde{d}^2 + S^{4.5}\kappa\widetilde{d}^{1.5} + S^5 \kappa \widetilde{d} \Big).$$

**Remark 2.** *It is possible to further reduce the regret bound in Theorem 5.2 to $\widetilde{\mathcal{O}}(S\widetilde{d}\sqrt{T/\kappa^*})$ by combining Theorem 3.1 and the logistic bandit analysis of Faury et al. (2022), which achieved $\widetilde{\mathcal{O}}(Sd\sqrt{T/\kappa^*})$. However, this approach requires a projection step for $\theta_t$, incurring an additional $\mathcal{O}(d^2 \log(1/\epsilon))$ computational cost for $\epsilon$-accuracy. A couple of recent works eliminated the dependence on $S$ in the leading term, achieving $\widetilde{\mathcal{O}}(d\sqrt{T/\kappa^*})$. Nonetheless, Sawarni et al. (2024) relied on a nonconvex optimization subroutine, while the PAC-Bayes analysis in Lee et al. (2024a) with a uniform prior does not yield data-adaptive regret.*

# 6 REGRET ANALYSES

This section outlines the regret analysis for Algorithms 1 and 2 and provides proof sketch for Theorems 4.6 and 5.2. Let us start by stating some basic results on the NTK analysis and logistic bandits. The following lemma shows that for all $x \in \mathcal{X}_t$ and $t \in [T]$, the true reward function $h(x)$ can be expressed as a linear function.

**Lemma 6.1** (Lemma 5.1, Zhou et al. (2020)). *If $m \geq C_0 T^4 K^4 L^6 \log(T^2 K^2 L/\delta)/\lambda_{\mathbf{H}}^4$ for some absolute constant $C_0 > 0$, then with probability at least $1 - \delta$, there exists $\theta^* \in \mathbb{R}^p$ such that*

$$h(x) = g(x;\theta_0)^\top (\theta^* - \theta_0), \quad \sqrt{m}\|\theta^* - \theta_0\|_2 \leq \sqrt{2\mathbf{h}^\top \mathbf{H}^{-1}\mathbf{h}} \leq S,$$

*for all $x \in \mathcal{X}_t$, $t \in [T]$.*

Based on Lemma 6.1, we define the parameter set $\Theta$ and the parameters $\kappa$ and $R$, which is consistent with the standard logistic bandits literature Faury et al. (2020):

**Definition 6.2.** *Let $\Theta := \{\theta \in \mathbb{R}^p : \sqrt{m}\|\theta - \theta_0\|_2 \leq S\}$ denote the parameter set.*

**Assumption 6.3** (Formal). *There exist constants $\kappa, R > 0$ such that for all $x \in \mathcal{X}$, $\theta \in \Theta$,*

$$1/\kappa \leq \dot{\mu}(g(x;\theta_0)^\top (\theta - \theta_0)) \leq R.$$

## 6.1 PROOF SKETCH OF THEOREM 4.6

Let $|\mu(h(x)) - \mu(f(x;\theta_{t-1}))|$ denote the *prediction error* of $x$ in round $t$, which is the estimation error between the true reward and our trained neural network. We show that with Lemma 4.5 and large enough $m$, the prediction error is upper bounded as follows:

**Lemma 6.4.** *Under Condition 4.4, for all $x \in \mathcal{X}_t$, $t \in [T]$, with probability at least $1 - \delta$,*

$$|\mu(h(x)) - \mu(f(x;\theta_{t-1}))| \leq R\sqrt{\kappa}\nu_t^{(1)}\|g(x;\theta_0)/\sqrt{m}\|_{V_{t-1}^{-1}} + \epsilon_{3,t-1},$$

*where $\epsilon_{3,t} = C_3 R m^{-1/6}\sqrt{\log m}L^3 t^{2/3}\lambda_0^{-2/3}$ for some absolute constant $C_3 > 0$.*

Based on the results so far, we can upper bound the *per-round regret* in round $t$ as follows:

$$\mu(h(x_t^*)) - \mu(h(x_t)) \leq \mu(f(x_t^*;\theta_{t-1})) + R\sqrt{\kappa}\nu_t^{(1)}\|g(x_t^*;\theta_0)/\sqrt{m}\|_{V_{t-1}^{-1}} + \epsilon_{3,t-1} - \mu(h(x_t))$$

$$\leq \mu(f(x_t;\theta_{t-1})) + R\sqrt{\kappa}\nu_t^{(1)}\|g(x_t;\theta_0)/\sqrt{m}\|_{V_{t-1}^{-1}} + \epsilon_{3,t-1} - \mu(h(x_t))$$

$$\leq 2R\sqrt{\kappa}\nu_t^{(1)}\|g(x_t;\theta_0)/\sqrt{m}\|_{V_{t-1}^{-1}} + 2\epsilon_{3,t-1},$$

where the first and last inequalities follow from Lemma 6.4, and the second inequality holds due to the optimistic rule of Algorithm 1. The cumulative regret can be decomposed as $\text{Regret}(T) = \sum_{t=1}^{T} \mu(h(x_t^*)) - \mu(h(x_t)) \leq 2R\sqrt{\kappa}\nu_T^{(1)}\sqrt{T\sum_{t=1}^{T}\left\|g(x_t;\theta_0)/\sqrt{m}\right\|_{V_{t-1}^{-1}}^2} + 2T\epsilon_{3,T}$, for which we use the Cauchy-Schwarz inequality. We have $\nu_T^{(1)} = \widetilde{O}(\sqrt{\widetilde{d}})$, and using the elliptical potential lemma (Lemma H.2) on $\sum_{t=1}^{T}\left\|g(x_t;\theta_0)/\sqrt{m}\right\|_{V_{t-1}^{-1}}^2$ gives $\widetilde{O}(\widetilde{d})$. Finally, setting $m$ large enough under Condition 4.4, the approximation error term gives $T\epsilon_{3,T} = \mathcal{O}(1)$. See Section G.1 for details.

## 6.2 Proof sketch of Theorem 5.2

Let $(x_t, \widetilde{\theta}_{t-1}) \in \mathcal{X}_t \times \mathcal{W}_{t-1}$ be selected by the optimistic rule at time $t$. The per-round regret can be decomposed with a second-order Taylor expansion as follows:

$$\mu(h(x_t^*)) - \mu(h(x_t)) \leq \mu(g(x_t;\theta_0)^\top(\widetilde{\theta}_{t-1} - \theta_0)) - \mu(g(x_t;\theta_0)^\top(\theta^* - \theta_0))$$
$$\leq \dot{\mu}(h(x_t))g(x_t;\theta_0)^\top(\widetilde{\theta}_{t-1} - \theta^*) + 1 \cdot \left[g(x_t;\theta_0)^\top(\widetilde{\theta}_{t-1} - \theta^*)\right]^2,$$

where the first inequality follows from the optimistic rule of Algorithm 2, and we use $\ddot{\mu}(\cdot) \leq 1$ for the second one. To analyze the first term on the right-hand side of the second inequality, we compare $\dot{\mu}(h(x_t))$ and $\dot{\mu}(f(x_t;\theta_t))$ and rewrite the term as $\sqrt{\dot{\mu}(h(x_t))}\|\sqrt{\dot{\mu}(f(x_t;\theta_t))}g(x_t;\theta_0)\|_{W_{t-1}}\|\widetilde{\theta}_{t-1} - \theta^*\|_{W_{t-1}^{-1}}$. Summing this for $t = 1, \ldots, T$, we apply the elliptical potential lemma (Lemma H.2) and Lemma 5.1. For the second term, since we do not enforce any projection or constraint during training, $\theta_{t-1}$ may stay outside $\Theta$. We show that the number of such rounds is $\widetilde{\mathcal{O}}(\kappa\widetilde{d}^2)$. Applying Assumption 6.3 then yields a crude bound of $\kappa\|g(x_t;\theta_0)\|_{V_{t-1}}^2\|\widetilde{\theta}_{t-1} - \theta^*\|_{W_t}^2$. Based on this, the second term can be bounded from above in a similar way. Details are covered in Section G.2.

## 7 Experiments

In this section, we empirically evaluate the performance of our algorithms. Additional results along with further details are deferred to Section A due to space constraints.

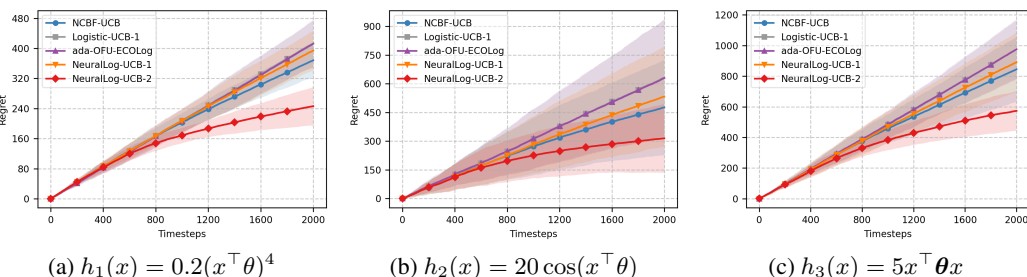

(a) $h_1(x) = 0.2(x^\top\theta)^4$     (b) $h_2(x) = 20\cos(x^\top\theta)$     (c) $h_3(x) = 5x^\top\boldsymbol{\theta}x$

Figure 1: Comparison of cumulative regret of baseline algorithms for nonlinear reward functions.

**Synthetic dataset.** We begin our experiments with a synthetic dataset. We use three nonlinear synthetic latent reward functions: $h_1(x) = 0.2(x^\top\theta)^4$, $h_2(x) = 20\cos(x^\top\theta)$, $h_3(x) = 5x^\top\boldsymbol{\theta}x$, where $x$ represents the features of a context-arm pair, and $\theta \in \mathbb{R}^d$ and $\boldsymbol{\theta} \in \mathbb{R}^{d\times d}$ are parameters whose elements are independently sampled from $\text{Unif}(-1, 1)$. Subsequently, the agent receives a reward generated by $r_t \sim \text{Bern}(\mu(h_i(x)))$, for $i \in \{1, 2, 3\}$. We set the feature vector dimension to $d = 20$ and the number of arms to $K = 5$. We compare our method against five baseline algorithms in Section 1: (1) NCBF-UCB Verma et al. (2025); (2) Logistic-UCB-1 Faury et al. (2020); (3) ada-OFU-ECOLog Faury et al. (2022); (4) NeuralLog-UCB-1; and (5) NeuralLog-UCB-2. For brevity, we will denote algorithms by their number (e.g. algorithm (1)).

Following practical adjustments from previous neural bandits experiments Zhou et al. (2020); Zhang et al. (2021); Verma et al. (2025), for algorithms (1,4,5), we use the gradient of the current neural network $g(x;\theta_t)$ instead of $g(x;\theta_0)$. We replace $g(x;\theta_t)/\sqrt{m}$ with $g(x;\theta_t)$ and

$m\lambda\|\theta - \theta_0\|_2^2/2$ with $\lambda\|\theta\|_2^2$. Previous works simplify the UCB estimation process by fixing parameters for the exploration bonus for practical reasons. In this work, however, we consider the time-varying data-adaptive values of the exploration bonus, characterized by $\text{UCB}_t(x) = \boldsymbol{\mu}(x; \theta_{t-1}) + \boldsymbol{\sigma}(x; \nu, \{x_i, \theta_{i-1}\}_{i=1}^{t-1}, \lambda, S, \kappa)$. Here, $\boldsymbol{\mu}$ is the mean estimate and $\boldsymbol{\sigma}$ is the exploration bonus, parameterized by an exploration parameter $\nu$, previous observations $\{x_i, \theta_{i-1}\}_{i=1}^{t-1}$, $\lambda$, $S$, and $\kappa$. Details of UCB for each algorithm are deferred to Section 5. We use $S = 1$, $\kappa = 10$ and fixed values of $\nu$ and $\lambda$ with the best parameter values using grid search over $\{0.01, 0.1, 1, 10, 100\}$.

We use a two-layer neural network $f(x; \theta)$ with a width of $m = 20$. As in Zhou et al. (2020), to reduce the computational burden of the high-dimensional design matrices $V_t$ and $W_t$, we approximated these matrices with diagonal matrices. We update the parameters every 50 rounds, using 100 gradient descent steps per update with a learning rate of 0.01. For each algorithm, we repeat the experiments 10 times over $T = 2000$ timesteps and compare the average cumulative regret with a 96% confidence interval.

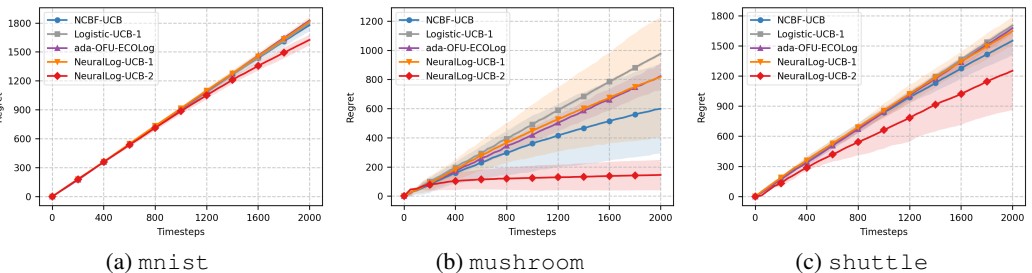

(a) `mnist`    (b) `mushroom`    (c) `shuttle`

Figure 2: Comparison of cumulative regret of baseline algorithms for real-world dataset.

**Real-world dataset.** In the real-world experiments, we use three datasets from $K$-class classification tasks: `mnist` LeCun et al. (1998), `mushroom`, and `shuttle` from the UCI Machine Learning Repository Dua & Graff (2019). To adapt these datasets to the $K$-armed logistic bandit setting, we construct $K$ context–arm feature vectors in each round $t$ as follows: given a feature vector $x \in \mathbb{R}^d$, we define $x^{(1)} = [x, \mathbf{0}, \ldots, \mathbf{0}], \ldots, x^{(K)} = [\mathbf{0}, \ldots, \mathbf{0}, x] \in \mathbb{R}^{dK}$. The agent receives a reward of 1 if it selects the correct class, and 0 otherwise. All other adjustments for the neural bandit experiments and the neural network training process follow the simulation setup. Details, including data preprocessing, are deferred to Section A.

**Regret comparison.** Figures 1 and 2 summarize the average cumulative regret for the five baseline algorithms (1–5) tested with the synthetic and real-world datasets, respectively. We observe that the algorithms using linear assumptions on the latent reward function $h(x)$, namely (2) and (3), exhibit the lowest performance, as the true function is nonlinear. Although algorithm (1) can handle nonlinear reward functions and achieves moderate performance, our proposed methods, especially (5), yield the best results by reducing the dependence on $\kappa$.

## 8 Conclusion and Future Work

In this paper, we study the *neural logistic bandit* problem. We identify the unique challenges of this setting and propose a novel approach based on a new tail inequality for martingales. This inequality enables an analysis that is both variance- and data-adaptive, yielding improved regret bounds for neural logistic bandits. We introduce two algorithms: NeuralLog-UCB-1 that achieves a regret bound of $\widetilde{\mathcal{O}}(\widetilde{d}\sqrt{\kappa T})$ and NeuralLog-UCB-2 that attains a tighter bound of $\widetilde{\mathcal{O}}(\widetilde{d}\sqrt{T/\kappa^*})$ by leveraging the neural network–estimated variance. Our experimental results validate these theoretical findings and demonstrate that our methods outperform the existing approaches.

One potential direction for future work is to improve the dependence on the norm of the unknown parameter $S$. Although recent frameworks due to Sawarni et al. (2024); Lee et al. (2024a) have removed the dependence on $S$ from the leading term, they require an additional training step or impose additional constraints. Such requirements are undesirable when trying to integrate neural bandit frameworks. Hence, it is a promising future research direction to eliminate the dependence on $S$ without additional computations. Additional future directions are deferred to Section **??**.

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

## A  DEFERRED EXPERIMENTS FROM SECTION 7

Here we introduce the deferred details and experiments from section 7. All experiments were conducted on a server equipped with an Intel Xeon Gold 6248R 3.00GHz CPU (32 cores), 512GB of RAM, and 4 GeForce RTX 4090 GPUs.

**Details of UCB.**  We define the UCB value as $\boldsymbol{\mu}(x; \theta_{t-1}) + \boldsymbol{\sigma}(x; \nu, \{x_i, \theta_{i-1}\}_{i=1}^{t-1}, \lambda, S, \kappa)$. For the exploration bonus $\boldsymbol{\sigma}$, we match the orders of $\lambda$, $S$, $\kappa$, and the effective dimension $\widetilde{d}$ for each algorithm and then multiply by the exploration parameter $\nu$. Specifically, the effective dimension is defined as follows: for algorithm (1), we use $\log \det(\sum \frac{1}{\kappa} g(x; \theta) g(x; \theta)^\top + \mathbf{I})$; for algorithms (2) and (3), we use $\log \det(\sum R x x^\top + \mathbf{I})$; and for algorithms (4) and (5) we use $\log \det(\sum R g(x; \theta) g(x; \theta)^\top + \mathbf{I})$.

Although algorithms (2) and (3) require an additional step (e.g., nonconvex projection) to ensure that $\theta_t$ remains in the desired set, empirical observations from Faury et al. (2020; 2022) indicate that $\theta_t$ almost always satisfies this condition. Consequently, we streamline all baseline algorithms into two steps: (i) choose the action with the highest UCB and (ii) update the parameters via gradient descent.

**Preprocessing for real-world datasets.**  For consistency with the synthetic environment, we rescale each component of every feature vector $x \in \mathbb{R}^d$ to the range $[-1, 1]$ by applying a normalization of $2\frac{[x]_j - \min(x)}{\max(x) - \min(x)} - 1$ for all $j \in [d]$. In the mnist dataset, we resize each $28 \times 28$ image to $7 \times 7$, flatten it, and treat the result as a 196-dimensional feature vector. The mushroom dataset provides 22 categorical features. We assign each character a random value in $[-1, 1]$ for normalization and set the label to 1 for edible ('e') and 0 for poisonous ('p') mushrooms. The shuttle dataset consists of 7 numerical features, to which we apply the same min–max normalization as used for mnist.

**Varying effective dimension $\widetilde{d}$.**  To evaluate the influence of $\widetilde{d}$ on data-adaptive algorithms, we compare cumulative regret across different values of $\widetilde{d}$. We control $\widetilde{d}$ by limiting the total number of context–arm feature vectors during training. Allowing redundant vectors reduces $\widetilde{d}$. For a low effective dimension (Figures 3a, 3d and 3g), we use only 10 feature vectors randomly placed across the training. For a medium $\widetilde{d}$ (Figures 3b, 3e and 3h), we use 50 vectors. For a high $\widetilde{d}$ (Figures 3c, 3f and 3i), we use 10000 distinct vectors. Note that Figures 1 and 2 use 2500 vectors. Figure 3 shows that our algorithm, especially algorithm (5), performs best across all those settings and adapts effectively to different environments of the effective dimensions.

## B  RELATED WORK

**Logistic bandits.**  Filippi et al. (2010) introduced the generalized linear bandit framework and derived a regret bound of $\widetilde{\mathcal{O}}(\kappa d \sqrt{T})$, laying the groundwork for modeling logistic bandits. Subsequent work, starting with Faury et al. (2020), has focused on reducing the dependence on $\kappa$ through variance-aware analyses (see also Dong et al. (2019); Abeille et al. (2021)). In particular, Abeille et al. (2021) established a lower bound of $\Omega(d\sqrt{T/\kappa^*})$, statistically closing the gap. However, there is still room for improvement in algorithmic efficiency Faury et al. (2022); Zhang & Sugiyama (2024); Lee & Oh (2024) and in mitigating the influence of the norm parameter $S$, with several recent advances addressing this issue Lee et al. (2024b); Sawarni et al. (2024); Lee et al. (2024a); Lee & Oh (2025). Another line of research investigates the finite-action setting. When feature vectors are drawn i.i.d. from an unknown distribution whose covariance matrix has a strictly positive minimum eigenvalue, Li et al. (2017) achieved a regret of $\widetilde{\mathcal{O}}(\kappa \sqrt{dT})$, while Kim et al. (2023) and Jun et al. (2021) further improved it to $\widetilde{\mathcal{O}}(\sqrt{\kappa dT})$ and $\widetilde{\mathcal{O}}(\sqrt{dT})$, respectively.

**Neural bandits.**  Advances in deep neural networks have spurred numerous methods that integrate deep learning with contextual bandit algorithms Riquelme et al. (2018); Zahavy & Mannor (2019); Kveton et al. (2020). Zhou et al. (2020) was among the first to formalize neural bandits, proposing the *NeuralUCB* algorithm, which attains a regret bound of $\widetilde{\mathcal{O}}(\widetilde{d}\sqrt{T})$ by leveraging neural tangent kernel theory Jacot et al. (2018). Building on this foundation, many studies have extended linear contextual bandit algorithms to the neural setting Zhang et al. (2021); Kassraie & Krause (2022); Ban et al. (2022); Xu et al. (2022); Jia et al. (2022). The work most closely related to ours is Verma

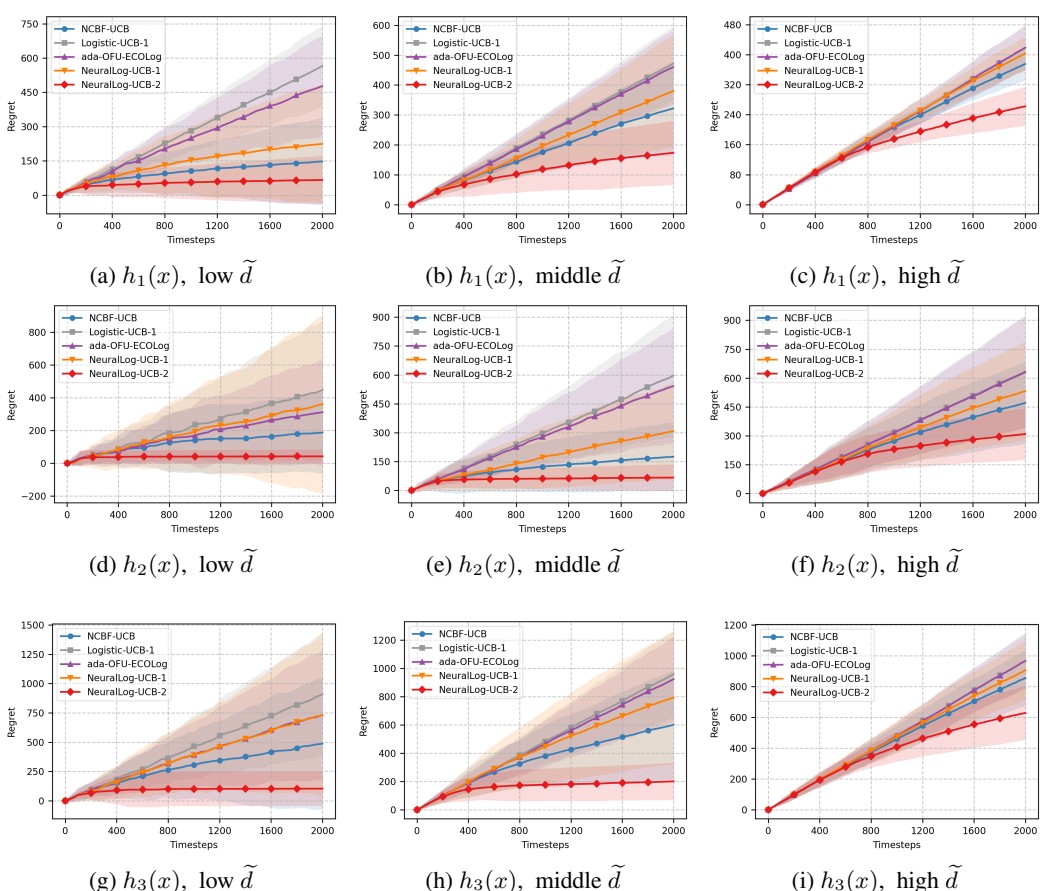

Figure 3: Comparison of cumulative regret of baseline algorithms with varying effective dimension $\widetilde{d}$.

et al. (2025), which first addressed both logistic and dueling neural bandits and proposed UCB- and Thompson-sampling-based algorithms with a regret bound of $\widetilde{\mathcal{O}}(\kappa \widetilde{d}\sqrt{T})$.

## C   USEFUL LEMMAS FOR NEURAL BANDITS

In this section, we present several lemmas that enable the neural bandit analysis to quantify the approximation error incurred when approximating the unknown reward function $h(x)$ with the neural network $f(x;\theta)$. We begin with the definition of the neural tangent kernel (NTK) matrix Jacot et al. (2018):

**Definition C.1.** *Denote all contexts until round $T$ as $\{x^i\}_{i=1}^{TK}$. For $i, j \in [TK]$, define*

$$\widehat{\mathbf{H}}_{i,j}^{(1)} = \mathbf{\Sigma}_{i,j}^{(1)} = \langle x^i, x^j \rangle, \quad \mathbf{A}_{i,j}^{(l)} = \begin{pmatrix} \mathbf{\Sigma}_{i,j}^{(l)} & \mathbf{\Sigma}_{i,j}^{(l)} \\ \mathbf{\Sigma}_{i,j}^{(l)} & \mathbf{\Sigma}_{i,j}^{(l)} \end{pmatrix},$$

$$\mathbf{\Sigma}_{i,j}^{(l+1)} = 2\mathbb{E}_{(u,v)\sim(\mathbf{0},\mathbf{A}_{i,j}^{(l)})} \max\{u, 0\} \max\{v, 0\},$$

$$\widehat{\mathbf{H}}_{i,j}^{(l+1)} = 2\widehat{\mathbf{H}}_{i,j}^{(l)}\mathbb{E}_{(u,v)\sim N(\mathbf{0},\mathbf{A}_{i,j}^{(l)})}\mathbf{1}(u \geq 0)\mathbf{1}(v \geq 0) + \mathbf{\Sigma}_{i,j}^{(l+1)}.$$

*Then, $\mathbf{H} = (\widehat{\mathbf{H}}^{(L)} + \mathbf{\Sigma}^{(L)})/2$ is called the NTK matrix on the context set.*

Next, we introduce a condition on the neural network width $m$, which is crucial for ensuring that the approximation error remains sufficiently small.

**Condition C.2.** *For an absolute constant $C_0 > 0$, the width of the NN $m$ satisfies:*

$$m \geq C_0 \max \left\{ T^4 K^4 L^6 \log(T^2 K^2 L/\delta) \lambda_{\mathbf{H}}^4, L^{-3/2} \lambda_0^{1/2} [\log(TKL^2/\delta)]^{3/2} \right\}$$

$$m(\log m)^{-3} \geq C_0 T^7 L^{21} \lambda_0^{-1} + C_0 T^{16} L^{27} \lambda_0^{-7} R^6 + C_0 T^{10} L^{21} \lambda_0^{-4} R^6 + C_0 T^7 L^{18} \lambda_0^{-4}.$$

We assume that $m$ satisfies Condition C.2 throughout. For readability, we denote the error probability by $\delta$ in all probabilistic statements. We now restate Lemma 6.1, which shows that, for every $x \in \mathcal{X}_t$ and $t \in [T]$, the true reward function $h(x)$ can be represented as a linear function:

**Lemma C.3** (Lemma 5.1, Zhou et al. (2020)). *If $m \geq C_0 T^4 K^4 L^6 \log(T^2 K^2 L/\delta)/\lambda_{\mathbf{H}}^4$ for some absolute constant $C_0 > 0$, then with probability at least $1 - \delta$, there exists $\theta^* \in \mathbb{R}^p$ such that*

$$h(x) = g(x; \theta_0)^\top (\theta^* - \theta_0), \quad \sqrt{m}\|\theta^* - \theta_0\|_2 \leq \sqrt{2\mathbf{h}^\top \mathbf{H}^{-1} \mathbf{h}} \leq S,$$

*for all $x \in \mathcal{X}_t$, $t \in [T]$.*

Assuming $\theta_t$ remains close to its initialization $\theta_0$, we can apply the following lemmas: Lemma C.4 provides upper bounds on the norms $\|g(x;\theta)\|_2$ and $\|g(x;\theta) - g(x;\theta_0)\|_2$, while Lemma C.5 bounds the approximation error between $f(x;\theta)$ and its linearization $g(x;\theta_0)^\top(\theta - \theta_0)$.

**Lemma C.4** (Lemma B.5 and B.6, Zhou et al. (2020)). *Let $\tau = 3\sqrt{\frac{t}{m\lambda_t}}$. Then there exist absolute constants $C_1, C_2 > 0$, such that for all $x \in \mathcal{X}_t$, $t \in [T]$ and for all $\|\theta - \theta_0\|_2 \leq \tau$, with probability of at least $1 - \delta$,*

$$\|g(x;\theta)\|_2 \leq C_1 \sqrt{mL}$$
$$\|g(x;\theta) - g(x;\theta_0)\|_2 \leq C_2 \sqrt{m \log m} \tau^{1/3} L^{7/2} = C_2 m^{1/3} \sqrt{\log m} t^{1/6} \lambda_t^{-1/6} L^{7/2}.$$

**Lemma C.5** (Lemma B.4, Zhou et al. (2020)). *Let $\tau = 3\sqrt{\frac{t}{m\lambda_t}}$. Then there exists an absolute constant $C_3 > 0$, for all $x \in \mathcal{X}_t$, $t \in [T]$, and for all $\|\theta - \theta_0\|_2 \leq \tau$, with probability of at least $1 - \delta$,*

$$\left| f(x;\theta) - g(x;\theta_0)^\top(\theta - \theta_0) \right|_2$$
$$\leq C_3 \tau^{4/3} L^3 \sqrt{m \log m} = C_3 m^{-1/6} \sqrt{\log m} L^3 t^{2/3} \lambda_t^{-2/3}.$$

Finally, we state a lemma that establishes an upper bound on the distance between $\theta_t$ and $\theta_0$. It also shows that, although the loss function $\mathcal{L}_t(\theta)$ is non-convex and hence the iterate $\theta_t$ obtained after $J$ steps of gradient descent may differ from the ideal maximum likelihood estimator, this discrepancy remains sufficiently small. The proof is deferred to Section C.1:

**Lemma C.6.** *Define the auxiliary loss function $\widetilde{L}(\theta)$ as*

$$\widetilde{\mathcal{L}}(\theta) = -\sum_{i=1}^{t} r_i \log \mu(g(x_i; \theta_0)^\top(\theta - \theta_0)) + (1 - r_i) \log(1 - \mu(g(x_i; \theta_0)^\top(\theta - \theta_0)))$$
$$+ \frac{m\lambda_t}{2} \|\theta - \theta_0\|_2^2,$$

*and the auxiliary sequence $\{\widetilde{\theta}^{(j)}\}_{j=1}^{J}$ associated with the auxiliary loss $\widetilde{L}(\theta)$. Let the MLE estimator as $\hat{\theta}_t = \arg\min_\theta \widetilde{\mathcal{L}}(\theta)$. Then there exist absolute constants $\{C_i\}_{i=1}^{5} > 0$ such that if $J = 2\log(\lambda_t S/(\sqrt{T}\lambda_t + C_4 T^{3/2} L)) TL/\lambda_t$ and $\eta = C_5(mTL + m\lambda_t)^{-1}$, then with probability at least $1 - \delta$,*

$$\|\theta_t - \widetilde{\theta}_t\|_2 \leq \sqrt{\frac{t}{m\lambda_t}}, \qquad \|\theta_t - \theta_0\|_2 \leq 3\sqrt{\frac{t}{m\lambda_t}} = \tau,$$
$$\|\theta_t - \hat{\theta}_t\|_2 \leq 2(1 - \eta m\lambda_t)^{J/2} t^{1/2} m^{-1/2} \lambda_t^{-1/2}$$
$$+ C_2 m^{-2/3} \sqrt{\log m} t^{7/6} \lambda_t^{-7/6} L^{7/2} + C_1 C_3 R m^{-2/3} \sqrt{\log m} L^{7/2} t^{5/3} \lambda_t^{-5/3}.$$

## C.1  PROOF OF LEMMA C.6

For simplicity, we omit the subscript $t$ by default. First, recall the definition of the auxiliary sequence $\{\widetilde{\theta}^{(j)}\}_{j=1}^{J}$ associated with the auxiliary loss $\widetilde{L}(\theta)$; its update rule is given by:

$$
\widetilde{\theta}^{(j+1)} = \widetilde{\theta}^{(j)} - \eta \nabla \widetilde{L}(\widetilde{\theta}^{(j)})
$$

$$
= \widetilde{\theta}^{(j)} - \eta \Big[ m\lambda \widetilde{\theta}^{(j)} - \sum_{i=1}^{t} (\mu(g(x_i;\theta_0)^\top (\widetilde{\theta}^{(j)} - \theta_0)) - r_i) g(x_i;\theta_0) \Big].
$$

Similarly, the update rule for $\theta^{(j)}$ is given by:

$$
\theta^{(j+1)} = \theta^{(j)} - \eta \Big[ m\lambda \theta^{(j)} - \sum_{i=1}^{t} (\mu(f(x;\theta^{(j)})) - r_i) g(x_i;\theta^{(j)}) \Big].
$$

Also notice that

$$
\nabla^2 \widetilde{L}(\theta) = \sum_{i=1}^{t} \mu(g(x_i;\theta_0)^\top (\theta - \theta_0))(1 - \mu(g(x_i;\theta_0)^\top (\theta - \theta_0))) g(x_i;\theta_0) g(x_i;\theta_0)^\top + m\lambda \mathbf{I}. \tag{7}
$$

Now, we start the proof with

$$
\|\theta^{(j+1)} - \hat{\theta}\|_2 \leq \underbrace{\|\widetilde{\theta}^{(j+1)} - \hat{\theta}\|_2}_{\textbf{(term 1)}} + \underbrace{\|\theta^{(j+1)} - \widetilde{\theta}^{(j+1)}\|_2}_{\textbf{(term 2)}} \tag{8}
$$

For **(term 1)**, observe from Equation (7) that $\widetilde{\mathcal{L}}$ is $(m\lambda)$-strongly convex, since $(m\lambda)\mathbf{I} \preceq \nabla^2 \widetilde{L}(\theta)$. Moreover, $\widetilde{\mathcal{L}}$ is a $C_5(tmL + m\lambda)$-smooth function for some absolute constant $C_5 > 0$, because

$$
\nabla^2 \widetilde{L}(\theta) \preceq \sum_{i=1}^{t} \frac{1}{2} \cdot \frac{1}{2} \cdot g(x_i;\theta_0) g(x_i;\theta_0)^\top + m\lambda \mathbf{I}
$$

$$
\preceq \Big( \sum_{i=1}^{t} \frac{1}{4} \|g(x_i;\theta_0)\|_2^2 + m\lambda \Big) \mathbf{I}
$$

$$
\preceq C_5(tmL + m\lambda)\mathbf{I},
$$

where the first inequality follows from $\mu(\cdot)(1 - \mu(\cdot)) \leq 1/4$, the second follows because for any $u, x \in \mathbb{R}^d$, $u^\top x x^\top u \leq \|u\|_2^2 \|x\|_2^2 \leq u^\top (\|x\|_2^2 I) u$, and the last inequality follows from Lemma C.4.

Then, with our choice of $\eta = C_5(tmL + m\lambda)$, standard results for gradient descent on $\widetilde{L}$ imply that $\widetilde{\theta}^{(j)}$ converges to $\hat{\theta}_t$ at the rate as

$$
\|\widetilde{\theta}^{(j)} - \hat{\theta}\|_2^2 \leq \frac{2}{m\lambda} \cdot (\widetilde{\mathcal{L}}(\widetilde{\theta}^{(j)}) - \widetilde{\mathcal{L}}(\hat{\theta}))
$$

$$
\leq (1 - \eta m\lambda)^j \cdot \frac{2}{m\lambda} \cdot \big( \widetilde{\mathcal{L}}(\theta_0) - \widetilde{\mathcal{L}}(\hat{\theta}) \big)
$$

$$
\leq (1 - \eta m\lambda)^j \cdot \frac{2}{m\lambda} \cdot \widetilde{\mathcal{L}}(\theta_0),
$$

where the first and the second inequalities follow from the strong convexity and the smoothness of $\widetilde{L}$. Furthermore, we have

$$
\widetilde{\mathcal{L}}(\theta_0) = -\sum_{i=1}^{t} r_i \log \mu(0) + (1 - r_i) \log(1 - \mu(0)) + \frac{m\lambda_t}{2} \|\theta_0 - \theta_0\|_2^2 = -\sum_{i=1}^{t} \log 0.5 \leq t,
$$

Plugging this back gives

$$
\|\widetilde{\theta}^{(j)} - \hat{\theta}\|_2 \leq (1 - \eta m\lambda)^{j/2} \sqrt{2t/(m\lambda)}. \tag{9}
$$

Next, consider **(term 2)**. From the definition of the update rule, it follows that:

$$\textbf{(term 2)} \leq (1 - \eta m\lambda)\|\theta^{(j)} - \widetilde{\theta}^{(j)}\|_2$$

$$+ \eta\Big\|\sum_{i=1}^{t}(\mu(f(x;\theta^{(j)}) - r_i)g(x_i;\theta^{(j)})) - \sum_{i=1}^{t}(\mu(g(x_i;\theta_0)^\top(\widetilde{\theta}^{(j)} - \theta_0)) - r_i)g(x_i;\theta_0))\Big\|_2$$

$$\leq (1 - \eta m\lambda)\|\theta^{(j)} - \widetilde{\theta}^{(j)}\|_2 + \eta\underbrace{\sum_{i=1}^{t}\Big\|(\mu(f(x;\theta^{(j)}) - r_i)[g(x_i;\theta^{(j)}) - g(x_i;\theta_0)])\Big\|_2}_{\textbf{(term 3)}}$$

$$+ \eta\underbrace{\sum_{i=1}^{t}\Big\|[\mu(f(x;\theta^{(j)})) - \mu(g(x;\theta_0)^\top(\widetilde{\theta}^{(j)} - \theta_0))]g(x_i;\theta_0)\Big\|_2}_{\textbf{(term 4)}}. \tag{10}$$

Considering each term of Equation (10), there exist absolute constants $C_1, C_2, C_3 > 0$ such that

$$\textbf{(term 3)} \leq \eta\sum_{i=1}^{t}\Big\|1\cdot[g(x_i;\theta^{(j)}) - g(x_i;\theta_0)]\Big\|_2 \leq C_2\eta m^{1/3}\sqrt{\log m}t^{7/6}\lambda^{-1/6}L^{7/2} \tag{11}$$

$$\textbf{(term 4)} \leq \eta\sum_{i=1}^{t}\Big\|R[f(x;\theta^{(j)}) - g(x;\theta_0)^\top(\widetilde{\theta}^{(j)} - \theta_0)]g(x_i;\theta_0)\Big\|_2$$

$$\leq \eta R\sum_{i=1}^{t}\Big\|f(x;\theta^{(j)}) - g(x;\theta_0)^\top(\widetilde{\theta}^{(j)} - \theta_0)\Big\|_2 \cdot \|g(x_i;\theta_0)\|_2$$

$$\leq C_3\eta R\sum_{i=1}^{t}m^{-1/6}\sqrt{\log m}L^3 t^{2/3}\lambda^{-2/3}\|g(x_i;\theta_0)\|_2$$

$$\leq C_1 C_3\eta R m^{1/3}\sqrt{\log m}L^{7/2}t^{5/3}\lambda^{-2/3}. \tag{12}$$

For **(term 3)**, we apply Lemma C.4. For **(term 4)**, the first inequality follows from the $R$-Lipschitz continuity of $\mu(\cdot)$, the second follows from the Cauchy–Schwarz inequality, the third follows from Lemma C.5, and the final inequality follows from Lemma C.4 after summing over $t$.

Substituting Equations (11) and (12) into Equation (10) yields

$$\|\theta^{(j+1)} - \widetilde{\theta}^{(j+1)}\|_2 \leq (1 - \eta m\lambda)\|\theta^{(j)} - \widetilde{\theta}^{(j)}\|_2$$

$$+ C_2\eta m^{1/3}\sqrt{\log m}t^{7/6}\lambda^{-1/6}L^{7/2} + C_1 C_3\eta R m^{1/3}\sqrt{\log m}L^{7/2}t^{5/3}\lambda^{-2/3} \tag{13}$$

By iteratively applying Equation (13) from 0 to $j$, we obtain

$$\|\theta^{(j+1)} - \widetilde{\theta}^{(j+1)}\|_2 \leq \frac{C_2\eta m^{1/3}\sqrt{\log m}t^{7/6}\lambda^{-1/6}L^{7/2} + C_1 C_3\eta R m^{1/3}\sqrt{\log m}L^{7/2}t^{5/3}\lambda^{-2/3}}{\eta m\lambda}$$

$$\leq C_2 m^{-2/3}\sqrt{\log m}t^{7/6}\lambda^{-7/6}L^{7/2} + C_1 C_3 R m^{-2/3}\sqrt{\log m}L^{7/2}t^{5/3}\lambda^{-5/3}. \tag{14}$$

By substituting Equations (9) and (14) into Equation (8) and setting $j = J - 1$, we complete the proof of the upper bound for $\|\theta_t - \hat{\theta}_2\|_2$. Likewise, from Equation (14), setting $j = J - 1$ and following the width condition in Condition 4.4 yields

$$\|\theta_t - \widetilde{\theta}_t\|_2 \leq \sqrt{\frac{t}{m\lambda_t}}\Big(C_2 m^{-1/6}\sqrt{\log m}t^{2/3}\lambda_t^{-2/3}L^{7/2} + C_1 C_3 R m^{-1/6}\sqrt{\log m}L^{7/2}t^{7/6}\lambda_t^{-7/6}\Big)$$

$$\leq \sqrt{\frac{t}{m\lambda_t}}\Big(C_2 m^{-1/6}\sqrt{\log m}T^{2/3}\lambda_0^{-2/3}L^{7/2} + C_1 C_3 R m^{-1/6}\sqrt{\log m}L^{7/2}T^{7/6}\lambda_0^{-7/6}\Big)$$

$$\leq \sqrt{\frac{t}{m\lambda}}, \tag{15}$$

which completes the bound on $\|\theta_t - \widetilde{\theta}_t\|_2$. Finally, observe that

$$\|\theta_t - \theta_0\|_2 \le \|\theta_t - \widetilde{\theta}_t\|_2 + \|\widetilde{\theta}_t - \theta_0\|_2,$$

where Equation (15) gives $\|\theta_t - \widetilde{\theta}_t\|_2 \le \tau/3$, and for the second term

$$\frac{m\lambda_t}{2}\|\theta_t - \theta_0\|_2^2 \le \widetilde{L}(\widetilde{\theta}_t) \le \widetilde{L}(\theta_0) = \sum_{i=1}^{t} r_i \log \mu(0) + (1 - r_i)\log(1 - \mu(0)) \le t \log 2,$$

which implies $\|\widetilde{\theta}_t - \theta_0\|_2 \le 2\sqrt{t/(m\lambda_t)} = 2\tau/3$. Combining these results completes the proof of the bound on $\|\theta_t - \theta_0\|_2$.

## D  PROOF OF THEOREM 3.1

Our proof technique is primarily inspired by the recent work of Zhou et al. (2021), which integrates non-uniform variance into the analysis of linear bandits. For brevity, let $\sigma_t^2 = \dot{\mu}(x_t^\top \theta^*)$, which yields

$$H_t = \sum_{i=1}^{t} \dot{\mu}(x_i^\top \theta^*)x_i x_i^\top + \lambda \mathbf{I} = \sum_{i=1}^{t} \sigma_i^2 x_i x_i^\top + \lambda \mathbf{I}.$$

We introduce the following additional definitions:

$$\beta_t = 8\sqrt{\log \frac{\det H_t}{\det \lambda I} \log(4t^2/\delta)} + \frac{4MN}{\sqrt{\lambda}}\log(4t^2/\delta)$$

$$s_t = \sum_{i=1}^{t} x_i \eta_i, \ Z_t = \|s_t\|_{H_t^{-1}}, \ w_t = \|x_t\|_{H_{t-1}^{-1}}, \ \mathcal{E}_t = \mathbb{1}\{0 \le s \le t, Z_s \le \beta_s\} \tag{16}$$

for $t \ge 1$, where we set $s_0 = 0, Z_0 = 0, \beta_0 = 0$.

Since $H_t = H_{t-1} + \sigma_t^2 x_t x_t^\top$, by the matrix inversion lemma,

$$H_t^{-1} = H_{t-1}^{-1} - \frac{H_{t-1}^{-1}(\sigma_t x_t)(\sigma_t x_t)^\top H_{t-1}^{-1}}{1 + (\sigma_t x_t)^\top H_{t-1}^{-1}(\sigma_t x_t)}$$

$$= H_{t-1}^{-1} - \frac{\sigma_t^2 H_{t-1}^{-1} x_t x_t^\top H_{t-1}^{-1}}{1 + \sigma_t^2 w_t^2}. \tag{17}$$

We begin by establishing a crude upper bound on $Z_t$. In particular, we have

$$Z_t^2 = \|s_t\|_{H_t^{-1}}^2 = (s_{t-1} + x_t \eta_t)^\top H_t^{-1}(s_{t-1} + x_t \eta_t)$$

$$= s_{t-1}^\top H_t^{-1} s_{t-1} + 2\eta_t x_t^\top H_t^{-1} s_{t-1} + \eta_t^2 x_t^\top H_t^{-1} x_t$$

$$\le Z_{t-1}^2 + \underbrace{2\eta_t x_t^\top H_t^{-1} s_{t-1}}_{\text{(term 1)}} + \underbrace{\eta_t^2 x_t^\top H_t^{-1} x_t}_{\text{(term 2)}},$$

where the inequality follows from the fact that $H_t \succeq H_{t-1}$. For **(term 1)**, from the matrix inversion lemma Equation (17), we have

$$\textbf{(term 1)} = 2\eta_t \left( x_t^\top H_{t-1}^{-1} s_{t-1} - \frac{\sigma_t^2 x_t^\top H_{t-1}^{-1} x_t x_t^\top H_{t-1}^{-1} s_{t-1}}{1 + \sigma_t^2 w_t^2} \right)$$

$$= 2\eta_t \left( x_t^\top H_{t-1}^{-1} s_{t-1} - \frac{\sigma_t^2 w_t^2 x_t^\top H_{t-1}^{-1} s_{t-1}}{1 + \sigma_t^2 w_t^2} \right)$$

$$= \frac{2\eta_t x_t^\top H_{t-1}^{-1} s_{t-1}}{1 + \sigma_t^2 w_t^2}.$$

For **(term 2)**, again from the matrix inversion lemma Equation (17), we have

$$\textbf{(term 2)} = \eta_t^2 \left( x_t^\top H_{t-1}^{-1} x_t - \frac{\sigma_t^2 x_t^\top H_{t-1}^{-1} x_t x_t^\top H_{t-1}^{-1} x_t}{1 + \sigma_t^2 w_t^2} \right) = \eta_t^2 \left( w_t^2 - \frac{\sigma_t^2 w_t^4}{1 + \sigma_t^2 w_t^2} \right) = \frac{\eta_t^2 w_t^2}{1 + \sigma_t^2 w_t^2}. \tag{18}$$

Therefore, we have

$$Z_t^2 \leq Z_{t-1}^2 + \frac{2\eta_t x_t^\top H_{t-1}^{-1} s_{t-1}}{1 + \sigma_t^2 w_t^2} + \frac{\eta_t^2 w_t^2}{1 + \sigma_t^2 w_t^2},$$

and by summing this up from $i = 1$ to $t$ gives,

$$Z_t^2 \leq \sum_{i=1}^{t} \frac{2\eta_i x_i^\top H_{i-1}^{-1} s_{i-1}}{1 + \sigma_i^2 w_i^2} + \sum_{i=1}^{t} \frac{\eta_i^2 w_i^2}{1 + \sigma_i^2 w_i^2}. \tag{19}$$

We give two lemmas to upper bound each term.

**Lemma D.1.** *Let $s_i$, $w_i$, $\mathcal{E}_i$ be as defined in Equation (16). Then, with probability at least $1 - \delta/2$, simultaneously for all $t \geq 1$ it holds that*

$$\sum_{i=1}^{t} \frac{2\eta_i x_i^\top H_{i-1}^{-1} s_{i-1}}{1 + \sigma_i^2 w_i^2} \mathcal{E}_{i-1} \leq \frac{3}{4} {\beta_t}^2.$$

**Lemma D.2.** *Let $w_i$ be as defined in Equation (16). Then, with probability at least $1 - \delta/2$, simultaneously for all $t \geq 1$ it holds that*

$$\sum_{i=1}^{t} \frac{\eta_i^2 w_i^2}{1 + \sigma_i^2 w_i^2} \leq \frac{1}{4} {\beta_t}^2.$$

Now consider the event $\mathcal{E}$ in which the conclusions of Lemma D.1 and Lemma D.2 hold. We claim that, on this event, for any $i \geq 0$, $Z_i \leq \beta_i$. We prove this by induction on $i$. For the base case $i = 0$, the claim holds by definition, since $\beta_0 = 0 = Z_0$. Now fix any $t \geq 1$ and assume that for all $0 \leq i < t$ we have $Z_i \leq \beta_i$. Under this induction hypothesis, it follows that $\mathcal{E}_1 = \mathcal{E}_2 = \cdots = \mathcal{E}_{t-1} = 1$. Then by Equation (19), we have

$$Z_t^2 \leq \sum_{i=1}^{t} \frac{2\eta_i x_i^\top H_{i-1}^{-1} s_{i-1}}{1 + \sigma_i^2 w_i^2} + \sum_{i=1}^{t} \frac{\eta_i^2 w_i^2}{1 + \sigma_i^2 w_i^2} = \sum_{i=1}^{t} \frac{2\eta_i x_i^\top H_{i-1}^{-1} s_{i-1}}{1 + \sigma_i^2 w_i^2} \mathcal{E}_{i-1} + \sum_{i=1}^{t} \frac{\eta_i^2 w_i^2}{1 + \sigma_i^2 w_i^2}. \tag{20}$$

Since on the event $\mathcal{E}$ the conclusion of Lemma D.1 and Lemma D.2 hold, we have

$$\sum_{i=1}^{t} \frac{2\eta_i x_i^\top H_{i-1}^{-1} s_{i-1}}{1 + \sigma_i^2 w_i^2} \mathcal{E}_{i-1} \leq \frac{3}{4} {\beta_t}^2, \quad \sum_{i=1}^{t} \frac{\eta_i^2 w_i^2}{1 + \sigma_i^2 w_i^2} \leq \frac{1}{4} {\beta_t}^2. \tag{21}$$

Therefore, substituting Equation (21) into Equation (20) yields $Z_t \leq \beta_t(\delta)$, which completes the induction. By a union bound, the events in Lemma D.1 and Lemma D.2 both hold with probability at least $1 - \delta$. Hence, with probability at least $1 - \delta$, for all $t$, $Z_t \leq \beta_t$.

### D.1 PROOF OF LEMMA D.1

We now proceed to apply Freedman's inequality, as stated in Lemma H.1. We have

$$\left| \frac{2 x_i^\top H_{i-1}^{-1} s_{i-1}}{1 + \sigma_i^2 w_i^2} \mathcal{E}_{i-1} \right| \leq \frac{2 \|x_i\|_{H_{i-1}^{-1}} \left[ \|s_{i-1}\|_{H_{i-1}^{-1}} \mathcal{E}_{i-1} \right]}{1 + \sigma_i^2 w_i^2} \leq \frac{2 w_i \beta_{i-1}}{1 + \sigma_i^2 w_i^2} \leq \min\{1/\sigma_i, 2w_i\} \beta_{i-1}. \tag{22}$$

Here, the first inequality follows from the Cauchy–Schwarz inequality, the second follows from the definition of $\mathcal{E}_{i-1}$, and the final inequality follows by simple algebra. For simplicity, let $\ell_i$ denote

$$\ell_i = \frac{2\eta_i x_i^\top H_{i-1}^{-1} s_{i-1}}{1 + \sigma_i^2 w_i^2} \mathcal{E}_{i-1}.$$

We now apply Freedman's inequality from Lemma H.1 to the sequences $(\ell_i)_i$ and $(\mathcal{G}_i)_i$. First, note that $\mathbb{E}[\ell_i | \mathcal{G}_i] = 0$. Moreover, by Equation (22), the following inequalities hold almost surely:

$$|\ell_i| \leq M \beta_{i-1} \min\{1/\sigma_i, 2w_i\} \leq \frac{2MN}{\sqrt{\lambda}} \beta_t, \tag{23}$$

where the last inequality follows since $(\beta_i)_i$ is non-decreasing in $i$ and by the fact that

$$w_i = \|x_i\|_{H_{i-1}^{-1}} \leq \|x_i\|_2/\sqrt{\lambda} \leq N/\sqrt{\lambda}. \tag{24}$$

We also have

$$\sum_{i=1}^{t} \mathbb{E}[\ell_i^2|\mathcal{G}_i] \leq \sum_{i=1}^{t} \sigma_i^2 \left( \frac{2x_i^\top H_{i-1}^{-1} s_{i-1}}{1 + \sigma_i^2 w_i^2} \mathcal{E}_{i-1} \right)^2$$

$$\leq \sum_{i=1}^{t} \sigma_i^2 \left( \min\{1/\sigma_i, 2w_i\}\beta_{i-1} \right)^2$$

$$= \sum_{i=1}^{t} \left( \min\{1, 2w_i\sigma_i\}\beta_{i-1} \right)^2$$

$$\leq 4\beta_t^2 \sum_{i=1}^{t} \min\{1, (w_i\sigma_i)^2\},$$

where the first inequality holds by the definition of $\sigma_i$, the second inequality follows from Equation (22), the third inequality holds since $(\beta_i)_i$ is non-decreasing. Since

$$\sum_{i=1}^{t} \min\{1, (w_i\sigma_i)^2\} = \sum_{i=1}^{t} \min\{1, \|\sigma_i x_i\|_{H_{i-1}^{-1}}^2\} \leq 2\log\frac{\det H_t}{\det \lambda \mathbf{I}}, \tag{25}$$

where the last inequality follows from Lemma H.2. Substituting this back yields,

$$\sum_{i=1}^{t} \mathbb{E}[\ell_i^2|\mathcal{G}_i] \leq 8\beta_t^2 \log\det\left( \sum_{i=1}^{t} \frac{\sigma_i^2}{\lambda} x_i x_i^\top + \mathbf{I} \right). \tag{26}$$

Therefore, by Equations (23) and (26), using Lemma H.1, we know that for any $t$, with probability at least $1 - \delta/(4t^2)$, we have

$$\sum_{i=1}^{t} \ell_i \leq \sqrt{16\beta_t^2 \log\det\left( \sum_{i=1}^{t} \frac{\sigma_i^2}{\lambda} x_i x_i^\top + \mathbf{I} \right) \log(4t^2/\delta)} + \frac{2}{3} \cdot \frac{2MN}{\sqrt{\lambda}} \beta_t \log(4t^2/\delta)$$

$$\leq \frac{\beta_t^2}{4} + 16 \log\det\left( \sum_{i=1}^{t} \frac{\sigma_i^2}{\lambda} x_i x_i^\top + \mathbf{I} \right) \log(4t^2/\delta) + \frac{\beta_t^2}{4} + \frac{4M^2N^2}{\lambda} \log^2(4t^2/\delta)$$

$$\leq \frac{\beta_t^2}{2} + \frac{1}{4}\left[ 8\sqrt{\log\det\left( \sum_{i=1}^{t} \frac{\sigma_i^2}{\lambda} x_i x_i^\top + \mathbf{I} \right) \log(4t^2/\delta)} + \frac{4MN}{\sqrt{\lambda}} \log(4t^2/\delta) \right]^2$$

$$= \frac{3}{4}\beta_t^2, \tag{27}$$

where the second inequality follows from the fact that $2\sqrt{|ab|} \leq |a| + |b|$, and the final equality follows from the definition of $\beta_t$. Applying a union bound to Equation (27) from $t = 1$ to $\infty$ and using the fact that $\sum_{t=1}^{\infty} t^{-2} < 2$ completes the proof.

### D.2 PROOF OF LEMMA D.2

Similarly to Lemma D.1, we apply Freedman's inequality from Lemma H.1 to the sequences $(\ell_i)_i$ and $(\mathcal{G}_i)_i$, where now

$$\ell_i = \frac{\eta_i^2 w_i^2}{1 + \sigma_i^2 w_i^2} - \mathbb{E}\left[ \frac{\eta_i^2 w_i^2}{1 + \sigma_i^2 w_i^2} \middle| \mathcal{G}_i \right].$$

First, with Equation (24), we derive a crude upper bound for the following term:

$$\left| \frac{\eta_i^2 w_i^2}{1 + \sigma_i^2 w_i^2} \right| \leq |\eta_i^2 w_i^2| \leq \frac{M^2 N^2}{\lambda}. \tag{28}$$

Now, for any $i$, we have $\mathbb{E}[\ell_i|\mathcal{G}_i] = 0$ almost surely. Furthermore, we can see that

$$\sum_{i=1}^{t} \mathbb{E}[\ell_i^2|\mathcal{G}_i] \leq \sum_{i=1}^{t} \mathbb{E}\left[\frac{\eta_i^4 w_i^4}{(1 + \sigma_i^2 w_i^2)^2}\bigg|\mathcal{G}_i\right]$$

$$\leq \frac{M^2 N^2}{\lambda} \sum_{i=1}^{t} \mathbb{E}\left[\frac{\eta_i^2 w_i^2}{1 + \sigma_i^2 w_i^2}\bigg|\mathcal{G}_i\right]$$

$$\leq \frac{M^2 N^2}{\lambda} \sum_{i=1}^{t} \frac{\sigma_i^2 w_i^2}{1 + \sigma_i^2 w_i^2}$$

$$\leq \frac{2M^2 N^2}{\lambda} \log\det\left(\sum_{i=1}^{t} \frac{\sigma_i^2}{\lambda} x_i x_i^\top + \mathbf{I}\right), \tag{29}$$

where the first inequality follows from the fact that $\mathbb{E}(X - \mathbb{E}[X])^2 \leq \mathbb{E}[X^2]$, the second follows from Equation (28), the third follows from the definition of $\eta_i$, and the fourth follows from the bound $\sigma_i^2 w_i^2/(1 + \sigma_i^2 w_i^2) \leq \min\{1, \sigma_i^2 w_i^2\}$ together with the result in Equation (25). Furthermore, applying Equation (28) again gives

$$|\ell_i| \leq \left|\frac{\eta_i^2 w_i^2}{1 + \sigma_i^2 w_i^2}\right| + \left|\mathbb{E}\left[\frac{\eta_i^2 w_i^2}{1 + \sigma_i^2 w_i^2}\bigg|\mathcal{G}_i\right]\right| \leq \frac{2M^2 N^2}{\lambda}, \tag{30}$$

almost surely. Therefore by Equation (29) and Equation (30), using Lemma H.1, we know that for any $t$, with probability at least $1 - \delta/(4t^2)$, we have

$$\sum_{i=1}^{t} \frac{\eta_i^2 w_i^2}{1 + \sigma_i^2 w_i^2}$$

$$\leq \sum_{i=1}^{t} \mathbb{E}\left[\frac{\eta_i^2 w_i^2}{1 + \sigma_i^2 w_i^2}\bigg|\mathcal{G}_i\right] + \sqrt{\frac{4M^2 N^2}{\lambda} \log\det\left(\sum_{i=1}^{t} \frac{\sigma_i^2}{\lambda} x_i x_i^\top + \mathbf{I}\right) \log(4t^2/\delta)}$$

$$+ \frac{2}{3} \cdot \frac{2M^2 N^2}{\lambda} \log(t^2/\delta)$$

$$\leq \sum_{i=1}^{t} \frac{\sigma_i^2 w_i^2}{1 + \sigma_i^2 w_i^2} + \sqrt{\frac{4M^2 N^2}{\lambda} \log\det\left(\sum_{i=1}^{t} \frac{\sigma_i^2}{\lambda} x_i x_i^\top + \mathbf{I}\right) \log(4t^2/\delta)}$$

$$+ \frac{4M^2 N^2}{\lambda} \log(4t^2/\delta)$$

$$\leq 2\log\det\left(\sum_{i=1}^{t} \frac{\sigma_i^2}{\lambda} x_i x_i^\top + \mathbf{I}\right) + \sqrt{\frac{4M^2 N^2}{\lambda} \log\det\left(\sum_{i=1}^{t} \frac{\sigma_i^2}{\lambda} x_i x_i^\top + \mathbf{I}\right) \log(4t^2/\delta)}$$

$$+ \frac{4M^2 N^2}{\lambda} \log(4t^2/\delta)$$

$$\leq \frac{1}{4} \cdot \left[8\sqrt{\log\det\left(\sum_{i=1}^{t} \frac{\sigma_i^2}{\lambda} x_i x_i^\top + \mathbf{I}\right) \log(4t^2/\delta)} + \frac{4MN}{\sqrt{\lambda}} \log(4t^2/\delta)\right]^2$$

$$= \frac{1}{4}\beta_t^2, \tag{31}$$

where the second inequality follows from the definition of $\sigma_i^2$, the third follows from the bound $\sigma_i^2 w_i^2/(1 + \sigma_i^2 w_i^2) \leq \min\{1, \sigma_i^2 w_i^2\}$ together with the result in Equation (25), and the final inequality follows from the definition of $\beta_t$. Applying a union bound to Equation (31) for $t = 1$ to $\infty$ and using the fact that $\sum_{t=1}^{\infty} t^{-2} < 2$ completes the proof.

# E PROOF OF LEMMAS IN SECTION 6

For clarity, we assume that Condition 4.4 always holds. We then define the quantity $\alpha(z', z'')$ via the mean-value theorem, and introduce two additional analogous definitions for brevity as follows:

$$\alpha(z', z'') = \frac{\mu(z') - \mu(z'')}{z' - z''} = \int_{v=0}^{1} \dot{\mu}\big(z' + v(z'' - z')\big) dv,$$

$$\alpha(x, \theta', \theta'') = \alpha\Big(g(x; \theta_0)^\top (\theta' - \theta_0), g(x; \theta_0)^\top (\theta'' - \theta_0)\Big),$$

$$\alpha(x', x'', \theta) = \alpha\Big(g(x'; \theta_0)^\top (\theta - \theta_0), g(x''; \theta_0)^\top (\theta - \theta_0)\Big). \tag{32}$$

For the design matrix $X_t$ associated with the time-varying regularization parameter $\lambda_t$, we denote by $\widetilde{X}_t$ the corresponding matrix formed using the initial regularization parameter $\lambda_0$. For example,

$$\widetilde{V}_t = \sum_{i=1}^{t} \frac{1}{m} g(x_i; \theta_0) g(x_i; \theta_0)^\top + \kappa \lambda_0 \mathbf{I},$$

$$\widetilde{H}_t(\theta) = \sum_{i=1}^{t} \frac{1}{m} \dot{\mu}(g(x_i; \theta_0)^\top (\theta_i - \theta_0)) g(x_i; \theta_0) g(x_i; \theta_0)^\top + \lambda_0 \mathbf{I}$$

$$\widetilde{W}_t = \sum_{i=1}^{t} \frac{1}{m} \dot{\mu}(f(x_i; \theta_i)) g(x_i; \theta_0) g(x_i; \theta_0)^\top + \lambda_0 \mathbf{I}. \tag{33}$$

## E.1 PROOF OF LEMMA 4.5

First, we define the auxiliary loss $\widetilde{L}_t(\theta)$

$$\widetilde{\mathcal{L}}_t(\theta) = -\sum_{i=1}^{t} r_i \log \mu(g(x_i; \theta_0)^\top (\theta - \theta_0)) + (1 - r_i) \log(1 - \mu(g(x_i; \theta_0)^\top (\theta - \theta_0)))$$

$$+ \frac{m\lambda_t}{2} \|\theta - \theta_0\|_2^2,$$

and its maximum likelihood estimator $\hat{\theta}_t = \arg\min_\theta \widetilde{\mathcal{L}}_t(\theta)$. Then, we use the following definitions:

$$\gamma_t(\theta) = \sum_{i=1}^{t} \frac{1}{m} \mu(g(x_i; \theta_0)^\top (\theta - \theta_0)) g(x_i; \theta_0) + \lambda_t (\theta - \theta_0)$$

$$\Gamma_t(\theta', \theta'') = \sum_{i=1}^{t} \frac{1}{m} \alpha(x_i, \theta', \theta'') g(x_i; \theta_0) g(x_i; \theta_0)^\top + \lambda_t \mathbf{I},$$

where $\alpha(x_i, \theta', \theta'')$ is defined at Equation (32). We can see that

$$\gamma_t(\theta) - \gamma_t(\theta^*)$$

$$= \sum_{i=1}^{t} \frac{1}{m} \Big( \mu(g(x_i; \theta_0)^\top (\theta - \theta_0)) - \mu(g(x_i; \theta_0)^\top (\theta^* - \theta_0)) \Big) g(x_i; \theta_0) + \lambda_t (\theta - \theta^*)$$

$$= \sum_{i=1}^{t} \frac{1}{m} \alpha(x_i, \theta, \theta^*) g(x_i; \theta_0) g(x_i; \theta_0)^\top (\theta - \theta^*) + \lambda_t (\theta - \theta^*)$$

$$= \Gamma_t(\theta, \theta^*)(\theta - \theta^*),$$

which implies that

$$\|\theta - \theta^*\|_{\Gamma_t(\theta, \theta^*)} = \|\gamma(\theta) - \gamma(\theta^*)\|_{\Gamma_t^{-1}(\theta, \theta^*)}. \tag{34}$$

Now we provide the following two lemmas:

**Lemma E.1.** *For $\delta \in (0, 1]$, define*

$$\mathcal{C}_t = \left\{\theta : \sqrt{m}\|\gamma_t(\theta) - \gamma_t(\hat{\theta}_t)\|_{H_t^{-1}(\theta)} \leq \iota_t\right\}, \tag{35}$$

*where $\iota_t$ is defined at Equation (3). Then for all $t \geq 0$, $\theta^* \in \mathcal{C}_t$ with probability at least $1 - \delta$*

**Lemma E.2.** *Let $\delta \in (0, 1]$. Define $\mathcal{C}_t$ as in Equation (35). There exists an absolute constant $C_1 > 0$ such that for all $\theta \in \mathcal{C}_t$,*

$$H_t(\theta) \preceq \left(1 + C_1^2 \frac{L}{\lambda_t}\iota_t^2 + C_1\sqrt{\frac{L}{\lambda_t}}\iota_t\right)\Gamma_t(\theta, \hat{\theta}_t), \quad H_t(\hat{\theta}_t) \preceq \left(1 + C_1^2 \frac{L}{\lambda_t}\iota_t^2 + C_1\sqrt{\frac{L}{\lambda_t}}\iota_t\right)\Gamma_t(\theta, \hat{\theta}_t).$$

Now we are ready to start the proof.

*Proof of Lemma 4.5.* For the absolute constants $\{C_i\}_{i=1}^3$, we can start with

$$\sqrt{m}\|\theta_t - \theta^*\|_{H_t(\theta^*)} \tag{36}$$
$$\leq \sqrt{m}\|\hat{\theta}_t - \theta^*\|_{H_t(\theta^*)} + \sqrt{m}\|\theta_t - \hat{\theta}_t\|_{H_t(\theta^*)}$$
$$\leq \sqrt{m}\|\hat{\theta}_t - \theta^*\|_{H_t(\theta^*)} + \sqrt{m}\|\theta_t - \hat{\theta}_t\|_2 \cdot (\lambda_t + C_1 tL)$$
$$\leq \underbrace{\sqrt{m}\|\hat{\theta}_t - \theta^*\|_{H_t(\theta^*)}}_{\textbf{(term 1)}} + \underbrace{2(\lambda_t + C_1 tL)(1 - \eta m \lambda_t)^{J/2} t^{1/2}\lambda_t^{-1/2}}_{\textbf{(term 2)}}$$
$$+ \underbrace{(\lambda_t + C_1 tL)\left[C_2 m^{-1/6}\sqrt{\log m}\, t^{7/6}\lambda_t^{-7/6}L^{7/2} + C_1 C_3 R m^{-1/6}\sqrt{\log m}\, L^{7/2}t^{5/3}\lambda_t^{-5/3}\right]}_{\textbf{(term 3)}}. \tag{37}$$

The first inequality is due to triangle inequality. The second inequality is due to $\lambda_{\max}(H_t(\theta^*)) \leq \lambda_t + t \times \|\sqrt{\dot{\mu}(\cdot)/m} \cdot g(\cdot)\|_2^2 \leq \lambda_t + C_1 tL$ where we used Lemma C.4. Finally, the last inequality follows from Lemma C.6.

For **(term 1)**, we rewrite the definition of $\iota_t$ and $\lambda_t$:

$$\iota_t = 16\sqrt{\log\det\left(\sum_{i=1}^t \frac{1}{4m^2\lambda_0}g(x_i; \theta_0)g(x_i; \theta_0)^\top + \mathbf{I}\right)\log\frac{4t^2}{\delta}} + 8C_1\sqrt{\frac{L}{\lambda_0}}\log\frac{4t^2}{\delta}$$

$$\lambda_t = \frac{64}{S^2}\log\det\left(\sum_{i=1}^t \frac{1}{4m\lambda_0}g(x_i; \theta_0)g(x_i; \theta_0)^\top + \mathbf{I}\right)\log\frac{4t^2}{\delta} + \frac{16C_1^2 L}{S^2\lambda_0}\log^2\frac{4t^2}{\delta}.$$

We can see that $\iota_t^2/\lambda_t \leq 8S^2$ (and $\iota_t/\sqrt{\lambda_t} \leq 2\sqrt{2}S$) by the fact that $(a + b)^2 \leq 2a^2 + 2b^2$. Therefore, applying these with Lemmas E.1 and E.2 gives

$$H_t(\theta^*) \preceq (1 + 2\sqrt{2}C_1\sqrt{L}S + 8C_1^2 LS^2)\Gamma_t(\theta^*, \hat{\theta}_t), \tag{38}$$

for some absolute constant $C_1 > 0$. Now, back to **(term 1)**, we have

$$\sqrt{m}\left\|\hat{\theta}_t - \theta^*\right\|_{H_t(\theta^*)} \leq \sqrt{m(1 + 2\sqrt{2}C_1\sqrt{L}S + 8C_1^2 LS^2)}\left\|\hat{\theta}_t - \theta^*\right\|_{\Gamma_t(\hat{\theta}_t, \theta^*)}$$
$$= \sqrt{m(1 + 2\sqrt{2}C_1\sqrt{L}S + 8C_1^2 LS^2)}\left\|\gamma(\hat{\theta}_t) - \gamma(\theta^*)\right\|_{\Gamma_t^{-1}(\hat{\theta}_t, \theta^*)}$$
$$\leq (1 + 2\sqrt{2}C_1\sqrt{L}S + 8C_1^2 LS^2)\sqrt{m}\left\|\gamma(\hat{\theta}_t) - \gamma(\theta^*)\right\|_{H_t^{-1}(\theta^*)}$$
$$\leq (1 + 2\sqrt{2}C_1\sqrt{L}S + 8C_1^2 LS^2)\iota_t.$$

where the first and the second inequalities follow from Equation (38), the equality is due to Equation (34), and the last inequality follows from Lemma E.1.

For **(term 2)**, plugging in $J = 2\log(\lambda_t S/(T^{1/2}\lambda_t + C_4 T^{3/2}L))TL/\lambda_t, \eta = C_5(mTL + m\lambda_t)^{-1}$ gives

$$2(\lambda_t + C_1 tL)(1 - \eta m\lambda_t)^{J/2} t^{1/2}\lambda_t^{-1/2}$$
$$\leq 2(\lambda_t + C_1 TL)(1 - \lambda_t/(TL))^{J/2} T^{1/2}\lambda_t^{-1/2}$$
$$\leq 2S\sqrt{\lambda_t}$$
$$\leq \iota_t,$$

where the last inequality follows from the definition of $\lambda_t$ and the fact that $\sqrt{a+b} \leq \sqrt{a} + \sqrt{b}$. For **(term 3)**, recall that $\lambda_0 \leq \min\{\lambda_t\}_{t \geq 1}$, then we have

$$C_2 m^{-1/6}\sqrt{\log m}t^{7/6}\lambda_t^{-1/6}L^{7/2} + C_1 C_3 R m^{-1/6}\sqrt{\log m}L^{7/2}t^{5/3}\lambda_t^{-2/3}$$
$$+ C_1 C_2 m^{-1/6}\sqrt{\log m}t^{13/6}\lambda_t^{-7/6}L^{9/2} + C_1^2 C_3 R m^{-1/6}\sqrt{\log m}L^{9/2}t^{8/3}\lambda_t^{-5/3}$$
$$\leq C_2 m^{-1/6}\sqrt{\log m}T^{7/6}\lambda_0^{-1/6}L^{7/2} + C_1 C_3 R m^{-1/6}\sqrt{\log m}L^{7/2}T^{5/3}\lambda_0^{-2/3}$$
$$+ C_1 C_2 m^{-1/6}\sqrt{\log m}T^{13/6}\lambda_0^{-7/6}L^{9/2} + C_1^2 C_3 R m^{-1/6}\sqrt{\log m}L^{9/2}T^{8/3}\lambda_0^{-5/3}$$
$$\leq 1,$$

where the last inequality can be verified that if the width of the NN $m$ is large enough, satisfying the condition on Condition C.2, **(term 3)** $\leq 1$.

Substituting **(term 1)**, **(term 2)**, and **(term 3)** back to Equation (37) gives

$$\sqrt{m}\|\theta_t - \theta^*\|_{H_t(\theta^*)} \leq (2 + 2\sqrt{2}C_1\sqrt{L}S + 8C_1^2 LS^2)\iota_t + 1$$
$$\leq C_6(1 + \sqrt{L}S + LS^2)\iota_t + 1,$$

for some absolute constant $C_6 > 0$, concludes the proof. $\qquad\square$

### E.2 Proof of Lemma E.1

Recall the definition of $\widetilde{\mathcal{L}}_t(\theta)$, $\hat{\theta}_t$, $\gamma_t(\theta)$, and $\Gamma_t(\theta', \theta'')$ from Section E.1. Since $\hat{\theta}_t$ is a maximum likelihood estimator, $\widetilde{L}_t(\hat{\theta}_t) = 0$, which gives

$$\sum_{i=1}^{t}\frac{1}{m}\mu(g(x_i;\theta_0)^\top(\hat{\theta}_t - \theta_0))g(x_i;\theta_0) + \lambda_t(\hat{\theta}_t - \theta_0) = \sum_{i=1}^{t}\frac{1}{m}r_i g(x_i;\theta_0). \tag{39}$$

Therefore, we can see that

$$\sqrt{m}\|\gamma(\hat{\theta}_t) - \gamma(\theta^*)\|_{H_t^{-1}(\theta^*)}$$
$$= \sqrt{m}\left\|\sum_{i=1}^{t}\frac{1}{m}[\mu(g(x_i;\theta_0)^\top(\hat{\theta}_t - \theta_0)) - \mu(g(x_i;\theta)^\top(\theta^* - \theta_0))]g(x_i;\theta_0) + \lambda_t\hat{\theta}_t - \lambda_t\theta^*\right\|_{H_t^{-1}(\theta^*)}$$
$$= \sqrt{m}\left\|\sum_{i=1}^{t}\frac{1}{m}[r_i - \mu(g(x_i;\theta_0)^\top(\theta^* - \theta_0))]g(x_i;\theta_0) - \lambda_t(\theta^* - \theta_0)\right\|_{H_t^{-1}(\theta^*)}$$
$$\leq \underbrace{\left\|\sum_{i=1}^{t}\frac{1}{\sqrt{m}}[r_i - \mu(g(x_i;\theta_0)^\top(\theta^* - \theta_0))]g(x_i;\theta_0)\right\|_{H_t^{-1}(\theta^*)}}_{\textbf{(term 1)}} + \underbrace{\sqrt{\lambda_t m}\|\theta^* - \theta_0\|_2}_{\textbf{(term 2)}}, \tag{40}$$

where the first equality follows from the definition, the second equality is due to Equation (39), and the first inequality follows from triangle inequality, and the fact that $\lambda_{\max}(H_t^{-1}(\theta^*)) \leq 1/\sqrt{\lambda_t}$.

For **(term 1)**, we are going to use our new tail inequality for martingales in Theorem 3.1. Define $\eta_i = r_i - \mu(g(x_i;\theta_0)^\top(\theta^* - \theta_0)) = r_i - \mu(h(x_i))$. Then, we can see the following conditions are satisfied:

$$|\eta_i| \leq 1, \ \mathbb{E}[\eta_i|\mathcal{G}_i] = 0, \ \mathbb{E}[\eta_i^2|\mathcal{G}_i] = \dot{\mu}(g(x_i;\theta_0)^\top(\theta^* - \theta_0)).$$

By Lemma C.4 we have $\|g(x_i; \theta_0)/\sqrt{m}\|_2 \leq C_1\sqrt{L}$ for some absolute constant $C_1 > 0$. Therefore, applying Theorem 3.1 gives

$$\left\| \sum_{i=1}^{t} \frac{1}{\sqrt{m}} \eta_t g(x_i; \theta_0) \right\|_{H_t^{-1}(\theta^*)}$$

$$\leq \left\| \sum_{i=1}^{t} \frac{1}{\sqrt{m}} \eta_t g(x_i; \theta_0) \right\|_{\widetilde{H}_t^{-1}(\theta^*)}$$

$$\leq 8\sqrt{\log \det \left( \sum_{i=1}^{t} \frac{1}{4m\lambda_0} g(x_i; \theta_0) g(x_i; \theta_0)^\top + I \right) \log \frac{4t^2}{\delta}} + 4C_1 \sqrt{\frac{L}{\lambda_0}} \log \frac{4t^2}{\delta}, \qquad (41)$$

with probability at least $1 - \delta$. Substituting Equation (41) into Equation (40) gives

$$\sqrt{m}\|\gamma(\hat{\theta}_t) - \gamma(\theta^*)\|_{H_t^{-1}(\theta^*)}$$

$$\leq 8\sqrt{\log \det \left( \sum_{i=1}^{t} \frac{1}{4m\lambda_0} g(x_i; \theta_0) g(x_i; \theta_0)^\top + \mathbf{I} \right) \log \frac{4t^2}{\delta}} + 4C_1 \sqrt{\frac{L}{\lambda_0}} \log \frac{4t^2}{\delta} + S\sqrt{\lambda_t}$$

$$\leq 16\sqrt{\log \det \left( \sum_{i=1}^{t} \frac{1}{4m\lambda_0} g(x_i; \theta_0) g(x_i; \theta_0)^\top + \mathbf{I} \right) \log \frac{4t^2}{\delta}} + 8C_1 \sqrt{\frac{L}{\lambda_0}} \log \frac{4t^2}{\delta}$$

$$= \iota_t. \qquad (42)$$

where the last inequality is due to the update rule of $\lambda_t$ and the fact that $\sqrt{a+b} \leq \sqrt{a} + \sqrt{b}$. We finish the proof.

### E.3 PROOF OF LEMMA E.2

We modified the previous results of Abeille et al. (2021) (Lemma 2), and Faury et al. (2022) (proof of Lemma 1), proper to our settings.

**Lemma E.3.** *Let $\delta \in (0,1]$. Define $\mathcal{C}_t$ as in Equation (35). There exists and absolute constant $C_1 > 0$ such that for all $\theta \in \mathcal{C}_t$:*

$$\sqrt{m}\|\gamma_t(\theta) - \gamma_t(\hat{\theta}_t)\|_{\Gamma_t^{-1}(\theta, \hat{\theta}_t)} \leq C_1 \sqrt{\frac{L}{\lambda_t}} \iota_t^2 + \iota_t.$$

The proof is deferred to Section E.4. Following the proof of Lemma E.3, from Equation (43), we have

$$\Gamma_t(\theta, \hat{\theta}_t) \geq \left( 1 + C_1 L^{1/2} \lambda_t^{-1/2} \cdot \sqrt{m}\|\gamma_t(\theta) - \gamma_t(\hat{\theta}_t)\|_{\Gamma_t^{-1}(\theta, \hat{\theta}_t)} \right)^{-1} H_t(\theta)$$

$$\geq \left( 1 + C_1^2 \frac{L}{\lambda_t} \iota_t^2 + C_1 \sqrt{\frac{L}{\lambda_t}} \iota_t \right)^{-1} H_t(\theta)$$

where the last inequality follows from applying Lemma E.3 again.

One can achieve the same result for $H_t(\hat{\theta}_t)$ in a similarly way by starting the proof of Lemma E.3 with

$$\Gamma_t(\theta, \hat{\theta}_t) = \sum_{i=1}^{t} \alpha(x_i, \theta, \hat{\theta}_t) g(x_i; \theta_0) g(x_i; \theta_0)^\top + \lambda_t \mathbf{I}$$

$$\geq \sum_{i=1}^{t} \left( 1 + |g(x_i; \theta_0)^\top (\theta - \hat{\theta}_t)| \right)^{-1} \dot{\mu}(g(x_i; \theta_0)^\top (\hat{\theta}_t - \theta_0)) g(x_i; \theta_0) g(x_i; \theta_0)^\top + \lambda_t \mathbf{I}$$

$$\geq \left( 1 + C_1 \sqrt{\frac{L}{\lambda_t}} \cdot \sqrt{m}\|\gamma_t(\theta) - \gamma_t(\hat{\theta}_t)\|_{\Gamma_t^{-1}(\theta, \hat{\theta}_t)} \right)^{-1} H_t(\hat{\theta}_t)$$

$$\geq \left( 1 + C_1^2 \frac{L}{\lambda_t} \iota_t^2 + C_1 \sqrt{\frac{L}{\lambda_t}} \iota_t \right)^{-1} H_t(\hat{\theta}_t),$$

where the first inequality follows from Lemma H.3, the second inequality follows the same process of Equation (43), and the last inequality follows from Lemma E.3, finishing the proof.

### E.4 PROOF OF LEMMA E.3

Recall the definition of $\Gamma_t$ and $\alpha(x, \theta', \theta'')$ from Equation (32). We start with

$$
\begin{aligned}
\Gamma_t(\theta, \hat{\theta}_t) &= \sum_{i=1}^{t} \alpha(x_i, \theta, \hat{\theta}_t) g(x_i; \theta_0) g(x_i; \theta_0)^\top + \lambda_t \mathbf{I} \\
&\geq \sum_{i=1}^{t} \big(1 + \underbrace{|g(x_i; \theta_0)^\top (\theta - \hat{\theta}_t)|}_{\textbf{(term 1)}}\big)^{-1} \dot{\mu}(g(x_i; \theta_0)^\top (\theta - \theta_0)) g(x_i; \theta_0) g(x_i; \theta_0)^\top + \lambda_t \mathbf{I},
\end{aligned}
$$

where the inequality follows from Lemma H.3. For **(term 1)**, we have,

$$
\begin{aligned}
\textbf{(term 1)} &\leq \|g(x_i; \theta_0)/\sqrt{m}\|_{\Gamma_t^{-1}(\theta, \hat{\theta}_t)} \cdot \sqrt{m}\|\theta - \hat{\theta}_t\|_{\Gamma_t(\theta, \hat{\theta}_t)} \\
&\leq C_1 L^{1/2} \lambda_t^{-1/2} \cdot \sqrt{m}\|\theta - \hat{\theta}_t\|_{\Gamma_t(\theta, \hat{\theta}_t)} \\
&\leq C_1 L^{1/2} \lambda_t^{-1/2} \cdot \sqrt{m}\|\gamma_t(\theta) - \gamma_t(\hat{\theta}_t)\|_{\Gamma_t^{-1}(\theta, \hat{\theta}_t)},
\end{aligned}
$$

where the first inequality follows from the Cauchy-Schwarz inequality, the second inequality follows from the fact that $\lambda_{\max}(\Gamma(\cdot)^{-1}) \leq \lambda_t^{-1}$ and Lemma C.4, and the last inequality follows from Equation (34). Substituting **(term 1)** back gives,

$$
\begin{aligned}
\Gamma_t(\theta, \hat{\theta}_t) &\geq \Big(1 + C_1 L^{1/2} \lambda_t^{-1/2} \cdot \sqrt{m}\|\gamma_t(\theta) - \gamma_t(\hat{\theta}_t)\|_{\Gamma_t^{-1}(\theta, \hat{\theta}_t)}\Big)^{-1} \\
&\quad \times \sum_{i=1}^{t} \dot{\mu}(g(x_i; \theta_0)^\top (\theta - \theta_0)) g(x_i; \theta_0) g(x_i; \theta_0)^\top + \lambda_t \mathbf{I} \\
&\geq \Big(1 + C_1 L^{1/2} \lambda_t^{-1/2} \cdot \sqrt{m}\|\gamma_t(\theta) - \gamma_t(\hat{\theta}_t)\|_{\Gamma_t^{-1}(\theta, \hat{\theta}_t)}\Big)^{-1} \\
&\quad \times \Big(\sum_{i=1}^{t} \dot{\mu}(g(x_i; \theta_0)^\top (\theta - \theta_0)) g(x_i; \theta_0) g(x_i; \theta_0)^\top + \lambda_t \mathbf{I}\Big) \\
&= \Big(1 + C_1 L^{1/2} \lambda_t^{-1/2} \cdot \sqrt{m}\|\gamma_t(\theta) - \gamma_t(\hat{\theta}_t)\|_{\Gamma_t^{-1}(\theta, \hat{\theta}_t)}\Big)^{-1} H_t(\theta) \qquad (43)
\end{aligned}
$$

Using this results, we can further obtain

$$
\begin{aligned}
&\sqrt{m}\|\gamma_t(\theta) - \gamma_t(\hat{\theta}_t)\|_{\Gamma_t^{-1}(\theta, \hat{\theta}_t)}^2 \\
&\leq \Big(1 + C_1 L^{1/2} \lambda_t^{-1/2} \cdot \sqrt{m}\|\gamma_t(\theta) - \gamma_t(\hat{\theta}_t)\|_{\Gamma_t^{-1}(\theta, \hat{\theta}_t)}\Big) \cdot \sqrt{m}\|\gamma_t(\theta) - \gamma_t(\hat{\theta}_t)\|_{H_t^{-1}(\theta)}^2 \\
&\leq \iota_t^2 + C_1 L^{1/2} \lambda_t^{-1/2} \iota_t^2 \cdot \sqrt{m}\|\gamma_t(\theta) - \gamma_t(\hat{\theta}_t)\|_{\Gamma_t^{-1}(\theta, \hat{\theta}_t)},
\end{aligned}
$$

where the last inequality follows from Lemma E.1. We solve the polynomial inequality in $\sqrt{m}\|\gamma_t(\theta) - \gamma_t(\hat{\theta}_t)\|_{\Gamma_t^{-1}(\theta, \hat{\theta}_t)}$ using a fact that for $b, c > 0$ and $x \in \mathbb{R}$, following implication holds: $x^2 \leq bx + c \implies x \leq b + \sqrt{c}$, which finally gives

$$
\sqrt{m}\|\gamma_t(\theta) - \gamma_t(\hat{\theta}_t)\|_{\Gamma_t^{-1}(\theta, \hat{\theta}_t)} \leq C_1 \sqrt{\frac{L}{\lambda_t}} \iota_t^2 + \iota_t
$$

### E.5 PROOF OF LEMMA 6.4

First, we show that we can upper bound on the prediction error for all $x \in \mathcal{X}_t$, $t \in [T]$, which is the difference between the true reward $\mu(h(x))$ with our prediction with the neural network $\mu(f(x; \theta_t))$.

For $x \in \mathcal{X}_{t+1}$ and the absolute constant $C_3 > 0$, the prediction error is defined as

$$
\begin{aligned}
|\mu(h(x)) &- \mu(f(x; \theta_t))| \\
&\leq R[h(x) - f(x; \theta_t)] \\
&= R[g(x; \theta_0)^\top (\theta^* - \theta_0) - f(x; \theta_t)] \\
&\leq R[\underbrace{g(x; \theta_0)^\top (\theta^* - \theta_0) - g(x; \theta_0)^\top (\theta_t - \theta_0)}_{\textbf{(term 1)}} + C_3 m^{-1/6} \sqrt{\log m} L^3 t^{2/3} \lambda_t^{-2/3}],
\end{aligned}
\tag{44}
$$

where the first inequality is due to the fact that $\mu(\cdot)$ is $R$-Lipschitz function, the equality follows from Lemma 6.1, and the last inequality follows from Lemma C.5. For **(term 1)**, we have

$$
\begin{aligned}
\textbf{(term 1)} &= g(x; \theta_0)^\top (\theta^* - \theta_t) \\
&= \frac{1}{\sqrt{m}} g(x; \theta_0)^\top \cdot H_t^{-1/2}(\theta^*) \cdot H_t^{1/2}(\theta^*) \cdot \sqrt{m}(\theta^* - \theta_t) \\
&\leq \|g(x; \theta_0)/\sqrt{m}\|_{H_t^{-1}(\theta^*)} \cdot \sqrt{m}\|\theta^* - \theta_t\|_{H_t(\theta^*)} \\
&\leq \sqrt{\kappa}\|g(x; \theta_0)/\sqrt{m}\|_{V_t^{-1}} \cdot \sqrt{m}\|\theta^* - \theta_t\|_{H_t(\theta^*)} \\
&\leq \sqrt{\kappa}\|g(x; \theta_0)/\sqrt{m}\|_{V_t^{-1}} \cdot \left(C_6(1 + \sqrt{L}S + LS^2)\iota_t + 1\right),
\end{aligned}
\tag{45}
$$

where the first inequality follows from the Cauchy-Schwarz inequality, the second inequality is due to the Assumption 6.3 that $\frac{1}{\kappa} V_t \preceq H_t^{-1}(\theta^*)$, and the last inequality follows from Lemma 4.5. Plugging Equation (45) into Equation (44) gives

$$
\begin{aligned}
|\mu(h(x)) &- \mu(f(x; \theta_t))| \\
&\leq R\sqrt{\kappa}\left(C_6(1 + \sqrt{L}S + LS^2)\iota_t + 1\right)\|g(x; \theta_0)/\sqrt{m}\|_{V_t^{-1}} + C_3 R m^{-1/6} \sqrt{\log m} L^3 t^{2/3} \lambda_t^{-2/3} \\
&\leq R\sqrt{\kappa}\left(C_6(1 + \sqrt{L}S + LS^2)\iota_t + 1\right)\|g(x; \theta_0)/\sqrt{m}\|_{V_t^{-1}} + \epsilon_{3,t},
\end{aligned}
\tag{46}
$$

where the second inequality follows from the fact that $\lambda_0 \leq \min\{\lambda_t\}_{t \geq 1}$ and the definition of $\epsilon_{3,t}$.

# F PROOF OF LEMMAS IN SECTION 5

## F.1 PROOF OF LEMMA 5.1

Recall the definition of $\widetilde{L}_t(\theta)$, $\hat{\theta}_t$, $\gamma_t(\theta)$, $\Gamma_t(\theta', \theta'')$, $\iota_t$, and $\lambda_t$ from Section E.1. We also use

$$
W_t = \sum_{i=1}^{t} \frac{\dot{\mu}(f(x_i; \theta_i))}{m} g(x_i; \theta_0) g(x_i; \theta_0)^\top + \lambda_t \mathbf{I}
$$

$$
H_t(\hat{\theta}_t) = \sum_{i=1}^{t} \frac{\dot{\mu}(g(x_i; \theta_0)^\top (\hat{\theta}_t - \theta_0))}{m} g(x_i; \theta_0) g(x_i; \theta_0)^\top + \lambda_t \mathbf{I}
$$

$$
Z_t = \sum_{i=1}^{t} \frac{|\dot{\mu}(f(x_i; \theta_i)) - \dot{\mu}(g(x_i; \theta_0)^\top (\hat{\theta}_t - \theta_0))|}{m} g(x_i; \theta_0) g(x_i; \theta_0)^\top + \lambda_t \mathbf{I}
$$

By the definition of $Z_t$, for any $x \in \mathbb{R}^p$, we have

$$
\|x\|_{W_t} \leq \|x\|_{H_t(\hat{\theta}_t) + Z_t} \leq \|x\|_{H_t(\hat{\theta}_t)} + \|x\|_{Z_t}.
$$

Now with the above inequality, we can start with

$$
\sqrt{m}\|\theta_t - \theta^*\|_{W_t} \leq \underbrace{\sqrt{m}\|\theta_t - \theta^*\|_{H_t(\hat{\theta}_t)}}_{\textbf{(term 1)}} + \underbrace{\sqrt{m}\|\theta_t - \theta^*\|_{Z_t}}_{\textbf{(term 2)}}.
\tag{47}
$$

For **(term 1)**, we directly follow the proof of Lemma 4.5 in Section E.1. Therefore, for the absolute constants $\{C_i\}_{i=1}^3$, we have

$$\sqrt{m}\|\theta_t - \theta^*\|_{H_t(\hat{\theta}_t)}$$

$$\leq \sqrt{m}\|\hat{\theta}_t - \theta^*\|_{H_t(\hat{\theta}_t)} + \sqrt{m}\|\theta_t - \hat{\theta}_t\|_{H_t(\hat{\theta}_t)}$$

$$\leq \underbrace{\sqrt{m}\|\hat{\theta}_t - \theta^*\|_{H_t(\hat{\theta}_t)}}_{\textbf{(term 3)}} + \underbrace{2(\lambda_t + C_1 t L)(1 - \eta m \lambda_t)^{J/2} t^{1/2} \lambda_t^{-1/2}}_{\textbf{(term 4)}}$$

$$+ \underbrace{(\lambda_t + C_1 t L)\Big[C_2 m^{-1/6}\sqrt{\log m} t^{7/6}\lambda_t^{-7/6}L^{7/2} + C_1 C_3 R m^{-1/6}\sqrt{\log m}L^{7/2}t^{5/3}\lambda_t^{-5/3}\Big]}_{\textbf{(term 5)}}.$$

Using the same argument as in Section E.1, we can see that

$$\textbf{(term 4)} \leq 2S\sqrt{\lambda_t} \leq \iota_t, \quad \textbf{(term 5)} \leq 1/2.$$

Note that the upper bound for **(term 5)** has been changed from 1 to $1/2$ solely to unify the constant in the concentration inequalities of Lemma 4.5 and Lemma 5.1. For **(term 3)**, we have

$$\sqrt{m}\left\|\hat{\theta}_t - \theta^*\right\|_{H_t(\hat{\theta}_t)} \leq \sqrt{m(1 + 2\sqrt{2}C_1\sqrt{L}S + 8C_1^2 LS^2)}\left\|\hat{\theta}_t - \theta^*\right\|_{\Gamma_t(\hat{\theta}_t, \theta^*)}$$

$$= \sqrt{m(1 + 2\sqrt{2}C_1\sqrt{L}S + 8C_1^2 LS^2)}\left\|\gamma(\hat{\theta}_t) - \gamma(\theta^*)\right\|_{\Gamma_t^{-1}(\hat{\theta}_t, \theta^*)}$$

$$\leq (1 + 2\sqrt{2}C_1\sqrt{L}S + 8C_1^2 LS^2)\sqrt{m}\left\|\gamma(\hat{\theta}_t) - \gamma(\theta^*)\right\|_{H_t^{-1}(\theta^*)}$$

$$\leq (1 + 2\sqrt{2}C_1\sqrt{L}S + 8C_1^2 LS^2)\iota_t,$$

where the first and the second inequalities follow from Lemma E.2, the equality follows from Equation (38), and the last inequality follows from Lemma E.1. Plugging **(term 3-5)** into **(term 1)** gives

$$\textbf{(term 1)} \leq (2 + C_1\sqrt{L}S + C_1 LS^2)\iota_t + 1/2.$$

Now, moving on to **(term 2)**, we have

$$\sqrt{m}\|\theta_t - \theta^*\|_{Z_t} \leq \underbrace{\sqrt{m}\|\theta_t - \theta^*\|_2}_{\textbf{(term 6)}} \times \underbrace{\lambda_{\max}^{1/2}(Z_t)}_{\textbf{(term 7)}}.$$

For **(term 7)**, we have

$$\lambda_{\max}^{1/2}(Z_t) = \lambda_{\max}^{1/2}\left(\sum_{i=1}^t \frac{|\dot{\mu}(f(x_i; \theta_i)) - \dot{\mu}(g(x_i; \theta_0)^\top(\hat{\theta}_t - \theta_0))|}{m}g(x_i; \theta_0)g(x_i; \theta_0)^\top\right)$$

$$\leq \lambda_{\max}^{1/2}\left(\sum_{i=1}^t \frac{C_3 R m^{-1/6}\sqrt{\log m}L^3 t^{2/3}\lambda_t^{-2/3}}{m}g(x_i; \theta_0)g(x_i; \theta_0)^\top\right)$$

$$\leq C_3 R^{1/2}m^{-1/12}(\log m)^{1/4}t^{5/6}L^2\lambda_t^{-1/3}.$$

Here, the first inequality follows from the Lipschitz continuity of $\dot{\mu}$, the bounds $|\ddot{\mu}| \leq \dot{\mu} \leq R$, and Lemma C.5, while the final inequality follows from $\lambda_{\max}(\sum_{i=1}^t x_i x_i^\top) \leq \sum_{i=1}^t \|x_i\|_2^2$ and used Lemma C.4. For **(term 6)** we have

$$\sqrt{m}\|\theta_t - \theta^*\|_2 \leq \sqrt{m}\|\theta_t - \theta_0\|_2 + \sqrt{m}\|\theta^* - \theta_0\|_2 \leq 3t^{1/2}\lambda_t^{-1/2} + S,$$

where the last inequality follows from Lemmas C.6 and 6.1. Plugging **(term 6-7)** back to **(term 2)** gives,

$$\textbf{(term 2)} \leq C_3 R^{1/2}m^{-1/12}(\log m)^{1/4}t^{4/3}L^2\lambda_t^{-5/6} + C_3 S R^{1/2}m^{-1/12}(\log m)^{1/4}t^{5/6}L^2\lambda_t^{-1/3}$$

$$\leq C_3 R^{1/2}m^{-1/12}(\log m)^{1/4}T^{4/3}L^2\lambda_0^{-5/6} + C_3 S R^{1/2}m^{-1/12}(\log m)^{1/4}T^{5/6}L^2\lambda_0^{-1/3}$$

$$\leq 1/2 + 2S\lambda_t^{1/2}$$

$$\leq 1/2 + \iota_t,$$

where the third inequality is followed by the condition on $m$ in Condition C.2, and the last inequality is due to the update rule of $\lambda_t$. Finally, substituting **(term 1-2)** into Equation (47) gives

$$\sqrt{m}\|\theta_t - \theta^*\|_{W_t} \le (3 + 2\sqrt{2}C_1\sqrt{L}S + 8C_1^2 LS^2)\iota_t + 1$$
$$\le C_7(1 + \sqrt{L}S + LS^2)\iota_t + 1,$$

for some absolute constant $C_7 > 0$, finishing the proof.

# G  REGRET ANALYSES

## G.1  PROOF OF THEOREM 4.6

We start with a proposition for the per-round regret:

**Proposition G.1.** *Under Condition 4.4, for all $x \in \mathcal{X}_t$, $t \in [T]$, with probability at least $1 - \delta$,*

$$\mu(h(x_t^*)) - \mu(h(x_t)) \le 2R\sqrt{\kappa}((CLS^2 + 2)\iota_{t-1} + 1)\|g(x_t;\theta_0)/\sqrt{m}\|_{V_{t-1}^{-1}} + 2\epsilon_{3,t-1}.$$

*Proof.* We follow the standard procedure to upper bound the per-round regret with the prediction error under the optimistic rule. For all $t \in [T]$ we have

$$\mu(h(x_t^*)) - \mu(h(x_t))$$
$$\le \mu(f(x_t^*;\theta_{t-1})) + R\sqrt{\kappa}\big(C_6(1 + \sqrt{L}S + LS^2)\iota_t + 1\big)\|g(x_t^*;\theta_0)/\sqrt{m}\|_{V_{t-1}^{-1}} + \epsilon_{3,t-1} - \mu(h(x_t))$$
$$\le \mu(f(x_t;\theta_{t-1})) + R\sqrt{\kappa}\big(C_6(1 + \sqrt{L}S + LS^2)\iota_t + 1\big)\|g(x_t;\theta_0)/\sqrt{m}\|_{V_{t-1}^{-1}} + \epsilon_{3,t-1} - \mu(h(x_t))$$
$$\le 2R\sqrt{\kappa}((CLS^2 + 2)\iota_{t-1} + 1)\|g(x_t;\theta_0)/\sqrt{m}\|_{V_{t-1}^{-1}} + 2\epsilon_{3,t-1},$$

where the first and the last inequalities follow from Lemma 6.4, the second inequality comes from the optimistic rule of Algorithm 1, finishing the proof. $\square$

With Proposition G.1, we have

$$\mu(h(x_t^*)) - \mu(h(x_t)) \le \min\left\{2R\sqrt{\kappa}\nu_{t-1}^{(1)}\|g(x_t;\theta_0)/\sqrt{m}\|_{V_{t-1}^{-1}} + 2\epsilon_{3,t-1}, 1\right\}$$
$$\le \min\left\{2R\sqrt{\kappa}\nu_{t-1}^{(1)}\|g(x_t;\theta_0)/\sqrt{m}\|_{V_{t-1}^{-1}}, 1\right\} + 2\epsilon_{3,t-1}$$
$$\le 2R\sqrt{\kappa}\nu_{t-1}^{(1)}\min\left\{\|g(x_t;\theta_0)/\sqrt{m}\|_{V_{t-1}^{-1}}, 1\right\} + 2\epsilon_{3,t-1}$$
$$\le 2R\sqrt{\kappa}\nu_T^{(1)}\min\left\{\|g(x_t;\theta_0)/\sqrt{m}\|_{V_{t-1}^{-1}}, 1\right\} + 2\epsilon_{3,T}.$$

Here, the first inequality follows from $0 \le |\mu(\cdot) - \mu(\cdot)| \le 1$, the second from the bound $\min\{a + b, 1\} \le \min\{a, 1\} + b$ for $b > 0$, the third from the facts that $2R\sqrt{\kappa} \ge 1$ and $\nu_t^{(1)} \ge 1$ for all $t$, thereby using $\min\{ab, 1\} \le a\min\{b, 1\}$ if $a \ge 1$, and the last inequality follows from the fact that both $\nu_t$ and $\epsilon_{3,t}$ are monotonically non-decreasing in $t$.

Now, we can proceed as

$$\text{Regret}(T) = \sum_{t=1}^{T} \mu(h(x_t^*)) - \mu(h(x_t))$$
$$\le 2R\sqrt{\kappa}\nu_T^{(1)}\sum_{t=1}^{T}\min\left\{\|g(x_t;\theta_0)/\sqrt{m}\|_{V_{t-1}^{-1}}, 1\right\} + 2T\epsilon_{3,T},$$

where we can see that by the condition of $m$ in Condition C.2,

$$T\epsilon_{3,T} = C_3 Rm^{-1/6}\sqrt{\log m}L^3 T^{5/3}\lambda_0^{-2/3} \le 1,$$

plugging this back gives,

$$\text{Regret}(T) \leq 2R\sqrt{\kappa}\nu_T^{(1)} \sum_{t=1}^T \min\left\{\|g(x_t;\theta_0)/\sqrt{m}\|_{V_{t-1}^{-1}}, 1\right\} + 1$$

$$\leq 2R\sqrt{\kappa}\nu_T^{(1)} \sqrt{T \sum_{i=1}^T \min\left\{\|g(x_t;\theta_0)/\sqrt{m}\|_{\widetilde{V}_{t-1}^{-1}}^2, 1\right\}} + 1$$

$$\leq 2R\sqrt{\kappa}\nu_T^{(1)} \sqrt{2T \log \det\left(\sum_{t=1}^T \frac{1}{\kappa m \lambda_0} g(x_t;\theta_0)g(x_t;\theta_0)^\top + \mathbf{I}\right)} + 1$$

$$\leq 2R\sqrt{\kappa}\nu_T^{(1)} \sqrt{2T\widetilde{d}} + 1,$$

where the second inequality follows from the Cauchy–Schwarz inequality and the relation $V_{t-1} \succeq \widetilde{V}_{t-1}$, the third follows from Lemma H.2, and the final inequality follows from the definition of $\widetilde{\widetilde{d}}$. Notice that

$$\iota_T = 16\sqrt{\log\det\left(\sum_{t=1}^T \frac{1}{4m\lambda_0}g(x_t;\theta_0)g(x_t;\theta_0)^\top + \mathbf{I}\right)\log\frac{4T^2}{\delta}} + 8C_1\sqrt{\frac{L}{\lambda_0}}\log\frac{4T^2}{\delta}$$

$$\leq 16\sqrt{\widetilde{d}\log(4T^2/\delta)} + \sqrt{4C_1(2L)^{1/2}S\log^{-1}(4/\delta)}\log(4T^2/\delta),$$

where the last inequality follows from the definition of $\widetilde{d}$ and the initialization rule of $\lambda_0$, which gives $\nu_T^{(1)} = \widetilde{\mathcal{O}}(S^2\sqrt{\widetilde{d}} + S^{2.5})$. Finally, plugging $\nu_T^{(1)}$ in gives,

$$\text{Regret}(T) = \widetilde{\mathcal{O}}\left(S^2\widetilde{d}\sqrt{\kappa T} + S^{2.5}\sqrt{\kappa\widetilde{d}T}\right),$$

finishing the proof.

### G.2 PROOF OF THEOREM 5.2

First, for each $t \in \mathbb{N}$, define the set of timesteps

$$\mathcal{T}_1(t) = \left\{t' \in [t] : \left|f(x_{t'};\theta_{t'}) - g(x_{t'};\theta_0)^\top(\theta^* - \theta_0))\right| \geq 1\right\}. \tag{48}$$

This set contains exactly those timesteps where $\theta_{t'}$ lies outside the parameter set (when $\|\theta_{t'} - \theta_0\|_2 > S$). Based on this, we form a pruned design matrix by removing the corresponding feature vectors while preserving their original order. In particular, for the regularized covariance matrix $V_t$, we obtain

$$\underline{V}_t = \sum_{i=1}^t \frac{1}{m}\mathbb{1}\{i \notin \mathcal{T}_1\}g(x_i;\theta_0)g(x_i;\theta_0)^\top + \lambda_t\mathbf{I} = \sum_{i=1}^{t-|\mathcal{T}_1(t)|} \frac{1}{m}g(x_{\tau(i)};\theta_0)g(x_{\tau(i)};\theta_0)^\top + \lambda_t\mathbf{I}.$$

Here, $\tau : \{1,\ldots,t-|\mathcal{T}_1(t)|\} \to \{1,\ldots,t\}$ maps each $j$ to the $j$-th smallest element of $[t] \setminus \mathcal{T}_1(t)$. Similarly, we define $H_t(\theta)$ and $W_t$ as:

$$\underline{H}_t(\theta) = \sum_{i=1}^{t-|\mathcal{T}_1(t)|} \frac{\dot{\mu}(g(x_{\tau(i)};\theta_0)^\top(\theta - \theta_0))}{m}g(x_{\tau(i)};\theta_0)g(x_{\tau(i)};\theta_0)^\top + \lambda_t\mathbf{I},$$

$$\underline{W}_t = \sum_{i=1}^{t-|\mathcal{T}_1(t)|} \frac{\dot{\mu}(f(x_{\tau(i)};\theta_{\tau(i)}))}{m}g(x_{\tau(i)};\theta_0)g(x_{\tau(i)};\theta_0)^\top + \lambda_t\mathbf{I}.$$

Same way as before, we will denote $\underline{\widetilde{V}}_t, \underline{\widetilde{H}}_t(\theta), \underline{\widetilde{W}}_t$ as the design matrix where the regularization parameter $\lambda_t$ is replaced to $\lambda_0$.

Using our new design matrices and the self-concordant property of the logistic function (see Lemma H.3; cf. Lemma 9 of Faury et al. (2020), Lemma 7 of Abeille et al. (2021), and Lemma 5 of Jun et al. (2021)), we can show that the true-variance design matrix $\underline{H}(\theta^*)$ is bounded by the empirical-variance design matrix $\underline{W}_t$.

**Proposition G.2.** *We have* $3\underline{H}_t(\theta^*) \succeq \underline{W}_t \succeq \frac{1}{3}\underline{H}_t(\theta^*)$.

Next, we define three additional sets of timesteps derived from $\mathcal{T}_1$:

$$\mathcal{T}_2 = \left\{ t \in [T - |\mathcal{T}_1(T)|] : \left| g(x_{\tau(t)}; \theta_0)^\top (\widetilde{\theta}_{\tau(t)-1} - \theta^*) \right| \geq 1 \right\},$$

$$\mathcal{T}_3 = \left\{ t \in [T - |\mathcal{T}_1(T)|] : \left\| g(x_{\tau(t)}; \theta_0)/\sqrt{m} \right\|_{\widetilde{V}_{\tau(t-1)}^{-1}} \geq 1 \right\},$$

$$\mathcal{T}_4 = \left\{ t \in [T - |\mathcal{T}_1(T)|] : \left\| \sqrt{\dot\mu(f(x_{\tau(t)}; \theta_{\tau(t)}))} g(x_{\tau(t)}; \theta_0)/\sqrt{m} \right\|_{\widetilde{W}_{\tau(t-1)}^{-1}} \geq 1 \right\}. \quad (49)$$

We define $\mathcal{T}_2$ to measure the distance between $g(x_{\tau(t)}; \theta_0)^\top \widetilde{\theta}_{\tau(t)-1}$ and $h(x_{\tau(t)})$ and control the estimation error of the neural network. We introduce $\mathcal{T}_3, \mathcal{T}_4$ to control the value of $\| g(x_{\tau(t)}; \theta_0)/\sqrt{m} \|_{\widetilde{V}_{\tau(t-1)}^{-1}}$ and $\| \sqrt{\dot\mu(f(x_{\tau(t)}; \theta_{\tau(t)}))} g(x_{\tau(t)}; \theta_0)/\sqrt{m} \|_{\widetilde{W}_{\tau(t-1)}^{-1}}$ in order to apply the elliptical potential lemma (Lemma H.2).

Next, we introduce two propositions to bound the cardinality of $\mathcal{T}_1(T)$, $\mathcal{T}_2$, $\mathcal{T}_3$ and $\mathcal{T}_4$:

**Proposition G.3.** *We have* $|\mathcal{T}_1(T)| \leq 4\kappa \widetilde{d} {\nu_T^{(1)}}^2 + 1$ *and* $|\mathcal{T}_2| \leq 24\kappa \widetilde{d} {\nu_T^{(2)}}^2$, *where* $\nu_t^{(1)}$ *and* $\nu_t^{(2)}$ *are defined at Equations* (2) *and* (5), *respectively.*

**Proposition G.4.** *We have* $|\mathcal{T}_3|, |\mathcal{T}_4| \leq 2\widetilde{d}$.

For Proposition G.3, we use the concentration inequalities between $\theta_{\tau(t)}$ and $\theta^*$, and $\widetilde{\theta}_{\tau(t)-1}$ and $\theta^*$ using Lemmas 4.5 and 5.1. For Proposition G.4 we modified previous results appropriate to our setting called the elliptical potential count lemma (Lemma 7 of Gales et al. (2022), Lemma 4 of Kim et al. (2022)).

Now we can start the proof of Theorem 5.2.

*Proof of Theorem 5.2.* At time $t$, from the optimistic rule in Equation (6), denote

$$(x_t, \widetilde{\theta}_{t-1}) \leftarrow \underset{x \in \mathcal{X}_t, \theta \in \mathcal{W}_{t-1}}{\arg\max} \langle g(x; \theta_0), \theta - \theta_0 \rangle \quad (50)$$

We use $\mathcal{T}_1(T) = \mathcal{T}_1$ for brevity. From Equations (48) and (49), we define the combined set of timesteps as

$$\mathcal{T} = \{\mathcal{T}_2 \cup \mathcal{T}_3 \cup \mathcal{T}_4\}.$$

Then we have,

$$\text{Regret}(T) \leq |\mathcal{T}_1| + \sum_{t=1}^{T-|\mathcal{T}_1|} \mu(h(x_{\tau(t)}^*)) - \mu(h(x_{\tau(t)}))$$

$$\leq |\mathcal{T}_1| + |\mathcal{T}_2| + |\mathcal{T}_3| + |\mathcal{T}_4| + \sum_{t=1}^{T-|\mathcal{T}_1|} \mathbb{1}\{t \notin \mathcal{T}\} \left[ \mu(h(x_{\tau(t)}^*)) - \mu(h(x_{\tau(t)})) \right]$$

$$\leq 4\kappa \widetilde{d} {\nu_T^{(1)}}^2 + 24\kappa \widetilde{d} {\nu_T^{(2)}}^2 + 4\widetilde{d} + 1 + \underbrace{\sum_{t=1}^{T-|\mathcal{T}_1|} \mathbb{1}\{t \notin \mathcal{T}\} \left[ \mu(h(x_{\tau(t)}^*)) - \mu(h(x_{\tau(t)})) \right]}_{=:\text{Regret}^c(T)},$$

$$(51)$$

where the second inequality follows from the definition of $\mathcal{T}$, and the last inequality follows from Propositions G.3 and G.4. For $\text{Regret}^c(T)$, we have

$$\text{Regret}^c(T) = \sum_{t=1}^{T-|\mathcal{T}_1|} \mathbb{1}\{t \notin \mathcal{T}\} \left[ \mu(g(x_{\tau(t)}^*; \theta_0)^\top (\theta^* - \theta_0)) - \mu(g(x_{\tau(t)}; \theta_0)^\top (\theta^* - \theta_0)) \right]$$

$$\leq \sum_{t=1}^{T-|\mathcal{T}_1|} \mathbb{1}\{t \notin \mathcal{T}\} \left[ \mu(g(x_{\tau(t)}; \theta_0)^\top (\widetilde{\theta}_{\tau(t)-1} - \theta_0)) - \mu(g(x_{\tau(t)}; \theta_0)^\top (\theta^* - \theta_0)) \right]$$

where the equality follows from Lemma 6.1, and the inequality follows from the optimistic rule in Equation (50) since $\widetilde{\theta}_{\tau(t)-1}, \theta^* \in \mathcal{W}_{\tau(t)-1}$. With the definition of $\alpha(x, \theta', \theta'')$ at Equation (32), we can continue with

$$\text{Regret}^c(T) \leq \underbrace{\sum_{t=1}^{T-|\mathcal{T}_1|} \mathbb{1}\{t \notin \mathcal{T}\}\big[\dot{\mu}(g(x_{\tau(t)}; \theta_0)^\top(\theta^* - \theta_0))g(x_{\tau(t)}; \theta_0)^\top(\widetilde{\theta}_{\tau(t)-1} - \theta^*)\big]}_{\textbf{(term 1)}}$$

$$+ \underbrace{\sum_{t=1}^{T-|\mathcal{T}_1|} \mathbb{1}\{t \notin \mathcal{T}\}\big[\alpha(x_{\tau(t)}, \widetilde{\theta}_{\tau(t)-1}, \theta^*)[g(x_{\tau(t)}; \theta_0)^\top(\widetilde{\theta}_{\tau(t)-1} - \theta^*)]^2\big]}_{\textbf{(term 2)}}, \quad (52)$$

where we used a second-order Taylor expansion and the fact that $|\ddot{\mu}| \leq \dot{\mu}$.

For **(term 2)** we have

$$\textbf{(term 2)} \leq \sum_{t=1}^{T-|\mathcal{T}_1|} \mathbb{1}\{t \notin \mathcal{T}\}\big[1 \cdot [g(x_{\tau(t)}; \theta_0)^\top(\widetilde{\theta}_{\tau(t)-1} - \theta^*)]^2\big]$$

$$\leq \sum_{t=1}^{T-|\mathcal{T}_1|} \mathbb{1}\{t \notin \mathcal{T}\} \cdot \|g(x_{\tau(t)}; \theta_0)/\sqrt{m}\|_{\underline{W}_{\tau(t-1)}^{-1}}^2 \cdot m\|\widetilde{\theta}_{\tau(t)-1} - \theta^*\|_{\underline{W}_{\tau(t-1)}}^2. \quad (53)$$

For $\|g(x_{\tau(t)}; \theta_0)/\sqrt{m}\|_{\underline{W}_{\tau(t-1)}^{-1}}$, we have

$$\|g(x_{\tau(t)}; \theta_0)/\sqrt{m}\|_{\underline{W}_{\tau(t-1)}^{-1}} \leq \sqrt{3}\|g(x_{\tau(t)}; \theta_0)/\sqrt{m}\|_{\underline{H}_{\tau(t-1)}^{-1}(\theta^*)}$$

$$\leq \sqrt{3\kappa}\|g(x_{\tau(t)}; \theta_0)/\sqrt{m}\|_{\underline{V}_{\tau(t-1)}^{-1}}, \quad (54)$$

where the first inequality follows from Proposition G.2 and the second inequality follows from Assumption 6.3. For $\sqrt{m}\|\widetilde{\theta}_{\tau(t)-1} - \theta^*\|_{\underline{W}_{\tau(t-1)}}$, we have

$$\sqrt{m}\|\widetilde{\theta}_{\tau(t)-1} - \theta^*\|_{\underline{W}_{\tau(t-1)}}$$

$$\leq \sqrt{m}\|\widetilde{\theta}_{\tau(t)-1} - \theta^*\|_{\underline{W}_{\tau(t)-1}}$$

$$\leq \sqrt{m}\|\widetilde{\theta}_{\tau(t)-1} - \theta^*\|_{W_{\tau(t)-1}}$$

$$\leq \left(\sqrt{m}\|\widetilde{\theta}_{\tau(t)-1} - \theta_{\tau(t)-1}\|_{W_{\tau(t)-1}} + \sqrt{m}\|\theta_{\tau(t)-1} - \theta^*\|_{W_{\tau(t)-1}}\right)$$

$$\leq 2\nu_{\tau(t)-1}^{(2)}, \quad (55)$$

where the first and the second inequality are due to the fact that $\tau(t-1) \leq \tau(t)-1$ and $\underline{W}_t \preceq W_t$ for all $t$, respectively. The third inequality follows from the triangle inequality, and the last inequality follows from Lemma 5.1 since $\widetilde{\theta}_{\tau(t)-1}, \theta^* \in \mathcal{W}_{\tau(t)-1}$. Plugging Equations (54) and (55) back to **(term 2)** gives

$$\textbf{(term 2)} \leq \sum_{t=1}^{T-|\mathcal{T}_1|} \mathbb{1}\{t \notin \mathcal{T}\} \cdot 3\kappa\|g(x_{\tau(t)}; \theta_0)/\sqrt{m}\|_{\underline{V}_{\tau(t-1)}^{-1}}^2 \cdot 4\big(\nu_{\tau(t)-1}^{(2)}\big)^2$$

$$\leq 12\kappa\big(\nu_T^{(2)}\big)^2 \sum_{t=1}^{T-|\mathcal{T}_1|} \mathbb{1}\{t \notin \mathcal{T}\} \cdot \|g(x_{\tau(t)}; \theta_0)/\sqrt{m}\|_{\underline{V}_{\tau(t-1)}^{-1}}^2,$$

where the inequality holds since $\nu_t^{(t)}$ is monotonically non-decreasing in $t$. By the definition of $\mathcal{T}_3$, we have $\|g(x_{\tau(t)}; \theta_0)/\sqrt{m}\|_{\underline{V}_{\tau(t-1)}^{-1}} < 1$ for all $t \in [T - |\mathcal{T}_1|]$. Therefore,

$$\mathbb{1}\{t \notin \mathcal{T}\} \cdot \|g(x_{\tau(t)}; \theta_0)/\sqrt{m}\|_{\underline{V}_{\tau(t-1)}^{-1}}^2 = \min\left\{1, \mathbb{1}\{t \notin \mathcal{T}\} \cdot \|g(x_{\tau(t)}; \theta_0)/\sqrt{m}\|_{\underline{V}_{\tau(t-1)}^{-1}}^2\right\}$$

$$\leq \min\left\{1, \|g(x_{\tau(t)}; \theta_0)/\sqrt{m}\|_{\underline{V}_{\tau(t-1)}^{-1}}^2\right\}$$

$$\leq \min\left\{1, \|g(x_{\tau(t)}; \theta_0)/\sqrt{m}\|_{\widehat{\underline{V}}_{\tau(t-1)}^{-1}}^2\right\}, \quad (56)$$

where the last inequality follows from the fact that $\lambda_0 \leq \lambda_t$. Substituting Equation (56) gives

$$\textbf{(term 2)} \leq 12\kappa\big(\nu_T^{(2)}\big)^2 \sum_{t=1}^{T-|\mathcal{T}_1|} \min\left\{1, \|g(x_{\tau(t)};\theta_0)/\sqrt{m}\|_{\underline{\widetilde{V}}_{\tau(t-1)}^{-1}}^2\right\}$$

$$\leq 24\kappa\big(\nu_T^{(2)}\big)^2 \log \frac{\det \underline{\widetilde{V}}_{\tau(T-|\mathcal{T}_1|)}}{\det \kappa\lambda_0 \mathbf{I}}$$

$$\leq 24\kappa\big(\nu_T^{(2)}\big)^2 \log\det\left(\sum_{t=1}^T \frac{1}{\kappa m \lambda_0} g(x_t;\theta_0)g(x_t;\theta_0)^\top + \mathbf{I}\right)$$

$$\leq 24\kappa\big(\nu_T^{(2)}\big)^2 \widetilde{d}, \tag{57}$$

where the second inequality follows from Lemma H.2, and the last inequality follows from the definition of $\widetilde{d}$.

For **(term 1)**, we consider 2 cases where:

**(case 1).** if $\dot{\mu}(h(x_{\tau(t)})) \leq \dot{\mu}(f(x_{\tau(t)};\theta_{\tau(t)}))$

**(case 2).** if $\dot{\mu}(h(x_{\tau(t)})) > \dot{\mu}(f(x_{\tau(t)};\theta_{\tau(t)}))$

In (case 1), for **(term 1)**, we continue with

$$\sum_{t=1}^{T-|\mathcal{T}_1|} \mathbb{1}\{t \notin \mathcal{T}\}\big[\dot{\mu}(g(x_{\tau(t)};\theta_0)^\top(\theta^* - \theta_0))g(x_{\tau(t)};\theta_0)^\top(\widetilde{\theta}_{\tau(t)-1} - \theta^*)\big]$$

$$= \sum_{t=1}^{T-|\mathcal{T}_1|} \mathbb{1}\{t \notin \mathcal{T}\} \cdot \sqrt{\dot{\mu}(h(x_{\tau(t)}))}\sqrt{\dot{\mu}(h(x_{\tau(t)}))}g(x_{\tau(t)};\theta_0)^\top(\widetilde{\theta}_{\tau(t)-1} - \theta^*)$$

$$\leq \sum_{t=1}^{T-|\mathcal{T}_1|} \mathbb{1}\{t \notin \mathcal{T}\} \cdot \sqrt{\dot{\mu}(h(x_{\tau(t)}))}\sqrt{\dot{\mu}(f(x_{\tau(t)};\theta_{\tau(t)}))}g(x_{\tau(t)};\theta_0)^\top(\widetilde{\theta}_{\tau(t)-1} - \theta^*), \tag{58}$$

where the last inequality follows from the assumption of (case 1). For brevity, we denote $\dot{g}(x_t;\theta_0) = \sqrt{\dot{\mu}(f(x_t;\theta_t))}g(x_t;\theta_0)$. Notice that we can represent $W_t$ as $W_t = \sum_{i=1}^t \frac{1}{m}\dot{g}(x_t;\theta_0)\dot{g}(x_t;\theta_0)^\top + \lambda_t\mathbf{I}$. Then we can continue as

$$\textbf{(term 1)} \leq \sum_{t=1}^{T-|\mathcal{T}_1|} \mathbb{1}\{t \notin \mathcal{T}\} \cdot \sqrt{\dot{\mu}(h(x_{\tau(t)}))} \cdot \dot{g}(x_{\tau(t)};\theta_0)^\top(\widetilde{\theta}_{\tau(t)-1} - \theta^*)$$

$$\leq \sum_{t=1}^{T-|\mathcal{T}_1|} \mathbb{1}\{t \notin \mathcal{T}\} \cdot \sqrt{\dot{\mu}(h(x_{\tau(t)}))} \cdot \|\dot{g}(x_{\tau(t)};\theta_0)/\sqrt{m}\|_{\underline{W}_{\tau(t-1)}^{-1}} \cdot \sqrt{m}\|\widetilde{\theta}_{\tau(t)-1} - \theta^*\|_{\underline{W}_{\tau(t-1)}}$$

For $\mathbb{1}\{t \notin \mathcal{T}\}\|\dot{g}(x_{\tau(t)};\theta_0)\|_{\underline{W}_{\tau(t-1)}^{-1}}$, we have

$$\mathbb{1}\{t \notin \mathcal{T}\} \cdot \|\dot{g}(x_{\tau(t)};\theta_0)\|_{\underline{W}_{\tau(t-1)}^{-1}} = \min\left\{1, \mathbb{1}\{t \notin \mathcal{T}\} \cdot \|\dot{g}(x_{\tau(t)};\theta_0)/\sqrt{m}\|_{\underline{W}_{\tau(t-1)}^{-1}}\right\}$$

$$\leq \min\left\{1, \|\dot{g}(x_{\tau(t)};\theta_0)/\sqrt{m}\|_{\widetilde{\underline{W}}_{\tau(t-1)}^{-1}}\right\}, \tag{59}$$

where the inequality follows from the definition of $\mathcal{T}_4$ and the fact that $\lambda_0 \leq \lambda_t$ for all $t$. Also, using the previous results of Equation (55), we have $\sqrt{m}\|\widetilde{\theta}_{\tau(t)-1} - \theta^*\|_{\underline{W}_{\tau(t-1)}} \leq 2\nu_{\tau(t)-1}^{(2)}$. Substituting these back gives

$$\textbf{(term 1)} \leq 2\nu_T^{(2)} \sum_{t=1}^{T-|\mathcal{T}_1|} \sqrt{\dot{\mu}(h(x_{\tau(t)}))} \cdot \left\{1, \|\dot{g}(x_{\tau(t)};\theta_0)/\sqrt{m}\|_{\widetilde{\underline{W}}_{\tau(t-1)}^{-1}}\right\}$$

$$\leq 2\nu_T^{(2)} \underbrace{\sqrt{\sum_{t=1}^{T-|\mathcal{T}_1|} \dot{\mu}(h(x_{\tau(t)}))}}_{\textbf{(term 3)}} \cdot \underbrace{\sqrt{\sum_{t=1}^{T-|\mathcal{T}_1|} \left\{1, \|\dot{g}(x_{\tau(t)};\theta_0)/\sqrt{m}\|_{\widetilde{\underline{W}}_{\tau(t-1)}^{-1}}^2\right\}}}_{\textbf{(term 4)}}$$

where the first inequality is by the monotonicity of $\mu_t^{(2)}$ in $t$, and the second inequality follows from the Cauchy-Schwarz inequality. For **(term 4)**, we have

$$\sqrt{\sum_{t=1}^{T-|\mathcal{T}_1|} \left\{ 1, \|\dot{g}(x_{\tau(t)}; \theta_0)/\sqrt{m}\|_{\widetilde{W}_{\tau(t-1)}^{-1}}^2 \right\}} \leq \sqrt{2 \log \frac{\det \widetilde{W}_{\tau(T-|\mathcal{T}_1|)}}{\det \lambda_0 \mathbf{I}}}$$

$$\leq \sqrt{2 \log \det \Big( \sum_{t=1}^{T} \frac{\dot{\mu}(f(x_t; \theta_t))}{m \lambda_0} g(x_t; \theta_0) g(x_t; \theta_0)^\top + \mathbf{I} \Big)}$$

$$\leq \sqrt{2\widetilde{d}},$$

where the first inequality follows from Lemma H.2, and the last inequality follows from the definition of $\widetilde{d}$.

For **(term 3)**, we have

$$\begin{aligned}
(\textbf{term 3})^2 &\leq \sum_{t=1}^{T} \dot{\mu}(g(x_t; \theta_0)^\top (\theta^* - \theta_0)) \\
&\leq \sum_{t=1}^{T} \dot{\mu}(g(x_t^*; \theta_0)^\top (\theta^* - \theta_0)) + \sum_{t=1}^{T} \alpha(x_t, x_t^*, \theta^*)(g(x_t; \theta_0) - g(x_t^*; \theta_0))^\top (\theta^* - \theta_0) \\
&= \frac{T}{\kappa^*} + \sum_{t=1}^{T} \alpha(x_t, x_t^*, \theta^*)(g(x_t; \theta_0) - g(x_t^*; \theta_0))^\top (\theta^* - \theta_0) \\
&\leq \frac{T}{\kappa^*} + \sum_{t=1}^{T} \alpha(x_t, x_t^*, \theta^*)(g(x_t^*; \theta_0) - g(x_t; \theta_0))^\top (\theta^* - \theta_0) \\
&= \frac{T}{\kappa^*} + \sum_{t=1}^{T} \mu(g(x_t^*; \theta_0)^\top (\theta^* - \theta_0)) - \mu(g(x_t; \theta_0)^\top (\theta^* - \theta_0)) \\
&= \frac{T}{\kappa^*} + \sum_{t=1}^{T} \mu(h(x_t^*)) - \mu(h(x_t)) \\
&= \frac{T}{\kappa^*} + \text{Regret}(T).
\end{aligned} \tag{60}$$

Here, the second inequality follows from a first-order Taylor expansion together with the bound $|\ddot{\mu}| \leq \dot{\mu}$ and the definition of $\alpha(x', x'', \theta)$ in Equation (32), the first equality follows from the definition of $\kappa^*$, namely $1/\kappa^* = \frac{1}{T} \sum_{t=1}^{T} \dot{\mu}(h(x_t^*))$, the third inequality uses the fact that $h(x_t^*) \geq h(x_t)$, the second equality follows from the mean-value theorem, and the final equality follows from the definition of regret.

Finally, substituting **(term 3)** and **(term 4)** back gives

$$(\textbf{term 1}) \leq 2\nu_T^{(2)} \sqrt{\text{Regret}(T) + T/\kappa^*} \cdot \sqrt{2\widetilde{d}}. \tag{61}$$

Now we consider about (case 2), where $\dot{\mu}(h(x_{\tau(t)})) > \dot{\mu}(f(x_{\tau(t)}; \theta_{\tau(t)}))$. For **(term 1)**, we have

$$\sum_{t=1}^{T-|\mathcal{T}_1|} \mathbb{1}\{t \notin \mathcal{T}\}\big[\dot\mu(g(x_{\tau(t)};\theta_0)^\top(\theta^* - \theta_0))g(x_{\tau(t)};\theta_0)^\top(\widetilde\theta_{\tau(t)-1} - \theta^*)\big]$$

$$\leq \underbrace{\sum_{t=1}^{T-|\mathcal{T}_1|} \mathbb{1}\{t \notin \mathcal{T}\} \cdot \dot\mu(g(x_{\tau(t)};\theta_0)^\top(\theta_{\tau(t)} - \theta_0))g(x_{\tau(t)};\theta_0)^\top(\widetilde\theta_{\tau(t)-1} - \theta^*)}_{\textbf{(term 4)}}$$

$$+ \underbrace{\sum_{t=1}^{T-|\mathcal{T}_1|} \mathbb{1}\{t \notin \mathcal{T}\} \cdot 1 \cdot [g(x_{\tau(t)};\theta_0)^\top(\theta_{\tau(t)} - \theta^*)g(x_{\tau(t)};\theta_0)^\top(\widetilde\theta_{\tau(t)-1} - \theta^*)]}_{\textbf{(term 5)}},$$

where the inequality follows from the Taylor expansion, and by the fact that $|\ddot\mu| \leq \dot\mu \leq 1$. For **(term 5)** we have,

$$\sum_{t=1}^{T-|\mathcal{T}_1|} \mathbb{1}\{t \notin \mathcal{T}\} \cdot 1 \cdot [g(x_{\tau(t)};\theta_0)^\top(\theta_{\tau(t)} - \theta^*)g(x_{\tau(t)};\theta_0)^\top(\widetilde\theta_{\tau(t)-1} - \theta^*)]$$

$$\leq \sum_{t=1}^{T-|\mathcal{T}_1|} \mathbb{1}\{t \notin \mathcal{T}\} \cdot \|g(x_{\tau(t)};\theta_0)/\sqrt{m}\|^2_{\underline W_{\tau(t-1)}^{-1}}$$

$$\times \sqrt{m}\|\theta_{\tau(t)} - \theta^*\|_{\underline W_{\tau(t-1)}} \times \sqrt{m}\|\widetilde\theta_{\tau(t)-1} - \theta^*\|_{\underline W_{\tau(t-1)}}.$$

We have $\mathbb{1}\{t \notin \mathcal{T}\} \cdot \|g(x_{\tau(t)};\theta_0)/\sqrt{m}\|_{\underline W_{\tau(t-1)}^{-1}} \leq 3\kappa \min\{1, \|g(x_{\tau(t)};\theta_0)/\sqrt{m}\|^2_{\underline{\widetilde V}_{\tau(t-1)}^{-1}}\}$ using

Equations (54) and (56). Also we have $\sqrt{m}\|\widetilde\theta_{\tau(t)-1} - \theta^*\|_{\underline W_{\tau(t-1)}} \leq 2\nu^{(2)}_{\tau(t)-1}$ using Equation (55).

For $\sqrt{m}\|\theta_{\tau(t)} - \theta^*\|_{\underline W_{\tau(t-1)}}$, we have

$$\sqrt{m}\|\theta_{\tau(t)} - \theta^*\|_{\underline W_{\tau(t-1)}} \leq \sqrt{m}\|\theta_{\tau(t)} - \theta^*\|_{\underline W_{\tau(t)}} \leq \sqrt{m}\|\theta_{\tau(t)} - \theta^*\|_{W_{\tau(t)}} \leq \nu^{(2)}_{\tau(t)}.$$

Plugging results back gives

$$\textbf{(term 5)} \leq \sum_{t=1}^{T-|\mathcal{T}_1|} 3\kappa \min\left\{1, \|g(x_{\tau(t)};\theta_0)/\sqrt{m}\|^2_{\underline{\widetilde V}_{\tau(t-1)}^{-1}}\right\} \cdot \nu^{(2)}_{\tau(t)} \cdot 2\nu^{(2)}_{\tau(t)-1}$$

$$\leq 6\kappa\big(\nu^{(2)}_T\big)^2 \sum_{t=1}^{T-|\mathcal{T}_1|} \min\left\{1, \|g(x_{\tau(t)};\theta_0)/\sqrt{m}\|^2_{\underline{\widetilde V}_{\tau(t-1)}^{-1}}\right\}$$

$$\leq 12\kappa\big(\nu^{(2)}_T\big)^2 \log\frac{\det \underline{\widetilde V}_{\tau(T-|\mathcal{T}_1|)}}{\det \kappa\lambda_0\mathbf{I}}$$

$$\leq 12\kappa\widetilde d\big(\nu^{(2)}_T\big)^2$$

where the second inequality is because $\nu^{(2)}_{\tau(t)}$ is non-decreasing in $t$, the third inequality follows from Lemma H.2, and the last inequality follows from the definition of $\widetilde d$.

Now, for **(term 4)**, we have

$$\sum_{t=1}^{T-|\mathcal{T}_1|} \mathbb{1}\{t \notin \mathcal{T}\} \cdot \dot\mu(g(x_{\tau(t)};\theta_0)^\top(\theta_{\tau(t)} - \theta_0))g(x_{\tau(t)};\theta_0)^\top(\widetilde\theta_{\tau(t)-1} - \theta^*)$$

$$= \underbrace{\sum_{t=1}^{T-|\mathcal{T}_1|} \mathbb{1}\{t \notin \mathcal{T}\} \cdot \dot\mu(f(x_{\tau(t)};\theta_{\tau(t)}))g(x_{\tau(t)};\theta_0)^\top(\widetilde\theta_{\tau(t)-1} - \theta^*)}_{\textbf{(term 6)}}$$

$$+ \underbrace{\sum_{t=1}^{T-|\mathcal{T}_1|} \mathbb{1}\{t \notin \mathcal{T}\} \cdot \Big(\dot\mu(g(x_{\tau(t)};\theta_0)^\top(\theta_{\tau(t)} - \theta_0)) - \dot\mu(f(x_{\tau(t)};\theta_{\tau(t)}))\Big)g(x_{\tau(t)};\theta_0)^\top(\widetilde\theta_{\tau(t)-1} - \theta^*)}_{\textbf{(term 7)}}.$$

For **(term 7)**, recall the definition of $\mathcal{T}_2$. Then for some absolute constant $C_3 > 0$, we have

$$
\begin{aligned}
\textbf{(term 7)} &\leq \sum_{t=1}^{T-|\mathcal{T}_1|} \left| \dot{\mu}(g(x_{\tau(t)}; \theta_0)^\top (\theta_{\tau(t)} - \theta_0)) - \dot{\mu}(f(x_{\tau(t)}; \theta_{\tau(t)})) \right| \cdot |g(x_{\tau(t)}; \theta_0)^\top (\widetilde{\theta}_{\tau(t)-1} - \theta^*)| \\
&\leq \sum_{t=1}^{T-|\mathcal{T}_1|} R \left| g(x_{\tau(t)}; \theta_0)^\top (\theta_{\tau(t)} - \theta_0) - f(x_{\tau(t)}; \theta_{\tau(t)}) \right| \cdot 1 \\
&\leq T \cdot C_3 R m^{-1/6} \sqrt{\log m} L^3 T^{2/3} \lambda_0^{-2/3} \\
&\leq 1,
\end{aligned}
$$

where the second inequality follows from the definition of $\mathcal{T}_2$, the third inequality is due to the fact that $\mu(\cdot)$ is a $R$-Lipschitz function, the third inequality follows from Lemma C.5, and the last inequality follows from the condition of $m$ in Condition C.2.

For **(term 6)**, we have

$$
\begin{aligned}
&\sum_{t=1}^{T-|\mathcal{T}_1|} \mathbb{1}\{t \notin \mathcal{T}\} \cdot \dot{\mu}(f(x_{\tau(t)}; \theta_{\tau(t)})) g(x_{\tau(t)}; \theta_0)^\top (\widetilde{\theta}_{\tau(t)-1} - \theta^*) \\
&\leq \sum_{t=1}^{T-|\mathcal{T}_1|} \mathbb{1}\{t \notin \mathcal{T}\} \cdot \sqrt{\dot{\mu}(h(x_{\tau(t)}))} \sqrt{\dot{\mu}(f(x_{\tau(t)}; \theta_{\tau(t)}))} g(x_{\tau(t)}; \theta_0)^\top (\widetilde{\theta}_{\tau(t)-1} - \theta^*),
\end{aligned}
$$

where the inequality follows from the assumption of (case 2). Notice that expression is same as the **(term 1)** of (case 1) at Equation (58). Therefore, using the result of Equation (61), we have **(term 6)** $\leq 2\nu_T^{(2)} \sqrt{\text{Regret}(T) + T/\kappa^*} \cdot \sqrt{2\widetilde{d}}$.

Finally, plugging **(term 4-7)** into **(term 1)** gives,

$$
\textbf{(term 1)} \leq 2\nu_T^{(2)} \sqrt{\text{Regret}(T) + T/\kappa^*} \cdot \sqrt{2\widetilde{d}} + 12\kappa \widetilde{d}(\nu_T^{(2)})^2 + 1.
$$

Recall the upper bound of **(term 1)** in (case 1) at Equation (61), which is **(term 1)** $\leq$ $2\nu_T^{(2)} \sqrt{\text{Regret}(T) + T/\kappa^*} \cdot \sqrt{2\widetilde{d}}$. Since the upper bound value in (case 2) is strictly larger than that of (case 1), we give a naive bound of **(term 1)** by using the result of (case 2).

Now, substituting **(term 1-2)** into Equation (52) gives

$$
\text{Regret}^c(T) \leq 2\nu_T^{(2)} \sqrt{\text{Regret}(T) + T/\kappa^*} \cdot \sqrt{2\widetilde{d}} + 36\kappa \widetilde{d}(\nu_T^{(2)})^2 + 1.
$$

Substituting $\text{Regret}^c(T)$ into Equation (51) gives

$$
\text{Regret}(T) \leq 2\nu_T^{(2)} \sqrt{\text{Regret}(T) + T/\kappa^*} \cdot \sqrt{2\widetilde{d}} + 4\widetilde{d} + 4\kappa \widetilde{d}(\nu_T^{(1)})^2 + 60\kappa \widetilde{d}(\nu_T^{(2)})^2 + 2.
$$

Finally, using the fact that for $b, c > 0$ and $x \in \mathbb{R}$, $x^2 - bx - c \leq 0 \implies x^2 \leq 2b^2 + 2c$, and substituting $\nu_T^{(1)}, \nu_T^{(2)} = \widetilde{\mathcal{O}}(S^2 \sqrt{\widetilde{d}} + S^{2.5})$, we have

$$
\begin{aligned}
\text{Regret}(T) &\leq 16(\nu_T^{(2)})^2 + 4\nu_T^{(2)} \sqrt{2\widetilde{d}T/\kappa^*} + 8\widetilde{d} + 8\kappa \widetilde{d}(\nu_T^{(1)})^2 + 120\kappa \widetilde{d}(\nu_T^{(2)})^2 + 4 \\
&\leq \widetilde{\mathcal{O}}\Big( S^2 \widetilde{d} \sqrt{T/\kappa^*} + S^{2.5} \widetilde{d}^{0.5} \sqrt{T/\kappa^*} + S^4 \kappa \widetilde{d}^2 + S^{4.5} \kappa \widetilde{d}^{1.5} + S^5 \kappa \widetilde{d} \Big),
\end{aligned}
$$

finishing the proof. $\qquad\square$

### G.3 Proof of Proposition G.2

We suitably modify Lemma 5 of Jun et al. (2021) for our setting. Define $d(t) = \left| f(x_t; \theta_t) - g(x_t; \theta_0)^\top (\theta^* - \theta_0)) \right|$. By the definition of $\mathcal{T}_1$, for all $t \notin \mathcal{T}_1(T)$, $d(t) \leq 1$. Recall

the definition of $\alpha(z', z'')$ at Equation (32). Then for all $t \notin \mathcal{T}_1(T)$, we have

$$
\begin{aligned}
\dot{\mu}(f(x_t; \theta_t)) &\geq \frac{d(t)}{\exp(d(t)) - 1} \cdot \alpha\Big(f(x_t; \theta_i), g(x_t; \theta_0)^\top (\theta^* - \theta_0)\Big) \\
&\geq \frac{d(t)}{\exp(d(t)) - 1} \cdot \frac{1 - \exp(-d(t))}{d(t)} \dot{\mu}(g(x_t; \theta_0)^\top (\theta^* - \theta_0)) \\
&= \frac{1}{\exp(d(t))} \cdot \mu(g(x_t; \theta_0)^\top (\theta^* - \theta_0)) \\
&\geq \frac{1}{d(t)^2 + d(t) + 1} \cdot \mu(g(x_t; \theta_0)^\top (\theta^* - \theta_0)) \\
&\geq \frac{1}{2d(t) + 1} \cdot \mu(g(x_t; \theta_0)^\top (\theta^* - \theta_0)),
\end{aligned}
$$

where the first and the second inequalities follow from the self-concordant property in Lemma H.3, the third and the fourth inequalities hold since $d(t) \leq 1$. This implies that

$$
\underline{W}_t \succeq \frac{1}{2 \max\{d(t')\}_{(t' \in [t]) \cap (t' \notin \mathcal{T}_1(t))} + 1} \underline{H}_t(\theta^*) \succeq \frac{1}{3} \underline{H}_t(\theta^*),
$$

In a similar way, we can have

$$
\underline{H}_t(\theta^*) \succeq \frac{1}{2 \max\{d(t')\}_{(t' \in [t]) \cap (t' \notin \mathcal{T}_1(t))} + 1} \underline{W}_t \succeq \frac{1}{3} \underline{W}_t.
$$

Combining these results, we finish the proof.

### G.4 PROOF OF PROPOSITION G.3

We start with the upper bound of $|\mathcal{T}_1|$. For an absolute constant $C_3 > 0$, we have:

$$
\begin{aligned}
|\mathcal{T}_1| \cdot \min\{1, 1^2\} &\leq \sum_{t=1}^{T} \min\left\{1, |f(x_t; \theta_t) - g(x_t; \theta_0)^\top (\theta^* - \theta_0)|^2\right\} \\
&\leq \sum_{t=1}^{T} \min\left\{1, 2|f(x_t; \theta_t) - g(x_t; \theta_0)^\top (\theta_t - \theta_0)|^2 + 2|g(x_t; \theta_0)^\top (\theta_t - \theta^*)|^2\right\},
\end{aligned}
$$

For $2|f(x_t; \theta_t) - g(x_t; \theta_0)^\top (\theta_t - \theta_0)|^2$, we have $|f(x_t; \theta_t) - g(x_t; \theta_0)^\top (\theta_t - \theta_0)| \leq C_3 m^{-1/6} \sqrt{\log m} L^3 t^{2/3} \lambda_t^{-2/3}$ using Lemma C.5. Since the error term is positive, we can take it out of the $\min\{1, \cdot\}$ term, which gives

$$
\begin{aligned}
|\mathcal{T}_1| &\leq \sum_{t=1}^{T} \min\left\{1, 2|g(x_t; \theta_0)^\top (\theta_t - \theta^*)|^2\right\} + C_3^2 m^{-1/3} (\log m) L^6 T^{7/3} \lambda_0^{-4/3} \\
&\leq 2 \sum_{t=1}^{T} \min\left\{1, |g(x_t; \theta_0)^\top (\theta_t - \theta^*)|^2\right\} + 1,
\end{aligned}
$$

where the last inequality is due to the condition of $m$ at Condition 4.4, and the fact that $\min\{1, ab\} \leq a \min\{1, b\}$ if $a \geq 1$. We further proceed as

$$
|\mathcal{T}_1| \leq 2 \sum_{t=1}^{T} \min\left\{1, \|g(x_t; \theta_0)/\sqrt{m}\|^2_{H_{t-1}^{-1}(\theta^*)} \cdot m\|\theta_t - \theta^*\|^2_{H_{t-1}(\theta^*)}\right\} + 1.
$$

For $m\|\theta_t - \theta^*\|^2_{H_{t-1}(\theta^*)}$, we have

$$
m\|\theta_t - \theta^*\|^2_{H_{t-1}(\theta^*)} \leq m\|\theta_t - \theta^*\|^2_{H_t(\theta^*)} \leq \left(\nu_t^{(1)}\right)^2.
$$

Since $\nu_t^{(1)} \geq 1$ we can take out of the $\min\{1, \cdot\}$ term, and by the monotonicity of $\nu_t^{(1)}$ in $t$, we have

$$|\mathcal{T}_1| \leq 2\big(\nu_T^{(1)}\big)^2 \sum_{t=1}^{T} \min\Big\{1, \|g(x_t; \theta_0)/\sqrt{m}\|_{H_{t-1}^{-1}(\theta^*)}^2\Big\} + 1$$

$$\leq 2\kappa\big(\nu_T^{(1)}\big)^2 \sum_{t=1}^{T} \min\Big\{1, \|g(x_t; \theta_0)/\sqrt{m}\|_{\widetilde{V}_{t-1}^{-1}}^2\Big\} + 1$$

$$\leq 4\kappa\big(\nu_T^{(1)}\big)^2 \log \frac{\det \widetilde{V}_T}{\det \kappa\lambda_0 \mathbf{I}} + 1$$

$$\leq 4\kappa\widetilde{d}\big(\nu_T^{(1)}\big)^2 + 1,$$

where the second inequality follows from $\kappa \geq 1$, and $H_t(\theta^*) \succeq (1/\kappa)V_t \succeq (1/\kappa)\widetilde{V}_t$, the third inequality follows from Lemma H.2, and the last inequality follows from the definition of $\widetilde{d}$.

Next we can show the upper bound of $|\mathcal{T}_2|$ in a similar way:

$$|\mathcal{T}_2| \cdot \min\{1, 1^2\} \leq \sum_{t=1}^{T-|\mathcal{T}_1(T)|} \min\Big\{1, |g(x_{\tau(t)}; \theta_0)^\top(\widetilde{\theta}_{\tau(t)-1} - \theta^*)|^2\Big\}$$

$$\leq \sum_{t=1}^{T-|\mathcal{T}_1(T)|} \min\Big\{1, \|g(x_\tau(t); \theta_0)/\sqrt{m}\|_{\underline{W}_{\tau(t-1)}^{-1}}^2 \cdot m\|\widetilde{\theta}_{\tau(t)-1} - \theta^*\|_{\underline{W}_{\tau(t-1)}}^2\Big\}$$

We have $\|g(x_{\tau(t)}; \theta_0)/\sqrt{m}\|_{\underline{W}_{\tau(t-1)}^{-1}}^2 \leq \sqrt{3\kappa}\|g(x_{\tau(t)}; \theta_0)/\sqrt{m}\|_{\underline{V}_{\tau(t-1)}^{-1}}^2$ using the result of Equation (54). Also, we have $m\|\widetilde{\theta}_{\tau(t)-1} - \theta^*\|_{\underline{W}_{\tau(t-1)}}^2 \leq 4\big(\nu_{\tau(t)-1}^{(2)}\big)^2$ using the result of Equation (55). Since $\nu_t^{(2)}$ is non-decreasing in $t$, substituting results back gives

$$|\mathcal{T}_2| \leq 12\kappa\big(\nu_T^{(2)}\big)^2 \sum_{t=1}^{T-|\mathcal{T}_1(T)|} \min\Big\{1, \|g(x_\tau(t); \theta_0)/\sqrt{m}\|_{\underline{V}_{\tau(t-1)}^{-1}}^2\Big\}$$

$$\leq 24\kappa\big(\nu_T^{(2)}\big)^2 \log \frac{\det \underline{\widetilde{V}}_{\tau(T-|\mathcal{T}_1|)}}{\det \kappa\lambda_0 \mathbf{I}}$$

$$\leq 24\kappa\widetilde{d}\big(\nu_T^{(2)}\big)^2,$$

where the second inequality follows from Lemma H.2, and the last inequality follows from the definition of $\widetilde{d}$, finishing the proof.

## G.5 Proof of Proposition G.4

We begin with the case of $\mathcal{T}_3$. We define a new design matrix that consists of all feature vectors of $\underline{V}_t$ up to time $t$, in their original order, including only those corresponding to timesteps in $\mathcal{T}_3(t)$:

$$\underline{\widetilde{V}}_t = \sum_{i=1}^{t-|\mathcal{T}_1(t)|} \frac{1}{m} \mathbb{1}\{i \in \mathcal{T}_3\} g(x_{\tau(i)}; \theta_0) g(x_{\tau(i)}; \theta_0)^\top + \lambda_0 \mathbf{I}$$

For brevity we define $j(t) = \tau(t - |\mathcal{T}_1(t)|)$. Then we have

$$\det(\underline{\widetilde{V}}_T) = \det\Big(\sum_{i=1}^{j(T)} \frac{1}{m} \mathbb{1}\{i \in \mathcal{T}_3\} g(x_{\tau(i)}; \theta_0) g(x_{\tau(i)}; \theta_0)^\top + \lambda_0 \mathbf{I}\Big)$$

$$= \det\Big(\underline{\widetilde{V}}_{\tau(j(T)-1)} + \frac{1}{m} \mathbb{1}\{\tau(j(T)) \in \mathcal{T}_3\} g(x_{\tau(j(T))}; \theta_0) g(x_{\tau(j(T))}; \theta_0)^\top\Big)$$

$$= \det\Big(\underline{\widetilde{V}}_{\tau(j(T)-1)}\Big)\Big(1 + \mathbb{1}\{\tau(j(T)) \in \mathcal{T}_3\} \|g(x_{\tau(j(T))}; \theta_0)/\sqrt{m}\|_{\underline{\widetilde{V}}_{\tau(j(T)-1)}^{-1}}^2\Big)$$

$$\geq \det\Big(\underline{\widetilde{V}}_{\tau(j(T)-1)}\Big)\Big(1 + \mathbb{1}\{\tau(j(T)) \in \mathcal{T}_3\}\Big),$$

where the third equality follows from the matrix determinant lemma, and the inequality follows from the definition of $\mathcal{T}_3$. Repeating inequalities to $\underline{V}_\tau(0)$ gives

$$\det(\widetilde{\underline{V}}_T) \geq \det\left(\widetilde{\underline{V}}_{\tau(0)}\right) \cdot \left(1 + \mathbb{1}\{\tau(j(T)) \in \mathcal{T}_3\}\right)^{T - |\mathcal{T}_1(T)|} = \det(\kappa\lambda_0\mathbf{I}) \cdot (1+1)^{|\mathcal{T}_3|}.$$

Therefore, we can rewrite as

$$|\mathcal{T}_3| \leq \frac{1}{\log 2} \cdot \log \frac{\det \widetilde{\underline{V}}_T}{\det \kappa\lambda_0\mathbf{I}} \leq \frac{1}{\log 2} \cdot \log \frac{\det \widetilde{V}_T}{\det \kappa\lambda_0\mathbf{I}} \leq 2\widetilde{d},$$

where the last inequality follows from the definition of $\widetilde{d}$. We can prove $|\mathcal{T}_4| \leq 2\widetilde{d}$ in a similar way, starting by defining $\widetilde{\underline{W}}_t = \sum_{i=1}^{t - |\mathcal{T}_1(t)|} \frac{\dot{\mu}(f(x_{\tau(i)}; \theta_0)}{m} \mathbb{1}\{i \in \mathcal{T}_4\} g(x_{\tau(i)}; \theta_0) g(x_{\tau(i)}; \theta_0)^\top + \lambda_0\mathbf{I}$ and following the above process.

## H  AUXILIARY LEMMAS

**Lemma H.1** (Freedman (1975)). *Let $M, v > 0$ be fixed constants. Let $\{x_i\}_{i=1}^n$ be a stochastic process, $\{\mathcal{G}_i\}_i$ be a filtration so that for all $i \in [n]$, $x_i$ is $\mathcal{G}_i$-measurable, while almost surely $\mathbb{E}[x_i|\mathcal{G}_{i-1}] = 0$, $|x_i| \leq M$ and*

$$\sum_{i=1}^n \mathbb{E}[x_i^2|\mathcal{G}_{i-1}] \leq v.$$

*Then, for any $\delta > 0$, with probability at least $1 - \delta$,*

$$\sum_{i=1}^n x_i \leq \sqrt{2v\log(1/\delta)} + 2/3 \cdot M\log(1/\delta).$$

**Lemma H.2** (Lemma 11 Abbasi-Yadkori et al. (2011)). *For any $\lambda > 0$ and sequence $\{x_t\}_{t=1}^T \in \mathbb{R}^d$, define $Z_t = \lambda\mathbf{I} + \sum_{i=1}^t x_i x_i^\top$. Then, provided that $\|x_t\|_2 \leq L$ holds for all $t \in [T]$, we have*

$$\sum_{t=1}^T \min\{1, \|x_t\|_{Z_{t-1}^{-1}}^2\} \leq 2\log \frac{\det Z_T}{\det \lambda\mathbf{I}} \leq 2d\log \frac{d\lambda + TL^2}{d\lambda}$$

**Lemma H.3** (Lemma 7 Abeille et al. (2021)). *For any $z', z'' \in \mathbb{R}$, we have,*

$$\dot{\mu}(z')\frac{1 - \exp(1 - |z' - z''|)}{|z' - z''|} \leq \int_0^1 \dot{\mu}(z' + v(z'' - v'))dv \leq \dot{\mu}(z')\frac{\exp(|z' - z''|) - 1}{|z' - z''|},$$

*Also, we have,*

$$\int_0^1 \dot{\mu}(z' + v(z'' - v'))dv \geq \frac{\dot{\mu}(z')}{1 + |z' - z''|}, \quad \int_0^1 \dot{\mu}(z' + v(z'' - v'))dv \geq \frac{\dot{\mu}(z'')}{1 + |z' - z''|}.$$

## I  THOMPSON SAMPLING-BASED VARIANTS

In this section, we introduce the Thompson sampling-based variants of Algorithm 1, which we call NeuralLog-TS-1. The proof for the regret of Neural-TS-1 can be obtained by exactly following the proof of Theorem 3.5 of Zhang et al. (2021). To reuse the result of previous work, we match the notations by using the following definitions:

$$\sigma_t(x)^2 := \kappa\lambda_t \|g(x; \theta_0)/\sqrt{m}\|_{V_t^{-1}}^2$$

$$\nu_t := \nu_t^{(1)}\lambda_t^{-1/2} = C_6\lambda_t^{-1/2}(1 + \sqrt{L}S + LS^2)\iota_t + \lambda_t^{-1/2}$$

$$c_t := \nu_T(1 + \sqrt{2\log(Kt^2)})$$

and denote $\mathcal{F}_t$ as a filtration containing the history of observations up to iteration $t$. Also define the set of saturated points as

$$\mathcal{S}_t = \{x \in \mathcal{X}_t : \Delta_t(x) > c_{t-1}\sigma_{t-1}(x) + 2\epsilon'_{t-1}\}, \tag{62}$$

where $\Delta_t(x) = h(x_t^*) - h(x)$ and $\epsilon_t' = R^{-1}\epsilon_{3,t}$. Note that $x_t^* \notin \mathcal{S}_t$.

In round $t$, for each $x \in \mathcal{X}_t$, we sample a latent reward $\widetilde{r}_t(x)$ from the normal distribution

$$\forall x \in \mathcal{X}_t, \quad \widetilde{r}_t(x) \sim \mathcal{N}(f(x;\theta_{t-1}), \nu_T^2 \sigma_{t-1}^2(x)),$$

and choose an arm following

$$x_t = \arg\max_{x \in \mathcal{X}_t} \widetilde{r}_t(x).$$

Now we introduce two good events: First, define the event $\mathcal{E}_1(t)$ when the following inequality holds for all $x \in \mathcal{X}_t$:

$$|h(x) - f(x;\theta_{t-1})| \leq \nu_T \sigma_{t-1}(x) + \epsilon_{t-1}'. \tag{63}$$

Then, by the direct result of Lemma 6.4, $\mathbb{P}(\mathcal{E}_1(t)) \geq 1 - \delta$. Next, define the event $\mathcal{E}_2(t)$ when the following inequality holds for all $x \in \mathcal{X}_t$:

$$|\widetilde{r}_t(x) - f(x;\theta_{t-1})| \leq \nu_T \sqrt{2\log(Kt^2)}\sigma_{t-1}(x). \tag{64}$$

Since $\widetilde{r}_t(x)$ is sampled from $\mathcal{N}(f(x;\theta_{t-1}), \nu_T^2 \sigma_{t-1}^2(x))$, we can use the concentration inequality on Gaussian distributions to obtain $\mathbb{P}(\mathcal{E}_2(t)|\mathcal{F}_{t-1}) \geq 1 - 1/t^2$ for any possible filtration $\mathcal{F}_{t-1}$.

Next, recall the definition of the set of saturated points in Equation (62). We reuse the result of Lemma 4.5 of Zhang et al. (2021) as follows

$$\mathbb{P}(x_t \in \mathcal{X}_t \setminus \mathcal{S}_t \mid \mathcal{F}_{t-1}, \mathcal{E}_1(t)) \geq (4e\sqrt{\pi})^{-1} - 1/t^2. \tag{65}$$

We skip the proof as the same argument can be found in Section B.4 of Zhang et al. (2021). Instead, we give a high-level intuition. By construction, saturated arms are those whose posterior mean reward is significantly worse than that of the optimal arm. Under the good events $\mathcal{E}_1(t)$ and $\mathcal{E}_2(t)$, this gap is reflected both in their true means and in their posterior samples, so with high probability a saturated arm cannot catch up to the optimal arm in terms of the sampled reward.

On the other hand, the posterior for the optimal arm enjoys an anti-concentration property, which is, with constant probability, its sample exceeds its mean by a suitable margin. This is where the factor $(4e\sqrt{\pi})^{-1}$ comes from. Combining these facts, with constant probability the sampled reward of the optimal arm is larger than the samples of all saturated arms, so the arm selected by Thompson sampling must be unsaturated. The $1/t^2$ term accounts for the small probability that one of the good events $\mathcal{E}_1(t)$ or $\mathcal{E}_2(t)$ fails.

Now, with the previous results in place, we derive an upper bound on the expected instantaneous regret. Define $d_t = h(x_t^*) - h(x_t)$. Again, we reuse the result of Lemma 4.6 of Zhang et al. (2021) as follows:

$$\mathbb{E}[d_t \mid \mathcal{F}_{t-1}, \mathcal{E}_1(t)] \leq 44e\sqrt{\pi} C_1 c_t \sqrt{L} \, \mathbb{E}\big[\min\{1, \sigma_t(x_t)\} \mid \mathcal{F}_{t-1}, \mathcal{E}_1(t)\big] + 4\epsilon_{t-1}' + 2/t^2,$$

where $C_1 > 0$ is the same absolute constant that appears in Lemma C.4. By Equation (65), Neural-TS-1 selects an unsaturated arm with constant probability, so in expectation the posterior standard deviation of the played arm is comparable to that of the best unsaturated arm. Under the good events, the posterior means stay close to the true means and saturated arms have very small gaps, which allows us to bound the instantaneous regret $d_t$ by a constant multiple of $\min\{1, \sigma_t(x_t)\}$ plus the approximation terms $4\epsilon_{t-1}' + 2/t^2$. Taking the conditional expectation and using a global control on the posterior variances over time then yields the stated bound.

Now we are ready to start the proof for the regret. Define a stochastic process $(Y_t)_{t=0}^T$ where

$$\bar{d}_t = d_t \mathbf{1}\{\mathcal{E}_1(t)\}$$
$$X_t = \bar{d}_t - 44e\sqrt{\pi}C_1 c_t \sqrt{L} \min\{1, \sigma_t(x_t)\} - 4\epsilon_{t-1}' - 2/t^2$$
$$Y_t = \sum_{i=1}^t X_i, \quad Y_0 = 0$$

We can see that $(Y_t)$ is a supermartingale with respect to $\mathcal{F}_t$ since $\mathbb{E}[Y_t - Y_{t-1} \mid \mathcal{F}_{t-1}] = \mathbb{E}[X_t \mid \mathcal{F}_{t-1}] \leq 0$. Now we prepare to apply the Azuma-Hoeffding inequality for a supermartingale:

**Lemma I.1** (Azuma-Hoeffding inequality for supermartingale). *If a supermartingale $Y_t$, corresponding to a filtration $\mathcal{F}_t$ satisfies $|Y_t - Y_{t-1}| \leq B_t$, then for any $(0,1)$, with probability at least $1 - \delta$,*

$$Y_t - Y_0 \leq \sqrt{2 \log(1/\delta) \sum_{i=1}^{t} B_i^2}.$$

To derive an upper bound on $|Y_t - Y_{t-1}|$, we have

$$|Y_t - Y_{t-1}| = |X_t| \leq |\bar{d}_t| + 44e\sqrt{\pi}C_1 c_t \sqrt{L} \min\{1, \sigma_t(x_t)\} + 4\epsilon'_{t-1} + 2/t^2$$
$$\leq 4 + 44e\sqrt{\pi}C_1^2 c_t L + 4\epsilon'_{t-1}.$$

where the last inequality follows from Lemma C.4, and $1/t^2 \leq 1$. Now, applying Lemma I.1 with $B_t = 4 + 44e\sqrt{\pi}C_1^2 c_t L + 4\epsilon'_{t-1}$ to $(Y_t)$, with probability at least $1 - \delta$, we have

$$\sum_{t=1}^{T} \bar{d}_t \leq \underbrace{\sum_{t=1}^{T} 44e\sqrt{\pi}C_1 c_t \sqrt{L} \min\{1, \sigma_t(x_t)\}}_{\textbf{(term 1)}} + \underbrace{\sum_{t=1}^{T} 4\epsilon'_{t-1} + \sum_{t=1}^{T} 2/t^2}_{\textbf{(term 2)}}$$
$$+ \underbrace{\sqrt{2 \log(1/\delta) \sum_{t=1}^{T} \left(4 + 44e\sqrt{\pi}C_1^2 c_t L + 4\epsilon'_{t-1}\right)^2}}_{\textbf{(term 3)}}. \tag{66}$$

For **(term 1)**, applying Cauchy-Schwarz inequality,

$$\textbf{(term 1)} \leq 44e\sqrt{\pi}C_1 \nu_T^{(1)}(1 + \sqrt{2\log(KT^2)})\sqrt{\kappa L}\sqrt{T \sum_{t=1}^{T} \min\{1, \|g(x_t; \theta_0)/\sqrt{m}\|_{V_{t-1}^{-1}}^2\}}$$
$$= \widetilde{\mathcal{O}}\left(S^2 \widetilde{d}\sqrt{\kappa T} + S^{2.5}\sqrt{\kappa \widetilde{d} T}\right)$$

For **(term 2)**, by the condition of $m$ in Condition 4.4, $\sum_{t=1}^{T} 4\epsilon'_{t-1} \leq 1$, and $\sum_{t=1}^{T} 2/t^2 \leq \pi^2/3$. For **(term 3)**, since $\nu_T^{(1)} = \widetilde{\mathcal{O}}(S^2\sqrt{\widetilde{d}} + S^{2.5})$, and $\lambda_0^{-1/2} = \mathcal{O}(S^{0.5})$

$$\textbf{(term 3)} \leq (4 + 44e\sqrt{\pi}C_1^2 \nu_T^{(1)}\lambda_0^{-1/2}(1 + \sqrt{2\log(KT^2)}) + 4\epsilon'_T)\sqrt{2\log(1/\delta)T}$$
$$= \widetilde{\mathcal{O}}\left(S^{2.5}\sqrt{\widetilde{d}T} + S^3\sqrt{T}\right)$$

Combining results, we have

$$\sum_{t=1}^{T} \widetilde{d}_t \leq \widetilde{\mathcal{O}}\left(S^2 \widetilde{d}\sqrt{\kappa T} + S^{2.5}\sqrt{\kappa \widetilde{d} T} + S^3\sqrt{T}\right)$$

with probability at least $1 - \delta$. Notice that $\text{Regret}(T) \leq \sum_{t=1}^{T} R|h(x_t^*) - h(x_t)|$. Therefore $R\sum_{t=1}^{T} \widetilde{d}_t$ upper bounds the regret with probability at least $1 - \delta$. Finally, replacing $\delta$ by $\delta/2$ for both cases and applying the union bound finishes the proof.

**Remark 3** (Discussion on Thompson sampling-based variants of NeuralLog-UCB-2). *In analogy with the Thompson sampling extension of NeuralLog-UCB-1, one can also consider a Thompson sampling-based variant of NeuralLog-UCB-2 as follows. Define $\sigma'_t(x)^2 := \lambda_t \|g(x; \theta_0)/\sqrt{m}\|_{W_{t-1}^{-1}}$ and $\nu'_t := \nu_{t-1}^{(2)}\lambda_t^{-1/2}$, and for all $x \in \mathcal{X}_t$ sample $\widetilde{r}'_t(x) \sim \mathcal{N}(g(x; \theta_0)^\top(\theta_{t-1} - \theta_0), \nu_T'^2\sigma_{t-1}'^2(x))$, then choose $x_t = \arg\max_{x \in \mathcal{X}_t} \widetilde{r}'_t(x)$. However, our current regret analysis for Thompson sampling-based algorithms proceeds by defining a stochastic process associated with the per-round regret and then applying a concentration inequality for this process to obtain an upper bound on the per-round regret. In order to fully exploit $W_t$ from Algorithm 2 within this framework, a much more delicate analysis of the second-order Taylor expansion of the per-round regret would be required.*

*More concretely, if we proceed the analysis in a naive way and consider the regret bound obtained for such a NeuralLog-TS-2 algorithm, then, denoting by **(term 1')** the counterpart of **(term 1)** in Equation (66), and focusing only on the dependence on $\kappa$, we obtain*

$$\textbf{(term 1')} \lesssim \sum_{t=1}^{T} \min\{1, \|g(x;\theta_0)/\sqrt{m}\|_{W_{t-1}^{-1}}\} \lesssim \sqrt{\kappa T \sum_{t=1}^{T} \min\{1, \|g(x;\theta_0)/\sqrt{m}\|_{V_{t-1}^{-1}}^2\}},$$

*where we see that the additional $\sqrt{\kappa}$ factor is reintroduced. Treating this issue within our current proof technique therefore appears to be a non-trivial problem, and we leave a sharper analysis of such Thompson sampling-based variants of NeuralLog-UCB-2 for future work.*

As we have seen in Remark 3, although NeuralLog-TS-2 does not attain a regret bound with the same dependence on $\kappa$ as NeuralLog-UCB-2, the algorithm itself is well defined, just like NeuralLog-TS-1. In Section J, we present additional experiments including these two algorithms and demonstrate their practical performance.

## J ADDITIONAL EXPERIMENTS

We compare five baseline algorithms with our algorithms including the Thompson sampling-based variants introduced in Section I. Where NeuralLog-TS-1 and NeuralLog-TS-2 both choose the arm with best sampled reward where

$$\text{For NeuralLog-TS-1,} \quad \widetilde{r}_t(x) \sim \mathcal{N}(f(x;\theta_{t-1}), \nu_T^2 \sigma_{t-1}^2(x)),$$
$$\text{For NeuralLog-TS-2,} \quad \widetilde{r}'_t(x) \sim \mathcal{N}(g(x;\theta_0)^\top(\theta_{t-1}-\theta_0), \nu_T'^2 \sigma_{t-1}'^2(x)).$$

We include the synthetic latent reward functions which are also used in Zhou et al. (2020): $h_4(x) = 10(x^\top\theta)^2$, $h_5(x) = x^\top\Theta^\top\Theta x$, $h_6(x) = \cos(3x^\top\theta)$. All other experimental parameters and details follow the same as described in Section 7.

Next, we include 3 more $K$-class classification tasks from Dua & Graff (2019): We reuse the same min–max normalization to $[-1, 1]$ as described in Section A. In the `magic` dataset (MAGIC Gamma Telescope), we convert all features to real-valued variables, impute any missing entries with 0, and then map the original class labels to a binary label by setting $y = 1$ for gamma ('g') events and $y = 0$ for hadron ('h') events. In the `banknote` dataset (UCI Banknote Authentication), we use the four real-valued attributes provided in the repository and keep the original binary labels $y \in \{0, 1\}$. For the `phoneme` dataset (Connectionist Bench (Nettalk Corpus)), we treat any categorical fields as numeric by casting them to an appropriate numeric type, and replace missing values with 0.

Figures 4 and 5 summarize the average cumulative regret of the five baseline algorithms together with our two Thompson sampling-based variants. Consistent with the results already observed in Figures 1 and 2, our NeuralLog-UCB-2 algorithm steadily achieves the best performance across the considered settings. Moreover, the two Thompson sampling-based variants also exhibit competitive performance compared to the baselines. Figure 6 demonstrates the influence of $\widetilde{d}$ on data-adaptive algorithms by comparing cumulative regret across different values of $\widetilde{d}$.

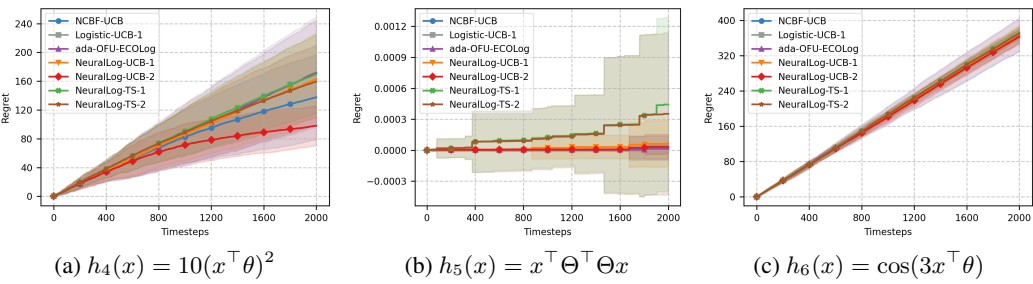

(a) $h_4(x) = 10(x^\top\theta)^2$      (b) $h_5(x) = x^\top\Theta^\top\Theta x$      (c) $h_6(x) = \cos(3x^\top\theta)$

Figure 4: Comparison of cumulative regret of baseline algorithms for nonlinear reward functions.

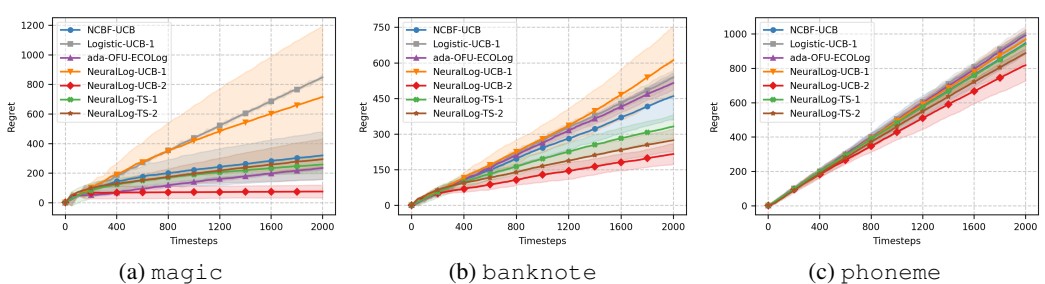

(a) `magic`  (b) `banknote`  (c) `phoneme`

Figure 5: Comparison of cumulative regret of baseline algorithms for real-world dataset.

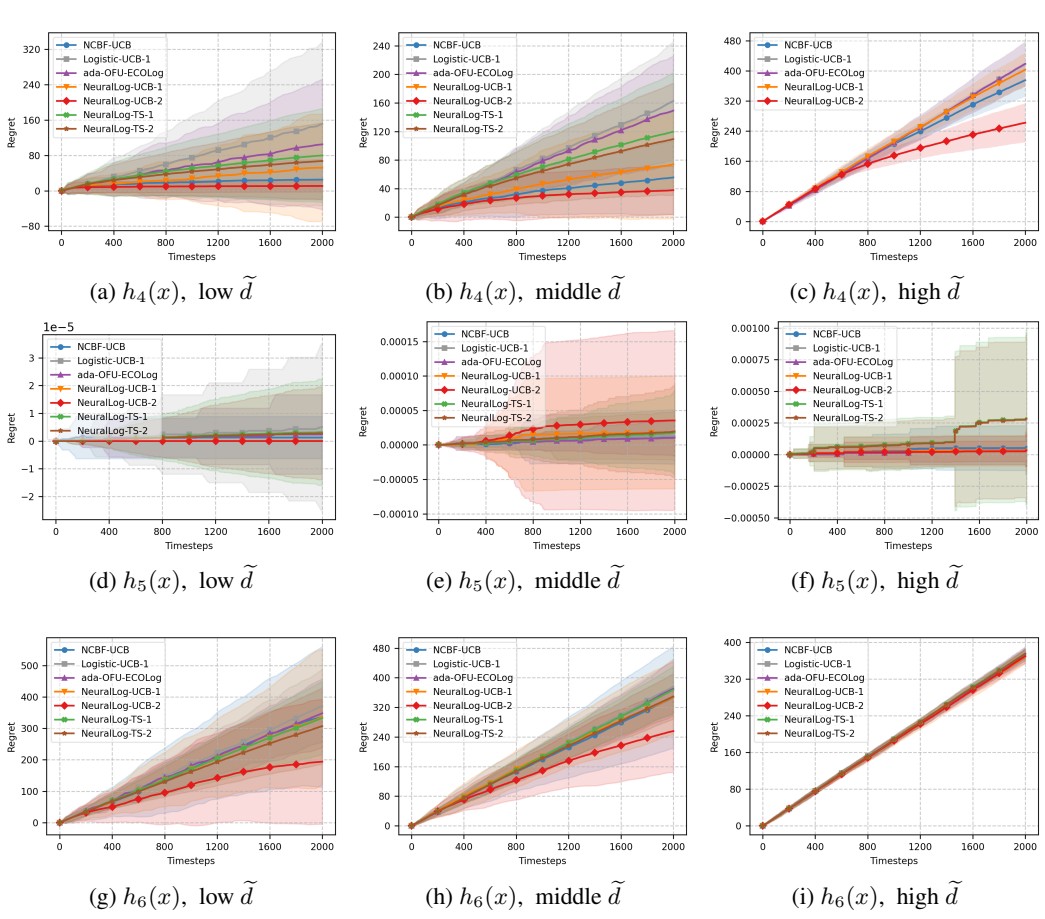

(a) $h_4(x)$, low $\widetilde{d}$  (b) $h_4(x)$, middle $\widetilde{d}$  (c) $h_4(x)$, high $\widetilde{d}$

(d) $h_5(x)$, low $\widetilde{d}$  (e) $h_5(x)$, middle $\widetilde{d}$  (f) $h_5(x)$, high $\widetilde{d}$

(g) $h_6(x)$, low $\widetilde{d}$  (h) $h_6(x)$, middle $\widetilde{d}$  (i) $h_6(x)$, high $\widetilde{d}$

Figure 6: Comparison of cumulative regret of baseline algorithms with varying effective dimension $\widetilde{d}$.

## K  ADDITIONAL FUTURE DIRECTIONS

Although we successfully remove the direct dependence on $p$ from the regret bound, a direct dependence on $p$ reappears when we examine the per-round computational complexity. This is problematic in neural bandit settings where $p$ scales with the horizon $T$, making the resulting algorithm computationally inefficient.

Let us briefly analyze the computational complexity of our algorithms. Since Algorithms 1 and 2 have the same order of complexity, we focus on Algorithm 1. For action selection, we must compute $f(x; \theta_{t-1})$ for $K$ actions, which costs $\mathcal{O}(p)$ per action, and the quantity $\|g(x; \theta_0)/\sqrt{m}\|_{V_{t-1}^{-1}}$, which costs $\mathcal{O}(p^2)$ per action. Hence, the action-selection step has complexity $\mathcal{O}(Kp^2)$. The updates of the parameters $\lambda_t$, $\iota$, and $\nu_t^{(1)}$ cost $\mathcal{O}(p^2)$ by their definitions. For neural network training, at round $t$ we apply gradient steps over the full dataset of size $t$, which costs $\mathcal{O}(tp)$ per gradient step. Performing $J_t$ iterations therefore costs $\mathcal{O}(J_t tp)$, where $J_t = \widetilde{\mathcal{O}}(TL/\lambda_t)$. Finally, updating the design matrix $V_t$ costs $\mathcal{O}(p^2)$. Altogether, the per-round computational complexity is $\mathcal{O}(J_t tp + Kp^2 + p^2)$. Moreover, Verma et al. (2025) can be seen to have essentially the same computational complexity, as their algorithm and training pipeline are close to ours.

In contrast, in the classical logistic bandit literature the algorithms operate directly in the feature space of dimension $d$, which is typically much smaller than $p$. For example, Filippi et al. (2010); Faury et al. (2020) obtain overall complexity on the order of $\mathcal{O}(d^2 K + d^2 T)$, and there has been significant recent progress on designing computationally efficient algorithms for logistic bandits: Abeille et al. (2021) achieve $\mathcal{O}(d^2 KT)$, and Faury et al. (2022) even propose an algorithm with complexity $\widetilde{\mathcal{O}}(d^2 K)$. However, these favorable guarantees rely crucially on the strong assumption that the latent reward model is linear in the feature representation. From the perspective of practical applications, it is therefore important to develop neural bandit algorithms that retain the modeling flexibility of neural networks while achieving comparable computational efficiency, which we view as an important direction for future work.

As one illustrative example, in light of the connection between NTK-based neural bandits and kernelized bandits, one could consider importing techniques such as Nyström approximation as in Zenati et al. (2022) to reduce the effective computational cost in the neural bandit setting. Another approach is to adapt the method proposed in Xu et al. (2022) where an NTK-based neural bandit formulation is also used, but the neural network is trained so that its output is not the reward itself, but instead a new $d$-dimensional feature vector. The problem is then reduced to solving a linear bandit in this learned feature space with respect to an unknown parameter $\theta^*$. This strategy can substantially reduce computational complexity and is attractive from an applied viewpoint, but it requires an additional Lipschitz-type assumption on the neural network on the theoretical side.

## L  USE OF LARGE LANGUAGE MODELS

This manuscript is reviewed and edited for grammar and clarity using ChatGPT-5.