# OpenReview forum: "Neural Logistic Bandits"
_ICLR.cc/2026/Conference — Submitted to ICLR 2026_

### Official Review · Reviewer_igeJ · 2025-10-27

**Soundness:** 2
**Presentation:** 3
**Contribution:** 3
**Rating:** 4
**Confidence:** 3

**Summary:**

This paper proposes NeuralLog-UCB, a framework for neural logistic bandits. NeuralLog-UCB utilizes a tail inequality for self-normalized vector-valued martingales to bypass a direct dependence on the ambient dimension. Theoretical results show an improvement in the cumulative regret bound for existing neural logistic bandits. Empirical results validate these findings using both synthetic and real-world datasets.

**Strengths:**

- This paper is generally well-written and clearly explains the algorithm's key aspects.
- The theoretical results are clear with mathematical notation, assumptions, statements, and proof of the proposed method.
- The theory is also confirmed by experimental evidence, e.g., cumulative regrets in Fig.1 and 2.

**Weaknesses:**

- The proposed method requires pre-defining and fine-tuning several parameters, such as a **known** minimum variance of reward distributions $1/\kappa$, a reward upper bound $R$, norm parameter $S$, exploration rate $\lambda$, etc. This overall raises concern about its robustness and usefulness in practice.
- Bersterin-type self-normalized inequality for vector-valued martingales is not new in bandits literature [1,2]. This reduces the soundness of this paper regarding theoretical contributions. Note that I still appreciate your analysis in the logistic bandits setting.
- Empirical results may need to be improved with a standard synthesis dataset in neural-contextual bandits (e.g., [3]) and more UCI datasets.

**Questions:**

1. In Alg.1 & 2, while $\lambda$, $\nu$ are updated by Eq.4 and Eq.2 (5), why do the authors fix these with the best parameter values using grid search in experiments?

2. How realistic of the assumption about a large enough width of the neural network $m$ in Condition C.2.? In practice, $T^4K^4L^6$ is often much higher than $m$, where $T$ is the time horizon, $K$ is the number of arms, and $L$ is the number of layers.

---
References:

[1] Zhou et al., Nearly minimax optimal reinforcement learning for linear mixture markov decision processes, CLT, 2021.

[2] Faury et al., Improved optimistic algorithms for logistic bandits, ICML, 2020.

[3] Zhou et al., Neural Contextual Bandits with UCB-based Exploration, ICML, 2021.

---

> ### Author Response · Authors · 2025-11-21
>
> ### [W1] The proposed method requires pre-defining and fine-tuning several parameters, such as a known minimum variance of reward distributions $1/\kappa$, a reward upper bound $R$, norm parameter $S$, exploration rate $\lambda$, etc. This overall raises concern about its robustness and usefulness in practice.
>
> We thank the reviewer for the comment. Let us clarify and explain how these parameters are chosen, in regard to the reviewer's comment.
>
> - (Parameter $\kappa$) While it is true that Algorithm 1 requires prior knowledge of $\kappa$, Algorithm 2 does not require prior knowledge of $\kappa$. For Algorithm 1, when defining the design matrix $V_t = \sum_{i=1}^t \frac{1}{m} g(x_i;\theta_0)g(x_i;\theta_0)^\top + \kappa\lambda_t \mathbf{I}$ and computing the UCB term for each arm. However, in Algorithm 2, we replace $V_t$ with the neural-network–estimated variance $W_t = \sum_{i=1}^t \frac{1}{m} \mu(f(x_i;\theta_i)) g(x_i;\theta_0) g(x_i;\theta_0)^\top + \lambda \mathbf{I}$.
> This substitution avoids the pessimistic upper bound $\kappa H_t \succeq V_t$ and removes the $\kappa$ factor from the UCB term. The resulting $\kappa$-free UCB yields both a stronger theoretical guarantee and an algorithm that does not require prior knowledge of $\kappa$ (which neither Algorithm 1 nor the previous method in [1] achieves), making it more practical.
>
> - (Parameter $R$) By definition, $R$ is the maximum value of the sigmoid derivative $\dot\mu(\cdot)$, so we can simply set $R = 1/4$.
>
> - (Parameter $S$) We agree with the reviewer that assuming prior knowledge of a norm parameter $S$ satisfying $S \geq \sqrt{2h^\top H^{-1}h}$ may be restrictive in practice. We emphasize, however, that this is conceptually analogous to the standard assumption in contextual linear bandits that the true parameter $\theta^\*$ satisfies $\|\theta^*\|_2 \leq S$ and that the learner knows such an $S$ (see, e.g., [1, 2]). In our analysis, $S$ plays the same role. It is a norm bound that the true parameter lies in $\|\theta^\*-\theta_0\|_2\leq S$, and can be viewed as a theoretical regularity parameter rather than something that must be computed exactly from the data.
>
> In addition, we see the approach of [3] as an interesting future work direction to avoid this dependence of $S$. In the linear bandit setting, [3] shows that one can avoid explicit knowledge of $S$ by introducing a carefully chosen time-varying regularizer $\lambda_t$ so that $\sqrt{\lambda_t} S \leq R \sqrt{d}$ holds, where $R$ is the $R^2$-sub-Gaussian noise parameter. This design removes $S$ from the confidence-radius expression. Adapting a similar decaying-regularization strategy to our neural bandit setting, so as to eliminate the explicit appearance of $S$, is an interesting and non-trivial direction for future work.
>
> - (Parameter $\lambda$) We agree that our algorithms require specifically choosing parameters $\lambda_t$'s. Theoretically, we showed that our choice of $\lambda_t$'s leads to a desired regret upper bound. At the same time, our experiments show that our algorithms perform well compared to existing logistic bandit algorithms. In fact, some level of tuning for the regularization parameter $\lambda$ is commonly required for logistic bandits and neural bandits.

---

> ### Author Response · Authors · 2025-11-21
>
> ### [W2] Bersterin-type self-normalized inequality for vector-valued martingales is not new in bandits literature [1,2]. This reduces the soundness of this paper regarding theoretical contributions. Note that I still appreciate your analysis in the logistic bandits setting.
>
> We agree with the reviewer that Bernstein-type self-normalized inequalities for vector-valued martingales have appeared before in the bandit literature (e.g., [1,2]). However, there is no single “off-the-shelf’’ bound that can simply be plugged into every bandit problem: different works prove different variants tuned to their own noise models, geometries, and algorithms. As we explain in Section 3, our main theoretical contribution is a variance- and data-dependent self-normalized inequality tailored to the neural logistic bandit setting, which is specifically constructed to (i) avoid any explicit dependence on the ambient dimension $d$, and (ii) achieve the refined $\kappa$-dependence and effective-dimension $\tilde d$-dependence required for our regret bounds, while integrating cleanly with our projection-free update for the neural logistic MLE. This construction goes beyond a straightforward or black-box reuse of existing tools from [1,2].
>
>
> To clarify the technical contribution, let us briefly outline where the main difficulty arises. The idea of reducing the $\kappa$ factor goes back to [2], who use a Bernstein-type condition on martingale increments to obtain sharper bounds. Their proof defines, for each $\xi \in \mathbb{R}^d$ and $t \ge 1$, a nonnegative supermartingale $M_t(\xi)$ and then applies the pseudo-maximization principle to construct another nonnegative supermartingale $\bar M_t := \int_\xi M_t(\xi)\, h(\xi)\, d\xi$, where $h(\xi)$ is the density of a Gaussian distribution. However, using Bernstein’s inequality in that framework forces them to restrict $\xi$ to the $\ell_2$ unit ball $\mathbb{B}_2(d)$, and truncating the Gaussian to this ball introduces a normalization factor that depends explicitly on the ambient dimension $d$. This is problematic in our neural logistic bandit setting, where we aim for bounds in terms of an effective NTK dimension $\tilde d$ rather than $d$.
>
> Our approach replaces this Gaussian pseudo-maximization step by an argument based on Freedman’s inequality, while still exploiting a Bernstein-like control on the martingale differences to obtain optimal dependence on both $\kappa$ and $\widetilde d$. A key technical challenge is incorporating the non-uniform logistic variances into the design matrix, which we encode as $H_t = \sum_{i=1}^t \sigma_i^2 x_i x_i^\top + \lambda \mathbf{I}$, where $\sigma_i^2$ is the conditional variance at round $i$. This modification injects factors of $\sigma_i^2$ into the denominators of two crucial quantities in the analysis, namely $B_1 = \frac{2 \eta_i x_i^\top H_{i-1}^{-1} s_{i-1} E_{i-1}}{1 + \sigma_i^2 w_i^2}$ and $B_2 = \frac{\eta_i^2 w_i^2}{1 + \sigma_i^2 w_i^2} - \mathbb{E}[\frac{\eta_i^2 w_i^2}{1 + \sigma_i^2 w_i^2} \mid F_{i-1}]$, where the precise definitions of $\eta_i$, $w_i$, $s_{i-1}$ and $E_{i-1}$ are given in Section D. Applying Freedman’s inequality requires tight bounds on $|B_j|$ and on $\sum_{i=1}^n \mathbb{E}[B_j^2 \mid \mathcal{F}_{i-1}]$ for $j = 1,2$. A naive bounding strategy reintroduces an extra factor of $\kappa$, where one of our main technical contributions is to avoid this extra $\kappa$ by carefully absorbing the variance terms into the logarithmic part of the bound, thereby preserving both the desired $\widetilde d$-dependence and the improved $\kappa$-dependence.
>
>
> Beyond the tail inequality, our theoretical analysis also has to control the behavior of the maximum likelihood estimator $\theta_t$. By definition, $\kappa$ satisfies $1/\kappa \le \min_{x \in \mathcal{X},\, \theta \in \Theta} \dot\mu(x^\top \theta)$, so the analysis is only valid as long as $\theta_t$ remains in the parameter set $\Theta$ where the logistic curvature is bounded away from zero. [2] enforce this by projecting $\theta_t$ onto $\Theta$, which requires solving a nonconvex optimization problem at every step. Although [2] make this projection step more tractable, it still incurs a computational cost on the order of $O(d^2 / \varepsilon)$ to achieve $\varepsilon$-accuracy, which is prohibitive in the neural bandit regime. In contrast, we design a new algorithm that avoids any explicit projection while implicitly keeping $\theta_t$ close to $\Theta$. We adapt the regularization parameter $\lambda_t$ based on the observed data and incorporate it directly into the loss $\mathcal{L}_t(\theta)$, which stabilizes the MLE updates and yields a concentration inequality between $\theta_t$ and $\theta^*$ without resorting to nonconvex projection.

---

> ### Author Response · Authors · 2025-11-21
>
> ### [W3] Empirical results may need to be improved with a standard synthesis dataset in neural-contextual bandits (e.g., [3]) and more UCI datasets.
>
> We appreciate the reviewer’s suggestion and have expanded our empirical evaluation accordingly. First, we have added experiments on the standard synthetic dataset used in prior neural contextual bandit work (e.g., [3]), as well as on several additional UCI datasets; the new results are reported in Section J.
>
> Furthermore, we implemented the Thompson sampling-based variants of our two algorithms (NeuralLog-TS-1 and NeuralLog-TS-2) and compared them against our UCB-based methods and existing baselines on these datasets. A brief regret analysis for NeuralLog-TS-1 is provided in Section I, and a discussion of the theoretical challenges in establishing a regret bound for NeuralLog-TS-2 is given in Remark 3.
>
> ----
>
> ### [Q1] In Alg.1 \& 2, while $\lambda$, $\nu$ are updated by Eq.4 and Eq.2 (5), why do the authors fix these with the best parameter values using grid search in experiments?
>
> We agree that there is a difference between the theoretical update rules and our experimental implementation. In theory, the time-varying schedules for $\lambda_t$ and $\nu_t$ in Eq.(4) and Eq.(2), (5) are chosen to obtain clean regret guarantees, but they depend on unknown problem-dependent parameters, that directly guessing them can be even more heuristic and computationally involved. Therefore, in practice, we follow the standard procedure in neural bandit work (e.g., [1,2]), and treat $\lambda$ and $\nu$ as fixed hyperparameters and select them by a simple grid search, which greatly simplifies implementation and leads to more stable performance.
>
> ----
>
> ### [Q2] How realistic of the assumption about a large enough width of the neural network $m$ in Condition C.2.? In practice, $T^4K^4L^6$ is often much higher than $m$, where $T$ is the time horizon, $K$ is the number of arms, and $L$ is the number of layers.
>
> We agree with the reviewer that, as pointed out, choosing the network width $m$ proportional to $T^4 K^4 L^6$ to satisfy the NTK-based theoretical guarantees is unrealistic in practice. This kind of gap between the theoretical condition on $m$ and the widths used in applications is standard in the neural bandit literature and reflects limitations of the current NTK theory.
>
> However, our experiments (including the new experiments with the Thompson sampling-based variants of Algorithm 1 and Algorithm 2, on additional simulated and real-world datasets reported in Section J) use networks whose widths do not satisfy Condition C.2, yet still show empirical performance. This suggests that the theory–practice gap can be mitigated in realistic settings. We leave tightening this gap and developing less conservative width conditions as an important direction for future work.
>
> ----
>
> - [1] Zhou et al., Nearly minimax optimal reinforcement learning for linear mixture markov decision processes, COLT, 2021.
> - [2] Faury et al., Improved optimistic algorithms for logistic bandits, ICML, 2020.
> - [3] Zhou et al., Neural Contextual Bandits with UCB-based Exploration, ICML, 2021.

---

### Official Review · Reviewer_F7Bv · 2025-10-29

**Soundness:** 3
**Presentation:** 3
**Contribution:** 2
**Rating:** 4
**Confidence:** 3

**Summary:**

This paper studies neural logistic bandits, a setting where an agent must choose actions over time and observe binary (Bernoulli) feedback whose probability is modeled by a logistic link on top of an unknown nonlinear reward function approximated with an overparameterized neural network. The authors develop a new Bernstein-style self-normalized concentration inequality that is both variance-adaptive (it avoids relying on worst-case logistic variance constants like \kappa) and data-adaptive (it depends on an effective dimension \tilde{d}, not the full parameter dimension), and use it to design two UCB-based algorithms, NeuralLog-UCB-1 and NeuralLog-UCB-2, with regret bounds.

**Strengths:**

- Improved variance dependence for neural logistic bandits

The paper introduces two UCB-style algorithms (NeuralLog-UCB-1 and NeuralLog-UCB-2) that achieve regret bounds with better dependence on the logistic variance parameters, improving over prior $\tilde{O}(\kappa \tilde{d}\sqrt{T})$ in neural logistic bandits.

- Data-adaptive exploration design

NeuralLog-UCB-2 estimates per-arm uncertainty using a learned, curvature-weighted design matrix rather than a crude global worst-case variance. This yields tighter confidence sets and, in experiments, significantly lower cumulative regret.

- Empirical support

On both synthetic nonlinear rewards and real datasets (MNIST, mushroom, shuttle), the proposed methods — especially NeuralLog-UCB-2 — consistently achieve the lowest regret among strong baselines, suggesting practical benefit and not just a theoretical improvement.

**Weaknesses:**

Weaknesses
-  Removing d is not new

The paper highlights that its regret bounds no longer depend directly on the ambient dimension d, but instead on the effective dimension \tilde{d}. However, this “$d \rightarrow \tilde{d}$” replacement has already been standard since NeuralUCB / NeuralTS–style analyses in neural bandits, where NTK-based arguments control regret via an effective log-det complexity term rather than the raw parameter dimension.

- The regret bounds are not strictly stronger than recent neural bandit results

The improvements mainly target neural logistic bandits, and still rely on strong NTK-style assumptions such as Assumption 4.1 (well-conditioned NTK / non-degeneracy). Some recent neural bandit work [1] weakens or removes such assumptions, so the structural assumptions here are not obviously milder.

[1] Robust neural contextual bandit against adversarial corruptions

The bounds still scale with a global effective dimension \tilde{d} defined using all arms / all rounds. Recent work [2] has started reducing or even removing this global \tilde{d} factor in the leading term.
In short, while the paper improves the $\kappa$ or $\kappa^*$ dependence for logistic neural bandits, it is not a general domination of existing neural bandit theory.

[2] Contextual bandits with online neural regression

- Regret still depends polynomially on S
Both main theorems include factors like $S^2 \tilde{d}\sqrt{\kappa T}$ and even higher-order terms in S (up to $S^5$ in NeuralLog-UCB-2). S measures how far the optimal parameter is from initialization.

**Questions:**

See Weakness

---

> ### Author Response · Authors · 2025-11-21
>
> ### [W1] Removing d is not new / The paper highlights that its regret bounds no longer depend directly on the ambient dimension d, but instead on the effective dimension $\widetilde{d}$. However, this “$d\to\widetilde d$” replacement has already been standard since NeuralUCB / NeuralTS–style analyses in neural bandits, where NTK-based arguments control regret via an effective log-det complexity term rather than the raw parameter dimension.
>
> We believe that there is a slight misunderstanding behind the reviewer's comment. Behind the claim that replacing the ambient dimension $d$ by an effective dimension $\widetilde d$ is now a routine step in neural bandit analyses, it seems implicitly assumed that there is a unified recipe that can always be applied within the NTK-based neural bandit framework.
>
> In our setting, however, this is precisely not the case. Our main technical contribution is to simultaneously obtain (i) a dependence on the effective dimension $\widetilde d$ and (ii) an improved dependence on the curvature factor $\kappa$ in the logistic neural bandit setting. We show that this joint improvement cannot be achieved by directly reusing existing self-normalized martingale tail inequalities, including previous results for logistic bandits [1, 2] and for neural bandits [3, 4]. Applying those tools in a straightforward way inevitably reintroduces a suboptimal $\kappa$-dependence or a direct dependence on $d$, as elaborated in Section 3.
>
> To overcome this, we prove a new self-normalized Bernstein-type inequality tailored to our problem, derived from a different technique based on Freedman's inequality. This variance- and data-adaptive tail inequality yields the most favorable concentration bound needed for our regret analysis in neural logistic bandits.
>
> In this sense, moving from $d$ to $\widetilde d$ in our regret bounds is not merely an automatic “$d \to \widetilde d$ replacement” following existing templates, but relies on a non-trivial new technical ingredient that is specific to neural logistic bandits.

---

> ### Author Response · Authors · 2025-11-21
>
> ### [W2] The regret bounds are not strictly stronger than recent neural bandit results / The improvements mainly target neural logistic bandits, and still rely on strong NTK-style assumptions such as Assumption 4.1 (well-conditioned NTK / non-degeneracy). Some recent neural bandit work [1] weakens or removes such assumptions, so the structural assumptions here are not obviously milder. / [1] Robust neural contextual bandit against adversarial corruptions / The bounds still scale with a global effective dimension $\widetilde{d}$ defined using all arms / all rounds. Recent work [2] has started reducing or even removing this global $\widetilde{d}$ factor in the leading term. In short, while the paper improves the $\kappa$ or $\kappa^*$ dependence for logistic neural bandits, it is not a general domination of existing neural bandit theory. / [2] Contextual bandits with online neural regression
>
> Before addressing the two specific comparisons with two neural bandit works **[1,2]**, we would like to clarify the scope of our contribution. Our goal is not to provide regret bounds that are uniformly stronger than all recent neural bandit results, many of which emphasize orthogonal aspects of the problem (e.g., robustness to adversarial corruptions, relaxed NTK structural assumptions, or realizability-based regression oracles that assume the reward model is exactly captured by the chosen network class). Rather, we focus on the standard neural logistic bandit setting and, within this regime, we obtain regret bounds that are strictly sharper than existing analyses in terms of their dependence on $\kappa$. The following two points explain in more detail.
>
>
>
> - **[1]** studies robust neural contextual bandits with adversarial reward corruptions and relaxes the usual NTK-based arm separateness condition. Instead of assuming that the NTK Gram matrix $H$ built from all context–arm pairs $(x^i)$ for $i=1,\dots,TK$ is positive definite (which forbids duplicate or parallel arms), they work with the NTK matrix $\breve H$ defined on the set of unique arms $\breve A_T$ and show that $\lambda_{\min}(\breve H)>0$ while allowing adversarially chosen, or duplicated, arm contexts. Their regret bounds are still expressed in terms of the NTK effective dimension, but no longer require $H \succ 0$ on the full multiset of arms.
>
> We can see that, this type of NTK relaxation is conceptually orthogonal to the main difficulty tackled in our neural logistic bandit analysis, where the bottleneck is to control the self-normalized error $Z_t:=\|s_t\|_{H_t^{-1}}$ of the logistic model via data- and variance- dependent Bernstein-type martingale inequality. However, incorporating the assumption **[1]** into our setting would require redoing this martingale inequality on a reduced NTK matrix for unique arms, and carefully tracking how degeneracies of the full Gram matrix interact. We therefore view combining their relaxed NTK assumptions with our refined logistic analysis as a complementary and interesting direction for future work.
>
>
> - **[2]** adopts a regression oracle framework with a 'realizability assumption': there exists a neural network $f^\star$ in the chosen class that exactly matches the conditional mean reward for every context–arm pair. Under this assumption, they obtain bandit regret bounds where the complexity of the function class is absorbed into the oracle, so the leading term does not display an explicit dependence on $\widetilde{d}$. In contrast, we study NTK-based neural bandits without assuming access to such a realizable regression oracle. Therefore, we control the self-normalized error directly under bandit feedback, which naturally leads to a dependence on the effective dimension $\widetilde d$.

---

> ### Author Response · Authors · 2025-11-21
>
> ### [W3] Regret still depends polynomially on S Both main theorems include factors like $S^2 \widetilde{d} \sqrt{\kappa T}$ and even higher-order terms in S (up to $S^5$ in NeuralLog-UCB-2). S measures how far the optimal parameter is from initialization.
>
> As we already emphasize in Remark 2 and in Section 8, reducing the dependence on $S$ has also been recognized as an important goal in recent logistic bandit works [5,6]. Both [5,6] succeed in removing the polynomial dependence on $S$ from the leading term of the regret bound. However, their techniques do not transfer directly to our neural logistic bandit setting. In particular, [5] relies on a nonconvex optimization subroutine, and the PAC-Bayes analysis with a uniform prior in [6] does not yield data-adaptive regret bounds. For these reasons, neither approach can be readily plugged into our NTK-based neural logistic bandit framework.
>
> We therefore fully agree with the reviewer that reducing the dependence on $S$ is an important and interesting future work direction. At the same time, extending such $S$-free techniques to the neural bandit setting appears to be a non-trivial problem that we regard as beyond the scope of the present work.
>
> ----
>
> - [1] Filippi et al., Parametric Bandits: The Generalized Linear Case, NIPS, 2010.
> - [2] Faury et al., Improved optimistic algorithms for logistic bandits, ICML, 2020.
> - [3] Zhou et al., Neural Contextual Bandits with UCB-based Exploration, ICML, 2021.
> - [4] Verma et al., Neural Dueling Bandits, ICLR, 2025.
> - [5] Sawarni et al., Generalized Linear Bandits with Limited Adaptivity, NIPS, 2024.
> - [6] Lee et al., A Unified Confidence Sequence for Generalized Linear Models, with Applications to Bandits, NIPS, 2024.

---

### Official Review · Reviewer_NggP · 2025-11-03

**Soundness:** 3
**Presentation:** 4
**Contribution:** 3
**Rating:** 6
**Confidence:** 3

**Summary:**

This paper studies the neural logistic bandit problem and proposes two algorithms with high-probability regret guarantees. The first algorithm achieves a regret of order $\tilde O(\tilde d\sqrt{\kappa T})$, improving the $\kappa$-dependence over prior neural logistic work (notably Verma et al. 2025); the second algorithm achieves a bound of order $\tilde O(\tilde d\sqrt{T/{\kappa^*}})$, i.e. a second order bound where the leading term is independent of the worst-case $\kappa$. To make these results dimension-adaptive (i.e. depend on the effective dimension $\tilde d$ rather than the ambient parameter dimension $p$, the paper introduces a new variance- and data-adaptive self-normalized martingale inequality, which can be viewed as combining the variance-aware idea of Faury et al. (2020) with a Freedman-style argument to remove the explicit $d$ tern in the confidence radius. This new concentration tool is then plugged into a NeuralUCB-style NTK analysis to obtain the final regret bounds.

In general I would give a weakly accept. The reasons are: strong and interesting concentration tool + improved regret guarantees in a neural logistic setting; assumptions and width requirement are heavy, and the algorithm needs a non-observable radius $S$, but these are common pain points in NTK-based bandit papers, and the contribution is still significant for ICLR.

**Strengths:**

1. Technically solid and interesting concentration result.
The new Bernstein-type, self-normalized martingale inequality is elegant and, as far as I can check, correct. It simultaneously achieves (i) variance adaptivity (using the true logistic variance instead of the worst-case $1/\kappa$) and (ii) data adaptivity (via the log-det / effective-dimension term), thereby bridging the gap between the variance-aware inequality of Faury et al. (2020), which still kept an explicit $d$, and the neural logistic analysis of Verma et al. (2025), which still carried a $\sqrt{\kappa}$ factor. The proof via Freedman is clearly laid out.

2. Improved regret guarantees.
The paper gives, to my knowledge, the first regret bounds for neural logistic bandits that (a) match the best-known $\kappa$-dependence from the logistic bandit literature and (b) replace the ambient dimension by the effective dimension $\tilde d$. This is a meaningful step because naïvely lifting logistic-bandit analysis to the neural setting reintroduces dependence on the number of parameters $p$, which is exactly what the authors avoid here.

3. Nontrivial adaptation of previous techniques.
Even though the analysis is clearly inspired by Faury et al. (2020), Abeille et al. (2021), and Jun et al. (2021), extending those ideas to the NTK-based neural setting, and making the variance-adaptive matrix compatible with the evolving, data-dependent regularization $\lambda_t$, is nontrivial. The proofs for Algorithms 1 and 2 look technically careful.

**Weaknesses:**

1. Algorithm requires knowing $S$.
The algorithm needs the learner to input the norm parameter $S$ (Condition 4.4: “set $S$ as a norm parameter satisfying $S \ge \sqrt{2 h^\top H^{-1} h}$”) but this quantity is defined in terms of the true latent reward vector $h$ and the NTK matrix over all future contexts — not something the learner can observe or compute online. So in practice this is an assumption, not an implementable step. The paper itself later notes that removing the dependence on $S$ is an open direction.

2. Network width requirement is unrealistically large.
The NTK part uses the usual “wide network $\Rightarrow$ linearization” argument, but the concrete requirement on $m$ grows polynomially in $T$, $K$, and $L$ (see Condition C.2), which makes the guarantee more asymptotic than finite-sample. In realistic settings one would not take $m = \Omega(T^4 K^4 L^6 / \lambda_H^4)$ just to run a bandit algorithm.

3. Empirical section is simplified.
To make things practical, the experiments diagonalize the design matrices and simplify baselines (dropping projection steps that are actually needed for the theory), so the empirical comparison is a bit informal and doesn’t fully test the theoretical contribution in a high-dimensional or large-$m$ regime. I won’t hold this strongly against the paper, but it means the experiments currently play more of an illustrative than a validating role.

4. Computational cost is high.
As written, the algorithm repeatedly computes gradients $g(x;\theta_0) \in \mathbb{R}^p$ and maintains design matrices / inverses in the $p$-dimensional tangent space. Even with diagonal approximations (as in the experiments) this is at least linear in $p$, and the theoretical version is more like $O(p^2)$–$O(p^3)$ per update because of the log-det / inverse terms. This contrasts with some recent logistic bandit algorithms whose per-round cost is only $O(d^2)$ or even linear in $d$.

5. Minor but real issue in Theorem 3.1 statement.
In the current draft, the statement/proof of Theorem 3.1 seems to interchange $N$ and $L$ in a couple of places — e.g., the bound uses $|x_t|_2 \le N$ but later the displayed inequality has an $L$-like symbol in the multiplicative constant. This is probably a notational slip, but it should be fixed since Theorem 3.1 is a main technical contribution.

**Questions:**

1. I'm not familiar with the NTK analyses but I find it intuitively hard to understand why would the requirement on the width of the network would scale with the number of layers $L$ and the horizon size. Intuitively I believe if one has wide networks in each layer then the number of layers could be small. My intuition for NTK-style bandits is that once the NTK is “frozen” at initialization, the width should depend on the richness of the context set, not on the time horizon — but here you have a dependence like $m \gtrsim T^4 K^4 L^6$ (Condition C.2).
2. The train–update subroutine uses a number of gradient descent steps $J$ that depends on $T$ (and on $\lambda_t$). This makes the algorithm non-anytime. Is this purely for proof convenience (i.e., could we make it depend on current time $t$ instead and keep the same order), or is there a real obstacle to making it fully online? What do Zhou et al. (2020) and Verma et al. (2025) do here?

3. Could you provide a short intuition for why your Freedman-based argument manages to keep both the variance term (like in Faury et al. 2020) and the log-det term (like in Abbasi-Yadkori et al. 2011) without reintroducing an explicit $d$? Right now the proof is clear but a bit “black-boxy”; adding intuition would make it more readable.

4. Can you compare the per-round (or per-update) computational complexity of NeuralLog-UCB-1/2 with (i) Faury et al. (2020), (ii) Abeille et al. (2021), and (iii) Verma et al. (2025)? Right now the proposed method seems to require operations in $\mathbb{R}^p$, while part of the motivation was to avoid explicit dependence on $p$ — so a short discussion of how to implement the algorithm efficiently (e.g., diagonal / low-rank approximations, periodic updates, NTK feature caching) would strengthen the paper.

---

> ### Author Response · Authors · 2025-11-21
>
> ### [W1] Algorithm requires knowing $S$. The algorithm needs the learner to input the norm parameter $S$ (Condition 4.4: “set $S$ as a norm parameter satisfying $S\geq \sqrt{2h^\top H^{-1} h}$”) but this quantity is defined in terms of the true latent reward vector $h$ and the NTK matrix over all future contexts — not something the learner can observe or compute online. So in practice this is an assumption, not an implementable step. The paper itself later notes that removing the dependence on $S$ is an open direction.
>
> We agree with the reviewer that requiring prior knowledge of a norm parameter $S$ satisfying $S\geq \sqrt{2h^\top H^{-1}h}$ can be a challenge.
>
>
>
> However, we first note that knowing $S$ is analogous to the standard assumption in contextual linear bandits that the true parameter $\theta^\*$ satisfies $\|\theta^\*\|_2 \leq S$ and that the bound $S$ is known to the learner (see, e.g., [1, 2]). In our setting, the role of $S$ is similar. It is a norm parameter that appears in the confidence bounds and regret analysis, and can be viewed as a theoretical assumption rather than a quantity that can be computed exactly in prior.
>
>
>
> To address the issue raised by the reviewer, we believe that the approach proposed in [3] suggests an interesting direction for future work. The authors of [3] show that, in the contextual linear bandit setting, one can avoid knowing the norm parameter $S$ by carefully designing a decaying time-dependent regularizer $\lambda_t$ so that $\sqrt{\lambda_t} S \leq R\sqrt{d}$ holds, where $R$ is the parameter of the $R^2$-sub-Gaussian noise. This construction eliminates the explicit dependence on $S$ in the confidence bounds. Investigating whether a similar strategy can be adapted to the neural (logistic) bandit setting, so as to remove the explicit dependence on the exact value of $S$ in the algorithm, is an interesting and non-trivial direction for future work.
>
> ----
>
> ### [W2] Network width requirement is unrealistically large. The NTK part uses the usual “wide network $\Rightarrow$ linearization” argument, but the concrete requirement on $m$ grows polynomially in $T$, $K$, and $L$ (see Condition C.2), which makes the guarantee more asymptotic than finite-sample. In realistic settings one would not take  $m=\Omega(T^4K^4L^6/\lambda_H^4)$ just to run a bandit algorithm.
>
> We acknowledge the discrepancy raised by the reviewer between theory and practice, which arises when constructing a wide neural network to deduce the standard neural bandit analysis. Such a discrepancy is well-known in the neural bandit literature, which stems from limitations of the current NTK theory. However, our experiments show that our algorithms perform well even when the network width does not satisfy the theoretical condition, suggesting that the gap can be mitigated in practice. We leave the problem of narrowing this theory-practice gap as future work.
>
> ----
>
> ### [W3] Empirical section is simplified. To make things practical, the experiments diagonalize the design matrices and simplify baselines (dropping projection steps that are actually needed for the theory), so the empirical comparison is a bit informal and doesn’t fully test the theoretical contribution in a high-dimensional or large-$m$ regime. I won’t hold this strongly against the paper, but it means the experiments currently play more of an illustrative than a validating role.
>
> We agree with the reviewer that our current experimental section can be viewed as oversimplified, especially since we simplify some of the implementation details for computational and practical reasons.
>
> Although fully addressing these issues is difficult under computational constraints in practice, we have tried to compensate by providing broader illustrative evidence, as such results can still convey useful intuition about the behavior and practical applicability of neural bandit algorithms. First, we introduce the Thompson sampling-based variants of our two UCB algorithms, NeuralLog-TS-1 and NeuralLog-TS-2 in Section I. For NeuralLog-TS-1, we provide a brief regret analysis and show that it achieves the same order of regret as NeuralLog-UCB-1. For NeuralLog-TS-2, we explain in Remark 3 that obtaining a regret bound matching that of NeuralLog-UCB-2 is non-trivial, and we highlight this as an interesting direction for future work. In addition, we compare Algorithm 1, Algorithm 2, their Thompson sampling variants, and the baseline methods on new simulated latent reward functions ($h_4(x)= 10(x^\top \theta)^2$, $h_5(x) = x^\top \Theta^\top \Theta x$, $h_6(x) = \cos (3 x^\top \theta)$) and additional real-world UCI datasets ('magic', 'banknote', 'phoneme'). The experimental details and results are reported in Section J.

---

> ### Author Response · Authors · 2025-11-21
>
> ### [W4] Computational cost is high. As written, the algorithm repeatedly computes gradients $g(x;\theta_0)\in\mathbb{R}^p$ and maintains design matrices / inverses in the $p$-dimensional tangent space. Even with diagonal approximations (as in the experiments) this is at least linear in $p$, and the theoretical version is more like $O(p^2)$–$O(p^3)$ per update because of the log-det / inverse terms. This contrasts with some recent logistic bandit algorithms whose per-round cost is only $O(d^2)$ or even linear in $d$.
>
> We agree with the reviewer that our neural bandit algorithms are computationally more demanding, since they repeatedly compute gradients $g(x;\theta_0)\in\mathbb{R}^p$ and maintain statistics in the $p$-dimensional tangent space, which leads to $\mathcal{O}(p^2)$, and can be a bottleneck in real-world applications. However, we can view this as the natural price for working with a complicated, non-linear real-world reward model while still obtaining provable regret guarantees. By contrast, the logistic bandit algorithms with per-round cost linear in $d$ is indeed more lighter than neural bandit, but rely on the much stronger assumption that the latent reward is linear in a $d$-dimensional space. In this sense, we claim that a higher computational cost is a characteristic trade-off of the neural bandit framework rather than a flaw.
>
> ----
>
> ### [W5] Minor but real issue in Theorem 3.1 statement. In the current draft, the statement/proof of Theorem 3.1 seems to interchange $N$ and $L$ in a couple of places — e.g., the bound uses $|x_t|_2\leq N$ but later the displayed inequality has an $L$-like symbol in the multiplicative constant. This is probably a notational slip, but it should be fixed since Theorem 3.1 is a main technical contribution.
>
> We sincerely thank the reviewer for catching this typo. We have corrected the incorrect use of $L$ and replaced it with $N$ in Section 3 and Section D (red marked).
>
> ----
>
> ### [Q1] I'm not familiar with the NTK analyses but I find it intuitively hard to understand why would the requirement on the width of the network would scale with the number of layers $L$ and the horizon size. Intuitively I believe if one has wide networks in each layer then the number of layers could be small. My intuition for NTK-style bandits is that once the NTK is “frozen” at initialization, the width should depend on the richness of the context set, not on the time horizon — but here you have a dependence like $m\gtrsim T^4K^4L^6$ (Condition C.2).
>
> We believe the reviewer’s intuition comes from the kernelized bandit setting, where once a kernel is fixed, its complexity is governed by the richness of the context set rather than directly by the horizon $T$. However, the condition $m \gtrsim T^4 K^4 L^6$ in Condition C.2 arises when we ask a finite-width neural network trained for $T$ rounds to approximate the corresponding infinite-width NTK model uniformly over all context–arm pairs and times, so a somewhat different line of reasoning is required. We elaborate on each factor below.
>
> - The $T$ and $K$ factors come from requiring that the finite-width network approximate the NTK uniformly over all context–arm pairs up to horizon $T$ and that the resulting Gram matrix be well-conditioned. As in [4], we define the NTK matrix $H$ on the full set $(x^i)$ for $i=1,\dots TK$. Lemma C.1 of [4] gives an entrywise approximation guarantee for a single pair: for $i,j\in[TK]$, if $m=\Omega(L^6\log(L/\delta)/\epsilon^4)$, then with probability at least $1-\delta$ we have $\lvert \langle g(x^i;\theta_0), g(x^j;\theta_0)\rangle/m - H_{i,j}\rvert \le \epsilon$. To make this hold for all entries of the $TK \times TK$ Gram matrix $G^\top G$, Lemma B.1 of [4] applies a union bound over $(TK)^2$ pairs, leading to the bound $\lVert G^\top G - H\rVert_F \le TK \epsilon$. Lemma 5.1 of [4] then enforces $G^\top G \succeq (\lambda_H/2) I$ so that the linearization around $\theta_0$ is stable. This is guaranteed if $\lVert G^\top G - H\rVert_F \le \lambda_H/2$, which is achieved by choosing $\epsilon = \lambda_H/(TK)$. Substituting this $\epsilon$ back into the width requirement from Lemma C.1 of [4] yields $m = \tilde{\Omega}(T^4 K^4 L^6)$. Thus, the dependence on $T$ and $K$ comes from demanding that a finite-width network approximate the infinite-width NTK uniformly over the entire $TK$ context set.
>
> - For the $L$ factor, recall Lemma C.1 of [4] in the previous bullet. For finite $m$, the discrepancy between the empirical kernel and the infinite-width NTK becomes more sensitive as $L$ grows, and the theorem quantifies how much $m$ must increase, polynomially in $L$, to keep this discrepancy below $\varepsilon$. The factor $L^6$ comes from this NTK result via bounds on gradient norms, layerwise Lipschitz constants, and a union bound over layers and parameters. The detailed proof is given in Theorem 3.1 of [5].

---

> ### Author Response · Authors · 2025-11-21
>
> ### [Q2] The train–update subroutine uses a number of gradient descent steps $J$ that depends on $T$ (and on $\lambda_t$). This makes the algorithm non-anytime. Is this purely for proof convenience (i.e., could we make it depend on current time $t$ instead and keep the same order), or is there a real obstacle to making it fully online? What do Zhou et al. (2020) and Verma et al. (2025) do here?
>
> We clarify that letting the number of gradient descent steps $J$ depend on the horizon $T$ (rather than the current round $t$) is purely for notational convenience, so that $J$ (and the step size $\eta$) can be treated as a fixed value. There is no analytic difference by replacing $T$ in the definitions of $J$ and $\eta$ by $t$, and the regret guarantees would remain unchanged. We also note that [4] similarly defines $J$ and $\eta$ using the horizon $T$, while [6] skips a detailed proof for the gradient descent updates.
>
> ----
>
> ### [Q3] Could you provide a short intuition for why your Freedman-based argument manages to keep both the variance term (like in Faury et al. 2020) and the log-det term (like in Abbasi-Yadkori et al. 2011) without reintroducing an explicit $d$? Right now the proof is clear but a bit “black-boxy”; adding intuition would make it more readable.
>
> Compared with our approach, the previous method for reducing the $\kappa$ factor originates from [2], whose authors leverage a Bernstein-like condition on martingale increments to obtain a tighter bound. Their proof begins by defining, for each $\xi\in\mathbb{R}^d$ and $t\ge1$, a nonnegative supermartingale $M_t(\xi)$, and then applying the pseudo-maximization principle to introduce another nonnegative supermartingale $\bar M_t := \int M_t(\xi)\, dh(\xi)$, where $h(\xi)$ is the density of a normal distribution. However, using Bernstein’s inequality restricts the support of $\xi$ to the $\ell_2$ unit ball $\mathbb{B}_2(d)$, and truncating the normal distribution induces a normalization factor that depends explicitly on $d$.
>
> In contrast, we bypass this $d$-dimensional vector-level problem by applying Freedman’s inequality to a scalar martingale $(\ell_i)$ on $i$ (defined in Sections D.1 and D.2) that is constructed to track the growth of the self-normalized error $Z_t := \|s_t\|_{H_t^{-1}}$. As a result, we obtain an inequality that depends on $\widetilde d$, and an improved result on $\kappa$, as Freedman’s inequality is variance-sensitive.
>
> We add a paragraph to discuss and explain this intuition at the end of Section 3.

---

> ### Author Response · Authors · 2025-11-21
>
> ### [Q4] Can you compare the per-round (or per-update) computational complexity of NeuralLog-UCB-1/2 with (i) Faury et al. (2020), (ii) Abeille et al. (2021), and (iii) Verma et al. (2025)? Right now the proposed method seems to require operations in $\mathbb{R}^p$, while part of the motivation was to avoid explicit dependence on $p$ — so a short discussion of how to implement the algorithm efficiently (e.g., diagonal / low-rank approximations, periodic updates, NTK feature caching) would strengthen the paper.
>
> - We have compared the per-round computational complexity of all algorithms and summarized the results in Section K. In particular, our two algorithms have complexity $\mathcal{O}(J_t t p + K p^2 + p^2)$, where $J_t$ is the number of gradient step where $J_t = \widetilde{\mathcal{O}}(TL/\lambda_t)$. We also elaborated the details to obtain the complexity of our algorithms in Section K. Algorithm (i) has complexity $\mathcal{O}(d^2 K + d^2 T)$, algorithm (ii) has complexity $\mathcal{O}(d^2 K T)$, although algorithm (iii), developed for neural dueling bandits, has the same complexity as ours. Also, the algorithm due to Faury et al. (2022) shows $\widetilde{\mathcal{O}}(d^2 K)$, which is efficient.
>
> - We view the reviewer’s question of whether one can avoid the direct dependence on $p$ not only in the regret bound (as we do) but also from the computational complexity as a very interesting open problem in the context of neural bandits, and we believe it constitutes an excellent direction for future work.
>
> As one illustrative example, when we consider the connection between NTK-based neural bandits and kernelized bandits, it is natural to consider applying techniques such as Nyström approximation, as in [7] to obtain faster computational complexity in the neural bandit setting. We have added this discussion to Section K.
>
>
> - Another approach to reducing computational complexity is to adapt the method proposed in [8]. This work also tackles contextual bandits via an NTK-based neural bandit formulation, but the neural network is trained so that its output is not the reward itself, but instead a $d$-dimensional feature vector. The problem is then reduced to solving a linear bandit in this learned feature space with respect to an unknown parameter $\theta^*$. This is an effective way to reduce computational complexity and has clear advantages from an application standpoint, but it requires an additional Lipschitz-like assumption on the neural network on the theoretical side. We have also added this discussion to Section K.
>
> ----
>
> - [1] Yabbasi-yadkori et al., Improved Algorithms for Linear Stochastic Bandits, NIPS, 2011.
> - [2] Faury et al., Improved optimistic algorithms for logistic bandits, ICML, 2020.
> - [3] Gales et al., Norm-Agnostic Linear Bandits, AISTATS, 2022.
> - [4] Zhou et al., Neural Contextual Bandits with UCB-based Exploration, ICML, 2021.
> - [5] Arora et al., On Exact Computation with an Infinitely Wide Neural Net, NIPS, 2019.
> - [6] Verma et al., Neural Dueling Bandits, ICLR, 2025.
> - [7] Zenati et al., Efficient Kernel UCB for Contextual Bandits, AISTATS, 2022.
> - [8] Xu et al., Neural Contextual Bandits with Deep Representation and Shallow Exploration, ICLR, 2022.

---

### Official Review · Reviewer_tUn9 · 2025-11-12

**Soundness:** 3
**Presentation:** 4
**Contribution:** 3
**Rating:** 6
**Confidence:** 4

**Summary:**

The paper studies the neural logistic bandits problem, where the rewards are binary and depend on a latent, non-linear reward function that is estimated using neural networks.
Existing methods depend on the minimum reward variance and feature dimension, making them impractical for high-dimensional settings. This paper proposes a novel Bernstein-type concentration inequality for self-normalized vector-valued martingales and then uses it to derive regret upper bounds that scale with the effective dimension (a data-dependent, typically much smaller quantity than the feature dimension) rather than the full network size.

The authors propose two new algorithms, NeuralLog-UCB-1 and NeuralLog-UCB-2, both achieving improved or matching best-known regret rates for neural logistic bandits, while fully removing dependence on worst-case variance and feature dimension. The authors also empirically validated the superior performance of the proposed algorithms on synthetic and real-world datasets.

**Strengths:**

**Strengths of the paper:**
1. This paper considers the neural logistic bandits problem, where an unknown latent non-linear reward function is estimated using neural networks.

2. This paper proposes a novel Bernstein-type concentration inequality for self-normalized vector-valued martingales and then uses it to derive tighter regret upper bounds.

3. The authors propose two new algorithms, NeuralLog-UCB-1 and NeuralLog-UCB-2, both achieving improved existing regret bounds, closing the gap between theory and practice.

4. The authors also validate the empirical performance of the proposed algorithms on synthetic and real-world datasets, showing superior cumulative regret performance over key baselines.

**Weaknesses:**

**Weaknesses of the paper:**
1. It is unclear how one can choose the right architecture of a neural network (NN) to estimate the underlying unknown reward function. If the NN architecture (too small or too large) is good enough for estimating the reward function, it may lead to mis-specification.

2. The empirical results can also include Thompson sampling-based variants of the proposed algorithms. Also, the authors can mention the key challenges to integrating Thompson sampling with the proposed algorithms.

**Questions:**

Please address the paper's weaknesses. I am open to changing my score based on the authors' responses.

**Details Of Ethics Concerns:**

Since this work is a theoretical paper, I do not find any ethical concerns.

---

> ### Author Response · Authors · 2025-11-21
>
> ### [W1] It is unclear how one can choose the right architecture of a neural network (NN) to estimate the underlying unknown reward function. If the NN architecture (too small or too large) is good enough for estimating the reward function, it may lead to mis-specification.
>
> Our analysis is based on a fully-connected neural network with input dimension $d$, width $m$, and number of layers $L$ that is wide enough to satisfy the standard theoretical conditions for neural bandits. However, reflecting the requirement that $m$ grows polynomially in $T,K,L$ in practical implementations is clearly unrealistic. We therefore agree with the reviewer that, in practice, one faces the nontrivial question of how to choose an appropriate neural network width. Such a discrepancy between the required network widths in theory and those used in practice is well-known in the neural bandit literature and stems from limitations of the current NTK theory. Nevertheless, our experiments show that our algorithms perform well even when the network width does not satisfy the theoretical condition, suggesting that this gap can be mitigated in practice. We leave the problem of narrowing this theory-practice gap as an important direction for future work.
>
> ----
>
> ### [W2] The empirical results can also include Thompson sampling-based variants of the proposed algorithms. Also, the authors can mention the key challenges to integrating Thompson sampling with the proposed algorithms.
>
> We thank the reviewer for the suggestion. Based on the comment, we have added Section I in the appendix to develop and analyze some Thompson sampling-based variants of our algorithms. To be more specific, we introduce the Thompson sampling-based variant of Algorithm 1 (NeuralLog-UCB-1), denoted NeuralLog-TS-1, and provide a brief regret analysis. We show that the regret of NeuralLog-TS-1 has the same order as that of NeuralLog-UCB-1, namely $\widetilde{\mathcal{O}}(\widetilde{d}\sqrt{\kappa T})$. We also present the Thompson sampling-based variant of Algorithm 2 (NeuralLog-UCB-2).
>
>
> However, as we explain in Remark 3, there is a challenge in obtaining the same order of regret as NeuralLog-UCB-2, in particular the same dependency on $\kappa$. To elaborate, as described in Section 6.2 (Proof sketch of Theorem 5.2), the improved regret of NeuralLog-UCB-2 is obtained by analyzing the per-round regret $\mu(h(x_t^\*)) - \mu(h(x_t))$ via a second-order Taylor expansion. Applying this expansion yields
> $\mu(h(x_t^\*)) - \mu(h(x_t))\leq \dot\mu(h(x_t)) g(x_t;\theta_0)^\top (\widetilde\theta_{t-1}-\theta^\*) + [g(x_t;\theta_0)^\top(\widetilde\theta_{t-1}-\theta^\*)]^2$.
> Focusing on the first term on the right-hand side, we consider the design matrix based on the neural network-estimated variance,
> $W_t=\sum_{i=1}^t \frac{\dot{\mu}(f(x_i;\theta_i))}{m} g(x_i;\theta_0)g(x_i;\theta_0)^\top$,
> and its associated feature vectors $\sqrt{\dot\mu(f(x_t;\theta_t))} g(x_t;\theta_0)$. Using these, we can rewrite the first term as
> $\sqrt{\dot{\mu}(h(x_t))}\lVert\sqrt{\dot\mu(f(x_t;\theta_t))} g(x_t;\theta_0)\rVert_{W_{t-1}}\lVert\widetilde \theta_{t-1}-\theta^*\rVert_{W_{t-1}^{-1}}$
> through additional manipulation. This allows us to move the factor $\sqrt{\dot\mu(f(x_t;\theta_t))}$, which controls the $\kappa$-dependence, inside the norm and thereby reduce the $\kappa$ factor in the resulting regret bound.
>
>
> In contrast, as can be seen in [1,2], the analyses of Thompson sampling proceed by bounding the expected difference between the reward of the Thompson-sampled arm and the reward obtained by the optimal arm. Since this per-round regret is controlled via such an expectation-based argument, it is challenging to directly apply the above Taylor-expansion-based technique in the Thompson sampling setting. In summary, obtaining an improved regret bound for NeuralLog-TS-2 that matches the improved regret of NeuralLog-UCB-2 is non-trivial, and we leave this as an important direction for future work.
>
> Regardless of theoretical discussions, both Thompson sampling-based variants (NeuralLog-TS-1, NeuralLog-TS-2) are well defined at the algorithmic level, and therefore, we include empirical results for them together. Since a rigorous proof for NeuralLog-TS-2 is not yet available, we do not incorporate these results into the main experiment figure. Instead, we implement additional simulated latent reward functions ($h_4(x)= 10(x^\top \theta)^2$, $h_5(x) = x^\top \Theta^\top \Theta x$, $h_6(x) = \cos (3 x^\top \theta)$) and additional real-world UCI datasets ('magic', 'banknote', 'phoneme') to illustrate their empirical performance, which covers the dataset used in the experiment of the previous work of [3]. The experimental details and results are reported in Section J.
>
> ----
>
> - [1] Zhang et al., Neural Thompson Sampling, ICLR, 2021.
> - [2] Verma et al., Neural Dueling Bandits, ICLR, 2025.
> - [3] Zhou et al., Neural Contextual Bandits with UCB-based Exploration, ICML, 2021.

---

### Author Response · Authors · 2025-12-03
**Final Remarks**

We have responded to all reviewer comments to clarify the problem setting and further elaborating on our contributions. We also implemented the suggestion to consider Thompson sampling-based variants of our algorithms. Furthermore, we have conducted new numerical experiments to test real-world datasets. Accordingly, we have updated our manuscript. Our response and updates to the paper are summarized as follows.

- **Main challenge/novelty/intuition for our new Bernstein-type tail inequality [R2-Q3, R3-W1, R4-W2]**: The main challenge is to simultaneously obtain improved dependence on both $\widetilde d$ and $\kappa$. Since none of the existing works on logistic and neural bandits, as well as any direct combination of their results, achieves this, we develop a novel data- and variance-dependent Bernstein-type tail inequality that integrates non-uniform variance into the design matrix to reduce $\kappa$ and, by applying Freedman’s inequality to a scalar martingale, avoids explicit dependence on $d$.

- **Gap between theoretical and practical requirements for the neural network width $m$ [R1-W1, R2-W2, R4-Q2]**: In our experiments, we found that a practical choice of $m$ performs well. Thus, our experimental results suggest that the well-known theory-practice gap in neural bandits can be mitigated.

- **Experimental improvements with new algorithms (Thompson sampling–based variants), new simulations, and real-world datasets [R1-W2, R2-W3, R4-W3]**: We have added Section I, where we analyze two Thompson sampling–based variants of our algorithms. In Section J, we present additional experiments on new simulations and real-world datasets, including all baselines, our algorithms, and their Thompson sampling–based variants.

- **Requirements for prior knowledge of parameters [R2-W1, R4-W1]**: (i) $\kappa$: Unlike Algorithm 1, Algorithm 2 avoids prior knowledge of $\kappa$, yielding a $\kappa$-free UCB algorithm and stronger results. (ii) $S$: Assuming prior knowledge of $S$ is analogous to the standard assumption in contextual linear bandits that the true parameter satisfies $|\theta^*|_2 \leq S$. (iii) $R$, $\lambda$: These can be set easily.

- **Computational cost of neural bandits [R2-W4, R2-Q4]**: We have added Section K, where we compare the per-round computational complexity of Algorithms 1 and 2 with that of previous works and discuss ways to reduce computational complexity.

- **Typo [R2-W5]**: We have corrected the typo in the manuscript.

- **Intuition for the requirements on the neural network width $m$ [R2-Q1]**: We explain in detail that the required scaling of $m$ with $T$, $K$, and $L$ arises from the approximation error of the NTK over all context–arm pairs.

- **Dependence on $T$ in the number of gradient descent steps $J$ [R2-Q2]**: The dependence on the horizon $T$ can be directly replaced by the current round $t$, thereby making the algorithm anytime.

- **Fixed parameter choice for the experiments [R4-Q1]**: In our experiments, we fix $\lambda$ and $\nu$ to practically chosen values following standard practice in neural bandit experiments, which simplifies implementation and yields more stable performance.

- **Comparison with recent neural bandit results and scope of our contribution [R3-W2]**: We first note that our contribution focuses on the standard neural logistic bandit setting, where we obtain regret bounds with strictly improved dependence on $\kappa$ by developing a new Bernstein-type tail inequality, rather than aiming to uniformly dominate all recent neural bandit results, which often target orthogonal goals (e.g., robustness). We then clarify that the two works mentioned by the reviewer as comparison targets---robust neural contextual bandits with relaxed NTK assumptions [1] and realizability-based regression-oracle approaches [2]---operate under different assumptions and problem settings and are therefore complementary to our work, rather than directly comparable baselines in our setting.

- **Regret dependence on $S$ [R3-W3]**: As discussed in Remark 2 and Section 8, no previous work has removed $poly(S)$ from the leading term in neural logistic bandits. We leave this as future work.

----
- [1] Qi et. al., Robust Neural Contextual Bandit against Adversarial Corruptions, NIPS, 2024.

- [2] Deb et. al., Contextual Bandits with Online Neural Regression, ICLR, 2024.

---

### Meta-Review · Area_Chair_NAah · 2026-01-07

**Summary:**

The paper studies the neural logistic bandits problem. The authors develop a variant of Bernstein-style inequality, and use it to design two UCB-based algorithms, NeuralLog-UCB-1 and NeuralLog-UCB-2, with regret bounds grows with the effective dimension. The reviewers raise several concerns.

1. Algorithm requires knowing S and $\lambda$.

2. Gap between theory and practice: network width requirement is unrealistically large; computational cost is high; regret has a high polynomial dependence on S.

3. Empirical studies use a simplified setting.

4. Limited technical novelty.

**Reviewer Concerns:**

Some concerns and limitations in this paper also exist in previous neural bandit literature. Within the neural bandit literature, I think the theory of this paper is solid. But practicality limitations of this setup make the reviewers not too excited about this work. Overall, this is a borderline submission.

**Reviewer Scores:**

6,6,4,4

---

### Decision · Program_Chairs · 2026-01-26

Reject